# Tracking business opportunities for climate solutions using AI in regulated accounting reports

Shirley Lu [1] ✉, George Serafeim[1], Simon Xu[1] & MarcAntonio Awada[2]

The transition to a low-carbon economy offers substantial business opportunities, yet most research focuses on risks. This study develops a metric to identify firms advancing "climate solutions" by applying large language models to 39,710 10-K filings from 4,483 U.S. firms (2005-2022). The metric reveals a rising emphasis on climate solutions, validated by its responsiveness to policy shocks (e.g., Inflation Reduction Act) and correlation with green revenues and innovation indicators. We apply the measure to three inquiries: (i) firms engaged in climate solutions experience higher revenue growth, especially in sectors with strong intellectual property protection and technologies with high abatement potential; (ii) a modest political divide exists, with firms located in states with predominantly Republican voting patterns exhibiting lower climate solutions—a gap that narrows for low-cost technologies; and (iii) seemingly unrelated industries converge around shared technologies, reflected in higher stock return synchronicity. These results illustrate the value of AI-based analysis of regulatory filings for uncovering climate-related business opportunities.

Climate change is often framed as a risk for businesses[1–4]. Carbon pricing regulations, such as the EU Emissions Trading System, impose penalties on high-emitting firms and industries. Climate disclosure mandates, such as the EU Corporate Sustainability Reporting Directive, often require the disclosure of greenhouse gas emissions as a measure of climate risk. Accordingly, the literature has centered on studying the implications of climate risks emerging from regulations, litigation, and technological changes in addition to physical impacts[5–10]. However, the transition to a low-carbon economy is not merely a source of risks; it also presents business opportunities for firms innovating for the climate transition[11]. These opportunities represent a growing market under rising climate transition risks, with examples including electric vehicle production, renewable energy generation, green hydrogen for heavy industry, and plant-based foods.

Climate risks and climate opportunities represent distinct dimensions of the low-carbon transition, with different financial characteristics, policy implications, and industry focus. Firms exposed to climate risks, such as those in high-emitting industries, have incentives to reduce emissions and often incur additional costs to do so[12]. Carbon pricing policies provide incentives for these firms to internalize the social cost of emissions. In contrast, firms pursuing climate opportunities generate value by meeting growing demand for decarbonization technologies and services, creating revenue growth potential[13]. These firms, such as battery producers, need not have high emissions themselves. Unlike emissions externalities, which are addressed through carbon pricing regulations, other policies, such as innovation subsidies, help mitigate the underinvestment in climate innovations subject to knowledge spillovers that benefit other firms[14,15]. Recognizing these distinctions, the Sixth Assessment Report of the IPCC emphasizes the importance of developing and deploying climate solutions and calls for a systemic view of climate innovation to guide effective policy design[16].

Existing research, however, focuses on climate innovation in specific technologies, with limited studies on a systemic view of

[1]Harvard Business School, Harvard University, Boston, MA, USA. [2]Harvard Business School and Harvard Extension School, Harvard University, Boston, MA, USA. ✉e-mail: slu@hbs.edu

climate solutions. For example, studies have examined topics related to energy storage[17–20], solar energy[21,22], and electric vehicles[23,24]. These studies also primarily emphasize technology development, with less attention on how incumbent firms scale and commercialize these solutions—critical for achieving widespread decarbonization[25]. One reason for this gap is the challenge of measurement: firms are not required to disclose climate-related investments, and such spending reflects inputs rather than the development and deployment of climate solutions as products and services[16]. Mandatory disclosure of scope 1 and 2 emissions does not capture these opportunities because companies that capitalize on these opportunities do not necessarily have low emissions themselves, as they typically help other firms lower emissions. Efforts like the EU taxonomy aim to measure green activities through revenue and investment disclosures, but data remains limited given its recent rollout, and companies have found the taxonomy hard to implement[26].

In this paper, we develop a measure of climate solutions that draws on both the capabilities of large language models (LLMs) and uses data that is consistently disclosed over a long time period with regulatory scrutiny: business descriptions from regulated financial filings. We define climate solutions as products and services that develop or deploy technologies in a transition to a low-carbon economy. We present evidence of the prevalence and evolution of climate solutions for 4,483 US public firms across 13 GICS industry groups (47 GICS industries) from fiscal years 2005 to 2022 (reports released in calendar years 2006 to 2023).

Our measure of climate solutions uses generative pre-trained transformers (GPT), an advanced neural network architecture designed for understanding and generating human-like text. Developed by OpenAI, GPT models are characterized by their ability to comprehend contextually relevant text based on their training on a diverse range of sources on the internet. We fine-tune the GPT model to perform the specific task of identifying sentences that relate to climate solutions. For example, our definition of climate solutions includes firms that offer electric vehicles, but excludes firms that use electric vehicles merely in the company fleet to transport company staff. This nuanced distinction complicates the task and requires an algorithm to differentiate between climate-solution topics and general climate-related topics.

We apply the climate solutions GPT model on Item 1 Business Description of the 10-K filing, which is the mandatory disclosure of the core products and services of a company for firms listed on US market exchanges. Focusing on the business description section allows us to focus on the products and services that companies develop and sell, rather than on the changes they make to reduce their carbon emissions in their supply chain or operations. Additionally, 10-K filings are standardized, strictly regulated, monitored by the US Securities and Exchange Commission (SEC), and scrutinized by auditors, lawyers, and courts, making them a trustworthy and comparable dataset to apply our LLM. Following the Sarbanes-Oxley Act in the United States, 10-K filings require a signature from the CEO and CFO of the issuer confirming their personal responsibility for the filing and carry criminal liabilities for misrepresentation. This requirement makes 10-K filings perhaps the most reliable description of actual company business operations, something that does not characterize other company disclosures, such as sustainability reports, corporate websites, or investor conference calls. In summary, the business description section of the 10-K represents a likely ubiquitous, relevant, reliable, and comparable source for measuring a company's engagement in climate solutions.

Applying AI to analyze companies' communication of climate solutions in the business description section of the 10-K allows us to derive measures of climate opportunities for each firm-year. While significant efforts have been made to increase corporate disclosure on the implications of climate change for businesses, through the issuance of sustainability reports and other voluntary disclosures[27,28], this study documents that existing mandatory disclosures contain useful information about climate opportunities. This result is important because efforts for new mandatory climate disclosures are challenged politically and legally[29] and voluntary disclosures suffer from reliability issues[30].

We start with descriptive results on the prevalence of climate solutions across industries and over time. We validate that our AI measure of climate solutions is positively associated with estimates of green revenues, green patents, and other innovation measures, and that the measure varies predictably in response to major policy interventions, such as the Inflation Reduction Act. At the same time, it is weakly correlated with different carbon emissions scopes proxying for a firm's climate risk exposure, suggesting that climate opportunities provide incremental understanding of a firm's climate involvement in addition to climate risks. Next, we use topic analysis to codify these opportunities into different climate technologies, providing a nuanced view of climate solutions by examining characteristics such as carbon abatement potential and cost. Finally, we demonstrate the usefulness of the climate solutions measure and topic analysis by applying the measures to examine three sets of research topics. First, we demonstrate that firms with higher climate solutions experience higher revenue growth, especially in industries with prevalent patent protection and for technologies with higher abatement potential. Second, we observe a gap in climate solutions in firms operating in states with different political affiliations, which is mitigated for climate solutions with lower costs. Third, the study illustrates how apparently dissimilar industries are becoming increasingly interconnected as they focus on the same climate solutions, and that industries focusing on the same climate solutions exhibit higher stock return synchronicity.

## Results
### Prevalence of climate solutions
After employing our climate solutions GPT model to categorize every sentence in the business description as either related to climate solutions or not, we define our primary metric as the climate solutions measure. This measure represents the proportion of sentences identified as climate solutions to the total number of sentences in the 10-K Item 1 Business Description.

Consistent with greater disclosure of climate-solution language in 10-K business descriptions, the climate solutions measure is trending upward over time, growing from 1% in 2005 to 4% in 2022 (Fig. 1a). The percentage of firms with a climate solutions measure exceeding 1% doubled from 20% in 2005 to 45% in 2022, whereas those exceeding 5% quadrupled from 5% in 2005 to 20% in 2022.

Discussion of climate solutions is most prevalent in the automobile and components, utilities, and capital goods industry groups of the economy (Fig. 1b). Within these industry groups, at the industry level, these trends are driven by automobiles (undergoing a transition to electric vehicles), renewable energy providers, and electrical equipment (primarily involved in battery production), respectively. In contrast, hard-to-abate industries show less pronounced transitions. For example, passenger airlines in the transportation industry group show only a moderate transition, likely reflecting the higher costs involved in sustainable aviation fuels[31]. Real estate investment trusts (REITs) exhibit the lowest adoption of climate solutions. This reluctance can be attributed to the slow-to-change nature of the construction industry, where experimentation with new solutions is costly due to the emphasis on safety and stability, and where it takes years to prove a technology[32]. Over time, we observe increases in the climate solutions measure across most industry groups. The upward trend is most consistent for the utilities and materials industry groups. In the most recent years, following support from the IRA, we observe the largest increases in automobile and components, capital goods,

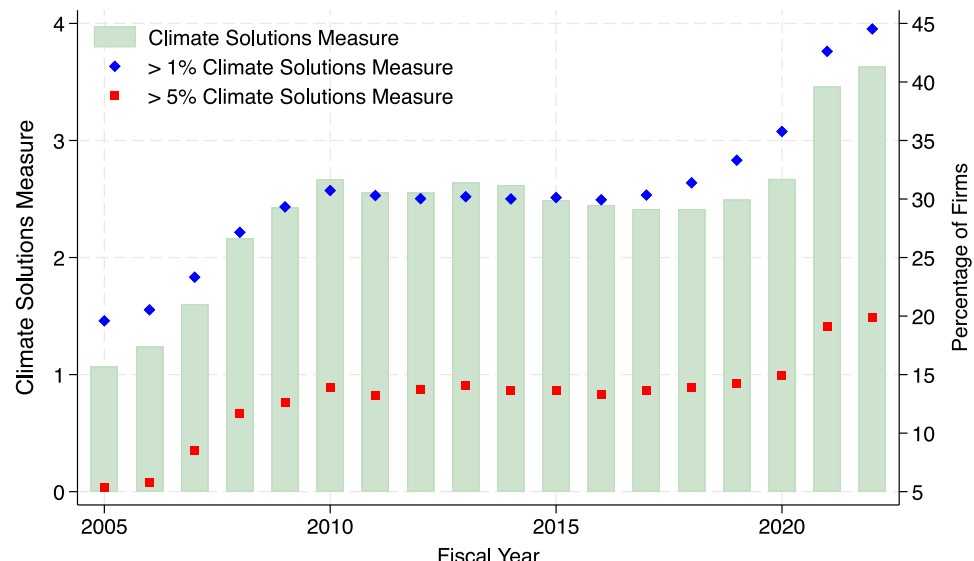

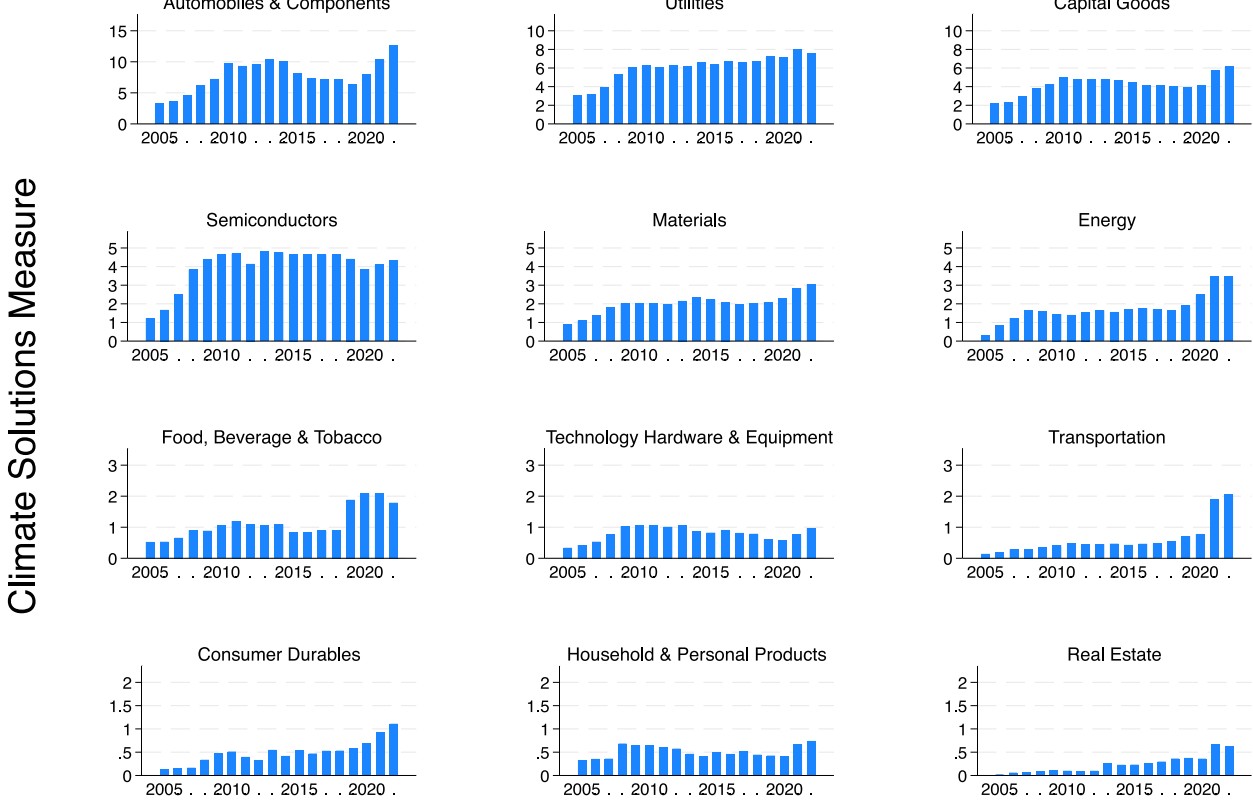

**Fig. 1 | Prevalence of climate solutions. a** Average climate solutions measure for each year and the percentage of firms with over 1% (blue) and 5% (red) climate solutions measure for each year starting fiscal year 2005. **b** This figure plots the average climate solutions measure over fiscal years by GICS industry groups in our sample. We rank the industry group with the highest average climate solutions measure, starting from the top left. To enable visual detection of trends, we reduce the maximum of the vertical axis for each subsequent row of industry graphs. Given the small number of observations and similar climate solutions patterns, we combine industry groups, real estate management & development, and equity real estate investment trusts, in one graph under the label real estate.

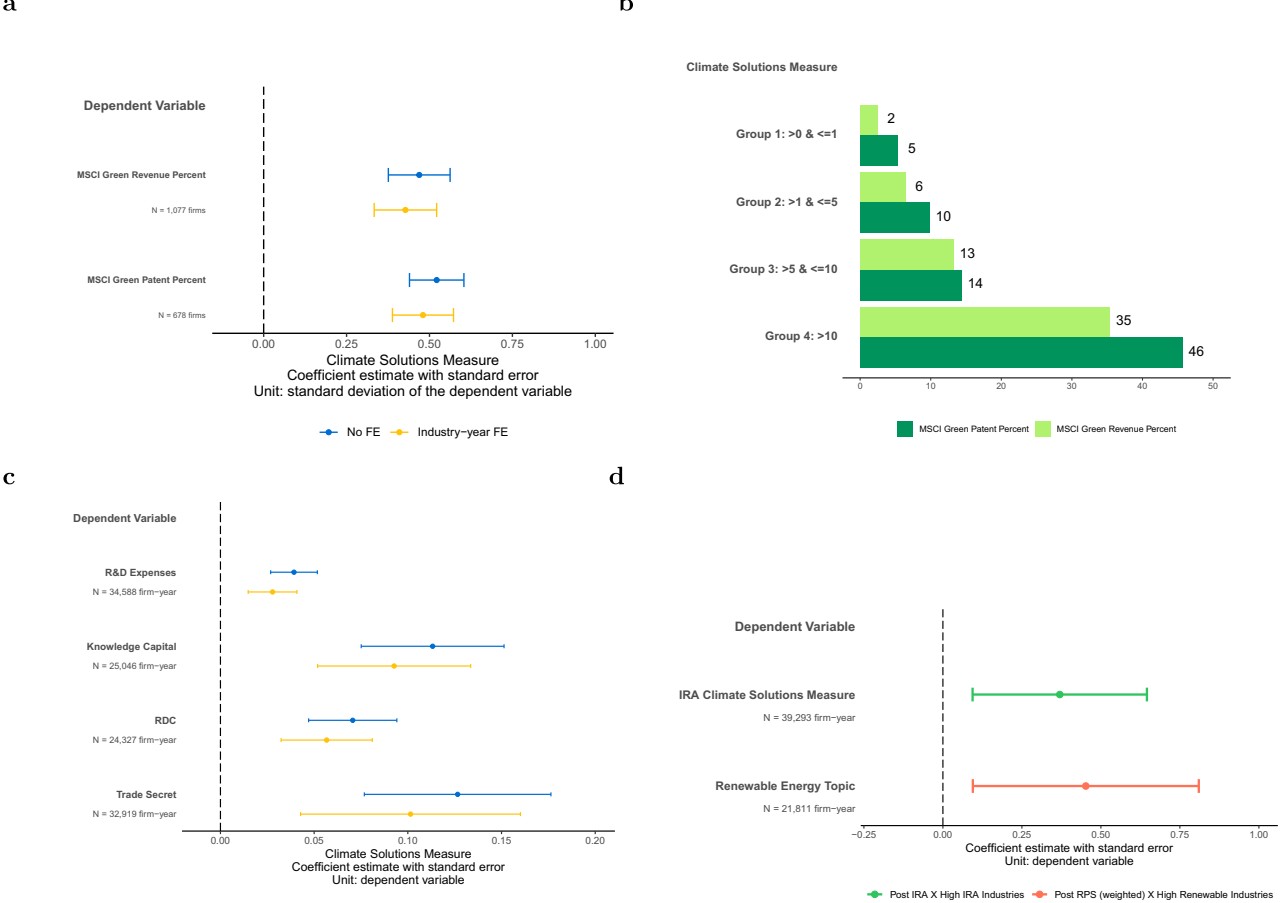

**Fig. 2 | Validation of climate solutions measure. a** Plot of the coefficient estimate and 95th percent confidence interval of regressing climate solutions measure on the percent of green revenue and green patent from MSCI, which is available for the fiscal year 2022. The coefficient reflects the association of one standard deviation change in the climate solutions measure on the dependent variables (in standard deviations). The plot shows specifications with and without industry-year fixed effects. See Supplementary Table 4a for regression details and results. **b** MSCI green revenue and patent percent by groups with different climate solutions measures. **c** Plot of the coefficient estimate and 95th percent confidence interval of regressing innovation measures on climate solutions measure. The coefficient reflects the association of one standard deviation change in the climate solutions measure on the dependent variables. Each plot includes controls for log revenue in period t-1 and firm age, and shows specifications with and without industry-year fixed effects. See Supplementary Table 4b for regression details and results. **d** Plot of the coefficient estimate and 95th percent confidence interval of regressing the relevant climate solutions measure on periods after policy interventions. The blue line reflects the change in IRA-related climate solutions for industries with more IRA coverage after the IRA. The red line reflects the change in renewable climate solutions for industries with more renewable technologies after RPS, which is weighted based on a firm's distribution of employees in each state. Firm fixed effects and year fixed effects are included. See Supplementary Table 4c for regression details and results.

energy, food, beverage and tobacco, transportation, consumer durable, and real estate industry groups.

## Validation of the climate solutions measure

The validation analysis further confirms that the climate solutions measure derived from 10-Ks reflects companies' business portfolio involvement in climate solutions. The climate solutions measure is positively associated with the percentage of green patents and green revenues estimated by MSCI, one of the largest data providers, for the most recent fiscal year with available data (Fig. 2a). A one standard deviation increase in climate solutions measure is associated with a 0.47 and 0.52 standard deviation increase in green revenue and green patent percent, respectively. Additionally, green revenues and green patents measures increase monotonically across groups of increasing levels of climate solutions measure (Fig. 2b). For example, firms with climate solutions measure greater than 0 but less than 1 (Group 1), and greater than 1 but less than 5 (Group 2), have an average green revenue percent of 2 and 6, respectively. This increases to ~13 percent for firms with climate solutions measures greater than 5 but less than 10 (Group

3). Firms with climate solutions measure greater than 10 (Group 4) have a green revenue percent of more than 35. Similarly, green patents increase from ~5 percent for Group 1 to ~46 percent for Group 4.

As a second validation test, we observe that a higher climate solutions measure is associated with higher innovation across four different measures. With industry-year fixed effects, a one standard deviation higher climate solutions measure is associated with a 3% higher research and development expense scaled by revenue, 9% higher knowledge capital scaled by revenue, 6% higher research and development capitalized (RDC) scaled by revenue, and 10% higher trade secret scaled by revenue (Fig. 2c). In the absence of data on climate investments, this positive association provides support that the climate solutions measure reflects firms engaging in more innovation.

In a third validation test, we find that the climate solutions measure varies predictably in response to major events. Most notably, we see the largest jump in the most recent years, which represents the information in fiscal years 2021 and 2022 disclosed in 2022 and 2023, respectively. The Inflation Reduction Act (IRA), passed in 2022,

provided hundreds of billions of dollars in incentives for the development and acceleration of climate solutions. In industries with more IRA opportunities, the climate solutions measure on topics covered by the IRA increases significantly in fiscal years 2021 and 2022, relative to prior years (Fig. 2d). We also observe an increase between 2005 and 2010, during a period where over 20 states in the United States passed the renewable portfolio standards (RPS) that mandate a minimum ratio of renewable energy supply in a state. The climate solutions measure on topics related to renewable energy increase significantly after the staggered adoption of RPS for firms in industries relevant to renewable energy (Fig. 2d).

### Relation between risks and opportunities

Given the increasing prevalence of climate solutions across multiple industries of the economy, we examine whether the opportunities for developing and deploying these solutions are correlated with the risks associated with transitioning to a low-carbon economy. We examine climate transition risks using greenhouse gas emissions, which are often viewed as a company's exposure to climate transition risk, driven by regulatory, legal, and societal sentiment changes[7,33]. We distinguish between a business transition risk perspective on climate change, which focuses on firms with high greenhouse gas emissions, and a business opportunity perspective, based on the climate solutions measure.

Ex-ante, it is not clear whether and how greenhouse gas emissions are correlated with the climate solutions measure. On the one hand, firms in some high-emissions industries, such as utilities and oil and gas, have more innovation opportunities as documented in prior literature through green patents[34]. On the other hand, climate solutions opportunities likely also arise in new industries that do not face high transition risks. For example, industries such as electrical equipment or automobile components have a high climate solutions measure but low scope 1 greenhouse gas emissions intensity (Fig. 3a). In contrast, industries such as passenger airlines, marine transportation, or construction materials have high scope 1 greenhouse gas emissions intensity but a comparatively lower climate solutions measure.

Overall, we observe a moderate relationship between climate transition risks and opportunities (Fig. 3b). Unconditionally on firm industry membership, a one standard deviation increase in climate solutions measure is associated with a 0.15 standard deviation higher scope 1 greenhouse gas emissions intensity. Conditional on industry membership, we do not find a statistically significant association between risk and opportunity. This relationship extends to other scopes of greenhouse gas emissions, and to absolute or intensity measures. Without industry-year fixed effects, the relation between risk and opportunity is statistically significant and varies in direction across different scopes of greenhouse gas emissions, but the magnitudes are modest (less than 0.2 standard deviations). With industry-year fixed effects, most relationships become statistically insignificant.

As an additional supporting measure, we examine a firm's climate risk management using emissions scores from Refinitiv and MSCI in Supplementary Table 5 Panel E. Consistent with the greenhouse gas results, we find that the relationship between emissions scores and climate solutions measure is modest (less than 0.15 standard deviations). Taken together, these empirical observations suggest that climate risks and opportunities are weakly related. Future research can further examine this potential distinction and explore the implications of different firms being exposed to climate risk versus innovating to capture climate opportunities.

### Climate solutions topics

Next, we decompose our overall climate solutions measure into 88 distinct climate solutions topics that provide insights into specific technologies adopted by different firms and industries. Using topic analysis, we assign each sentence to the closest climate solution identified by Project Drawdown. Figure 4a lists the top 30 topics ranked starting from the most common on the left. The most common topics are renewable energy, energy efficiency, and biomass power.

We further group the topics into nine broad categories to observe the distribution of topics across different industry groups. Figure 4b further confirms the validity of the climate solutions measure for climate opportunities, aligning topics to industry value chains. We observe the highest concentration of electrification in the automobile and components industry group, where electric vehicles are the primary climate solution. Topics related to using waste, including the use of biofuels, are concentrated in the energy, food, beverage, and tobacco industry groups. Meanwhile, topics related to renewable energy are concentrated in the utilities and semiconductor and semiconductor equipment industry groups.

The construction of the climate solutions measure provides insights into how companies within industries are evolving their technological and product strategies over time. We report three examples of this phenomenon. First, for the automobile manufacturer industry, we observe a decline in companies' focus on fuel cell and hybrid vehicles and an increasing focus on electric vehicles (Fig. 5a). For the utility industry, in terms of power generation, we observe an increasing emphasis on solar energy and a decline in the focus on geothermal energy and nuclear energy (Fig. 5b). Finally, in the food and beverage industry, we observe an increasing emphasis on plant-based products and less emphasis on biomass power, such as ethanol use (Fig. 5c).

### Carbon abatement costs and abatement potential

Identifying topics allows the analysis of the carbon abatement costs and abatement potential to achieve decarbonization across industries and firms. By aligning topics with the list of climate solutions from Project Drawdown, we can analyze these two characteristics for each climate solution, as Project Drawdown offers estimates of both the abatement potential and the associated costs required to achieve these reductions.

Figure 6a plots each climate solutions topic on these two dimensions, with the size of the marker representing the prevalence of the topic in 10-K Item 1. In general, there are more climate solutions that have higher abatement potential and lower abatement costs. One exception is electric cars, which have moderate abatement potential but relatively high costs. The higher adoption of electric cars likely reflects regulatory requirements to phase out internal combustion engines, such as California's rule requiring all new car sales after 2035 to be zero-emission vehicles, and federal tax refunds incentivizing their purchase. Figure 6b plots each industry on these two dimensions, with the size of the marker representing the average climate solutions measure. Most industries with high climate solutions measure concentrate around high abatement potential and low cost per abatement, such as in semiconductors and semiconductor equipment, and electrical equipment industries. Again, the exception is in the automobile industry, consistent with the observation that electric cars have relatively higher costs per abatement.

### Application 1: Climate solutions and revenue growth

If the climate business opportunities communicated in the business description section of the 10-K indicate a company's capacity to commercialize climate solutions, and that climate solutions represent a growing market under the rise of climate transition risks, then firms with higher climate solutions measures might demonstrate greater revenue growth. Consistent with this prediction, prior literature finds that innovation in emerging technologies is a key driver of firm growth[13,35]. At the same time, it is not clear if firms engaging in climate solutions could exhibit higher revenue growth because there is high

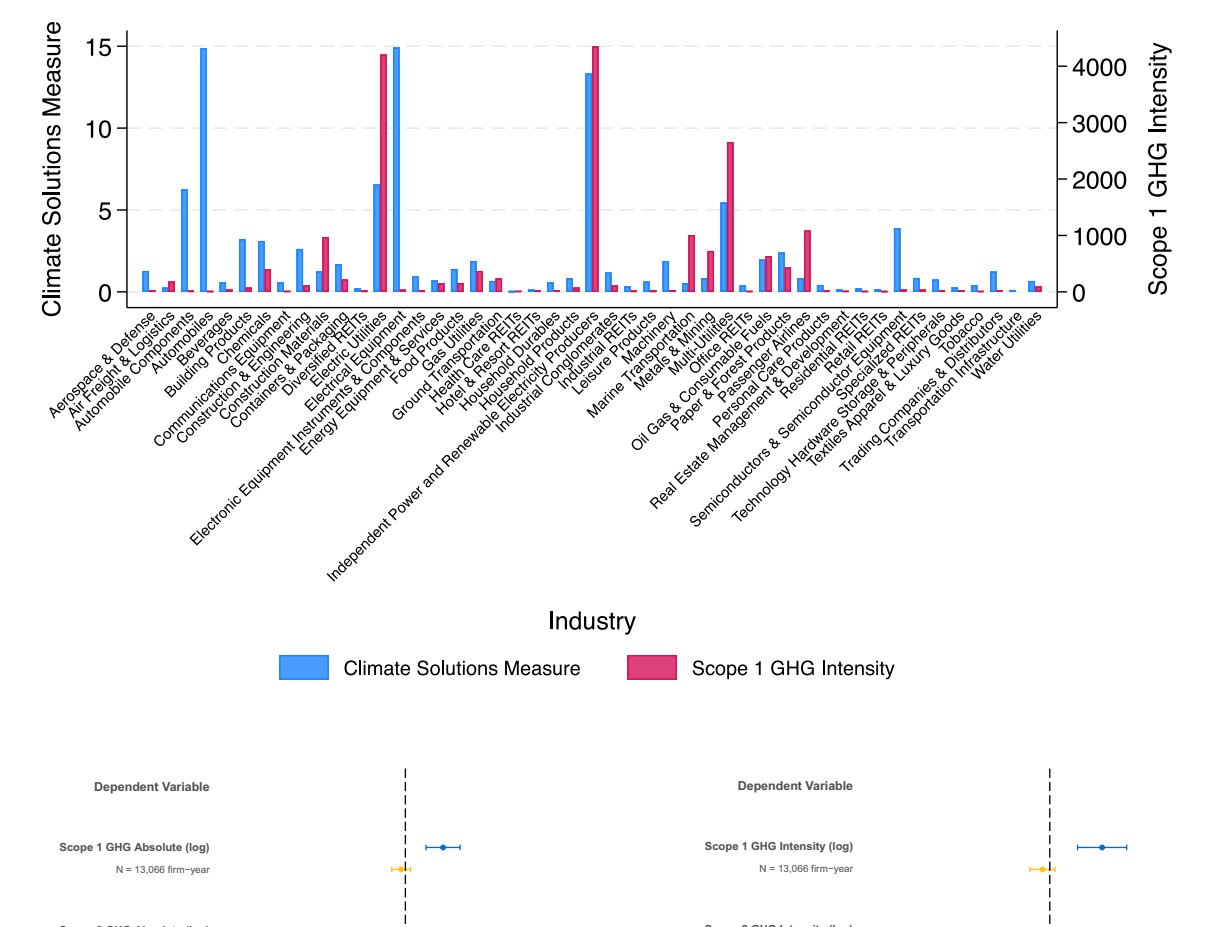

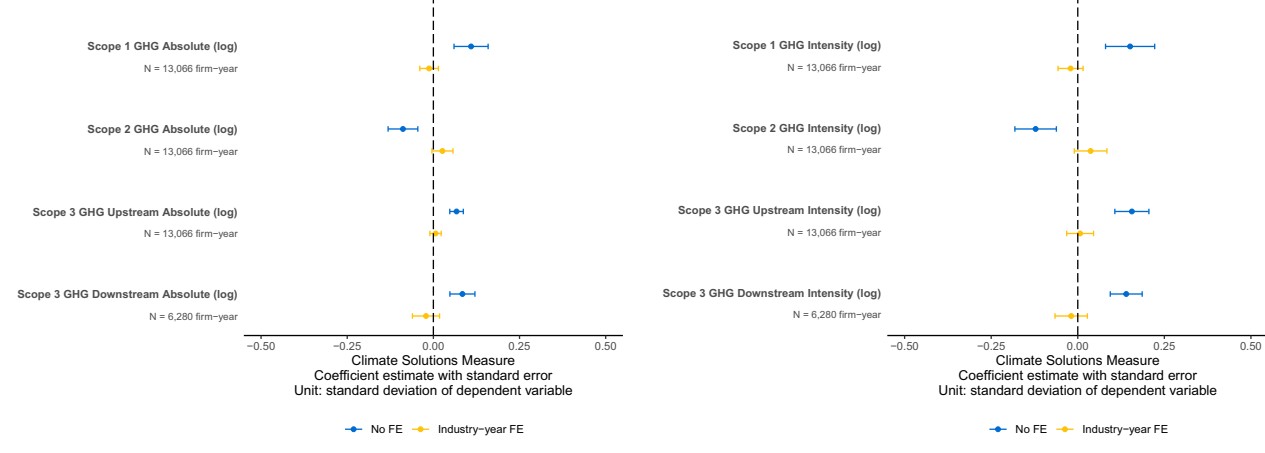

**Fig. 3 | Climate solutions and climate risks. a** Bar chart of average climate solutions measure and scope 1 greenhouse gas by GICS industry. **b** Plot of the coefficient estimate and 95th percent confidence interval of regressing greenhouse gas measures on climate solutions measure. The coefficient reflects the association of one standard deviation change in climate solutions measure on the dependent variables (in standard deviations). Greenhouse gas measures include scopes 1, 2, 3 upstream, and 3 downstream emissions. Each plot includes controls for log revenue in period t-1 and firm age, and shows specifications with and without industry-year fixed effects. The left figure shows results using absolute emissions, and the right figure shows results using emissions intensity scaled by a firm's revenue. See Supplementary Table 5 for regression details and results.

uncertainty regarding the demand for climate solutions, given uncertainties in climate regulations, market preference, and technology developments[36]. As such, we examine if firms with higher climate solutions measure experience higher revenue growth, and also examine how this relationship varies with the prevalence of patent protection across industries and across climate solutions with different characteristics.

Overall, the climate solutions measure is positively and statistically significantly associated with revenue growth. With industry-year fixed effects, a one standard deviation higher climate solutions measure is associated with a 2% higher revenue growth (Fig. 7a). This positive association between climate solutions measure and revenue

growth is consistent with the measure reflecting exposure to and commercialization of real economic activities. This result is robust to using alternative measures of revenue growth, controlling for firm time-invariant characteristics, and using entropy-balanced sample controlling for a firm's other financial characteristics (Supplementary Table 6 Panels C, D).

Next, we examine under what conditions the revenue growth is more pronounced. Motivated by IPCC's observation that the role of intellectual property rights in climate innovation is not well understood, with some viewing it as a barrier and others as an enabler, we examine how the relationship with revenue growth varies in industries with more patent protection[16]. We separate observations based on the

a

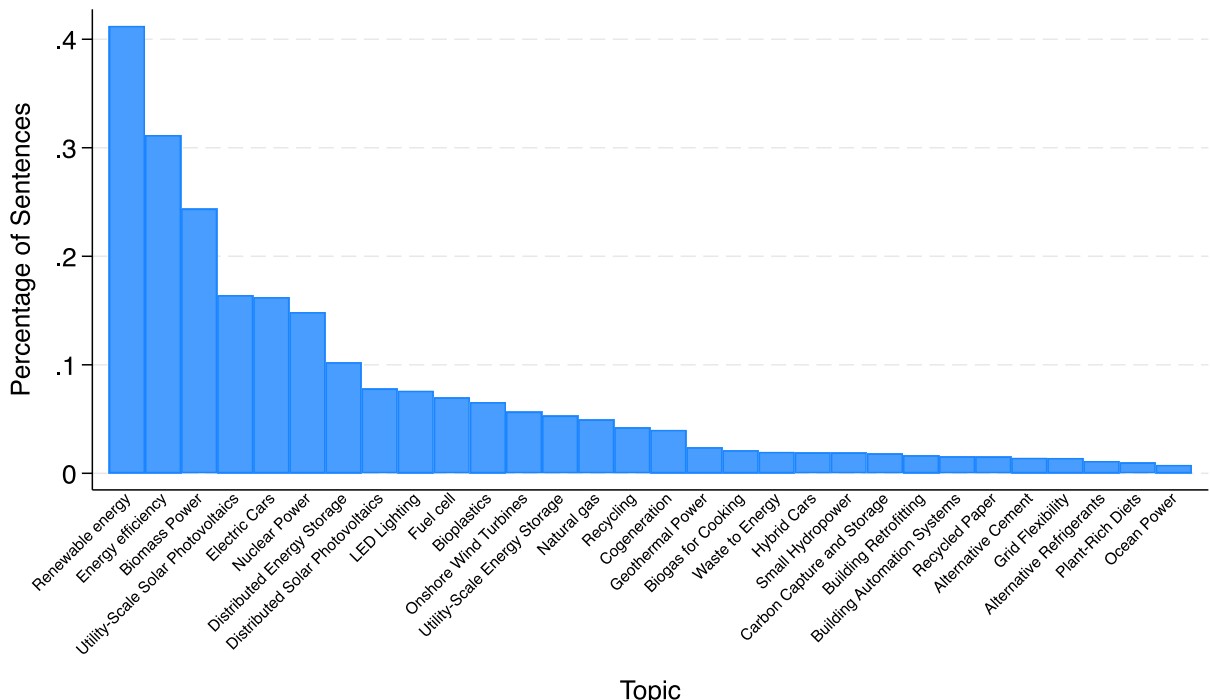

b

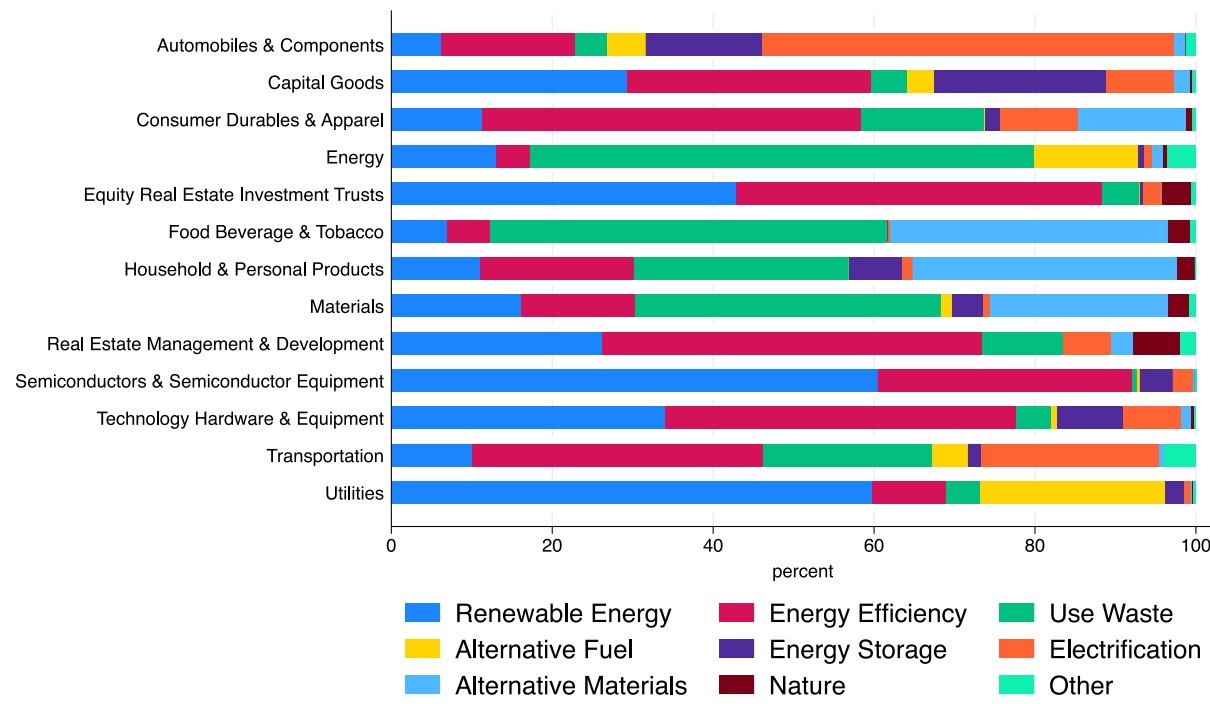

**Fig. 4 | Climate solutions topics. a** This bar chart lists the top 30 topics ranked from the highest to lowest based on the number of sentences containing the topic scaled by the total number of sentences in 10-K Item 1. **b** Average percentage of climate solutions sentences belonging to each topic group by GICS industry groups.

median industry-level patent amount, and find that the estimated coefficient is only significantly positive in high-patent industries (Fig. 7a).

Moreover, using the climate solutions topics, we examine how revenue growth varies between technologies with different abatement potential and costs (Fig. 7b). We find similar estimated coefficients across high- and low-abatement-cost technologies. However, the association is estimated more reliably for low-abatement-cost technologies, consistent with larger variation in the ability to successfully commercialize high-cost technologies

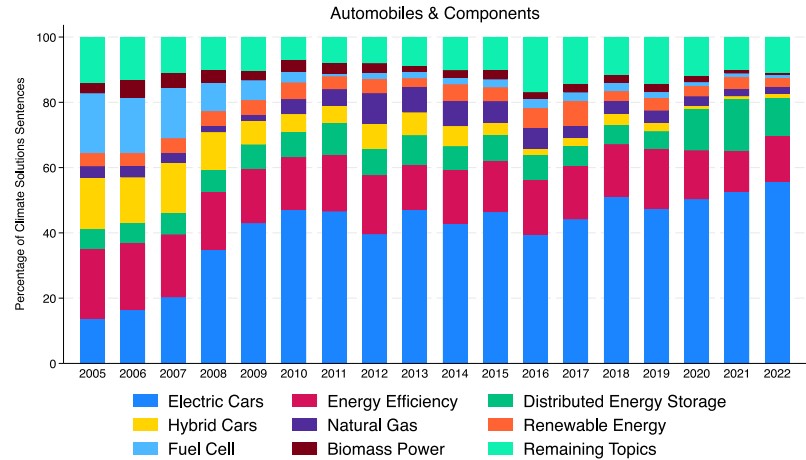

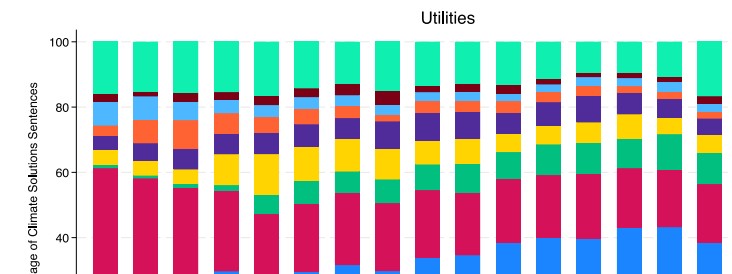

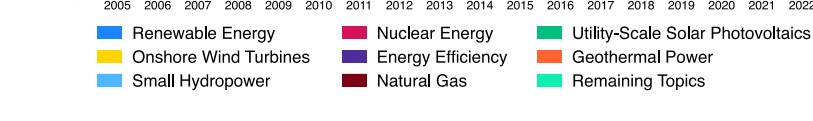

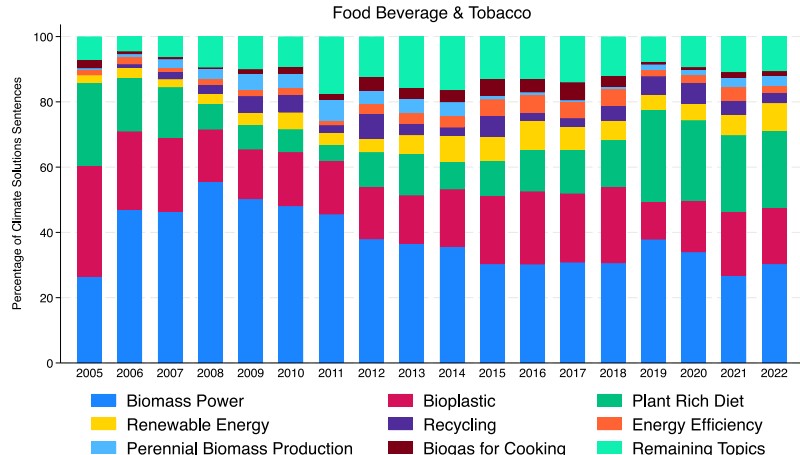

**Fig. 5 | Climate solutions topics in select industries. a** Percentage of climate solutions sentences of the top eight topics in automobile and components for each year starting from the fiscal year 2005. **b** Percentage of climate solutions sentences of the top eight topics in utilities for each year starting fiscal year 2005. **c** Percentage of climate solutions sentences of the top eight topics in food, beverage, and tobacco for each year starting fiscal year 2005.

that could depend more on the existence of public policy support. Consistent with the expectation that technologies with higher abatement potential create greater value by helping other firms reduce emissions—thereby driving higher demand—the estimated association with revenue growth is larger among technologies with higher abatement potential, whereas the relationship is not statistically significant for technologies with lower abatement potential.

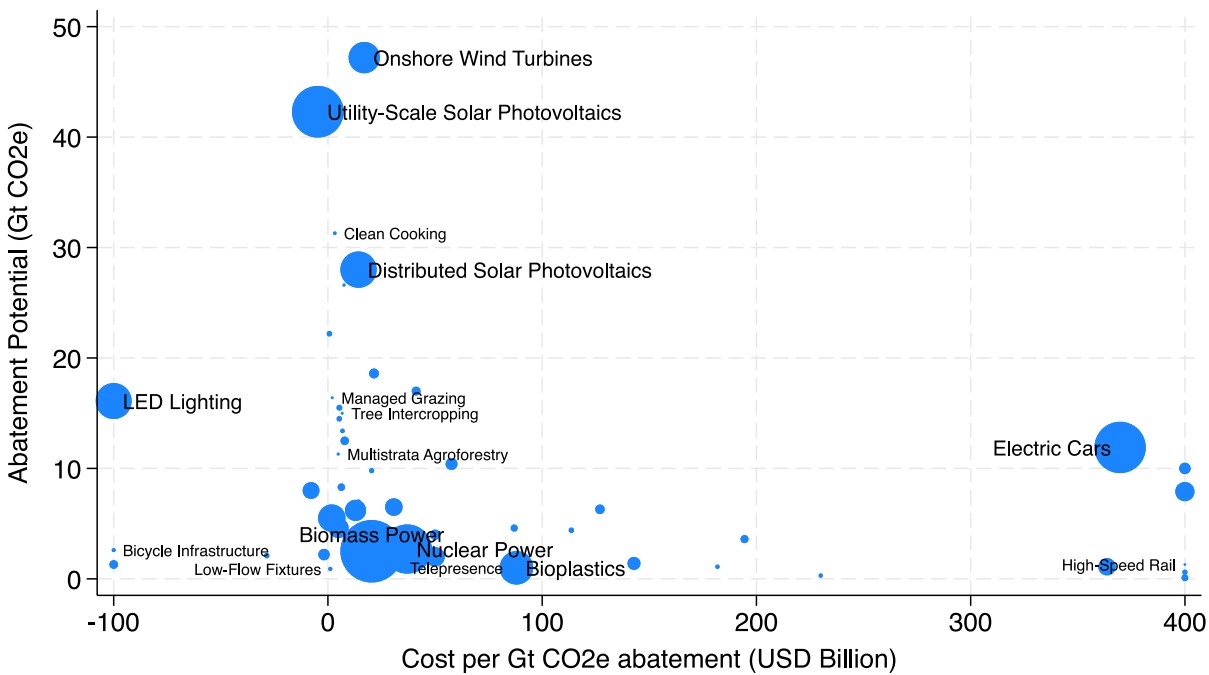

**a** Marker size is proportional to CS measure for the topic. Top (bottom) 8 topics labelled in large (small) fonts.

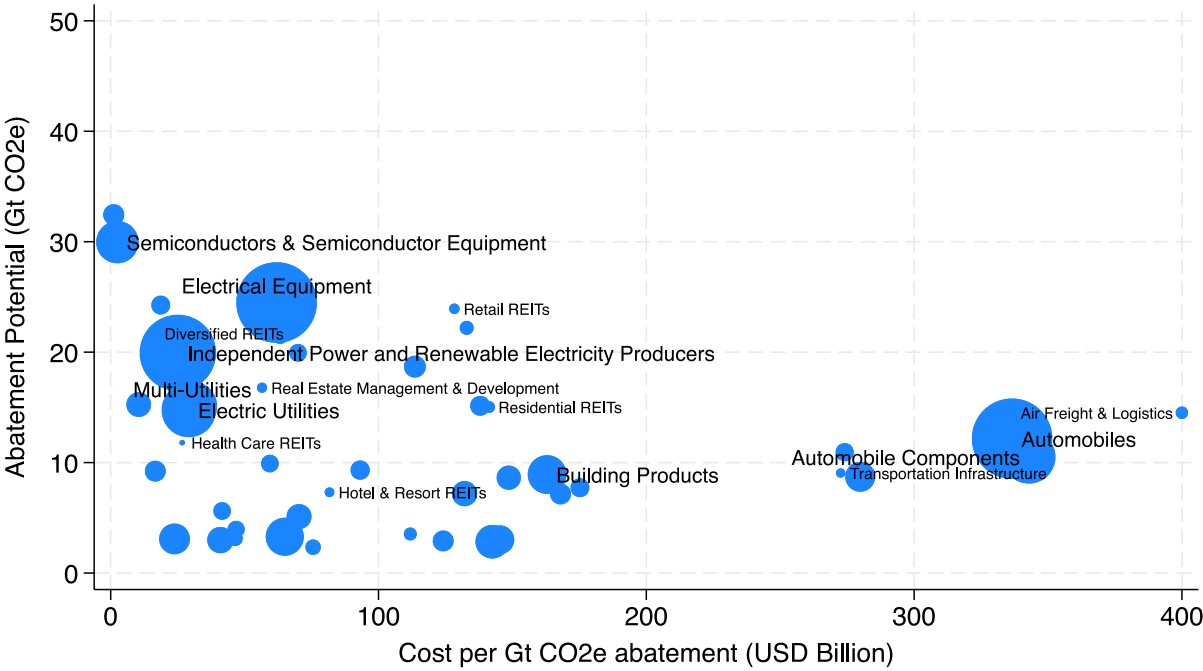

**b** Marker size is proportional to CS measure for the topic. Top (bottom) 8 industries labelled in large (small) fonts.

**Fig. 6 | Carbon abatement costs and abatement potential. a** This chart plots the abatement potential and cost per abatement of climate solutions topics based on estimates from Project Drawdown. Each marker represents a topic, and the size of the marker is proportional to the average number of sentences containing that topic, scaled by the total number of sentences in 10-K Item 1. Only topics with data from Project Drawdown are included. Cost per abatement is winsorized at 400. **b** This chart plots the average abatement potential and cost per abatement for each industry. Each marker represents an industry, and the size of the marker is proportional to the average climate solutions measure. The abatement potential and cost per abatement for each industry are based on the climate solutions topics covered in the industry. Only topics with data from Project Drawdown are included. Cost per abatement is winsorized at 400.

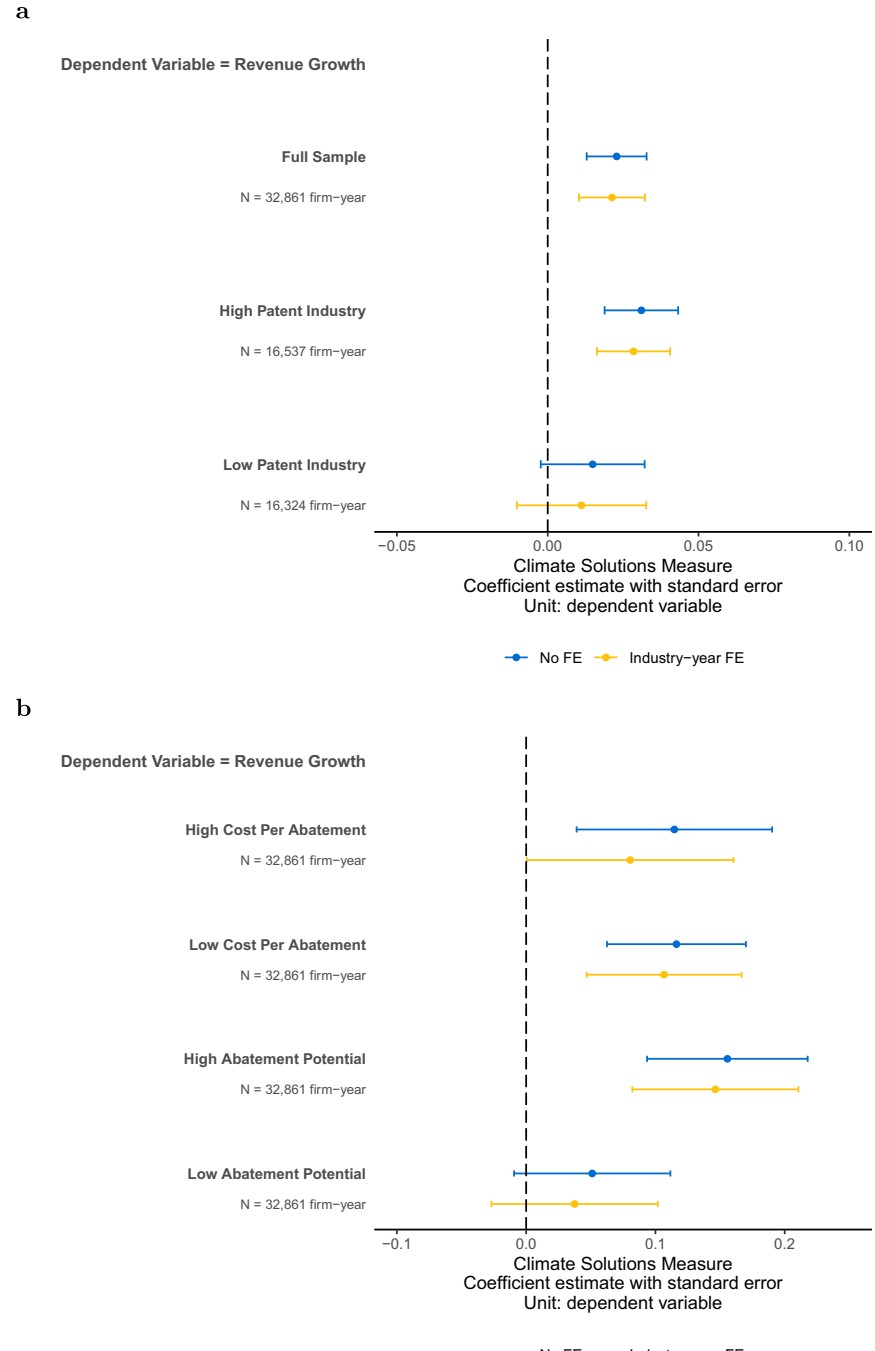

**Fig. 7 | Climate solutions and revenue growth. a** Plot of the coefficient estimate and 95th percent confidence interval of regressing revenue growth on climate solutions measure. **b** Plot of the coefficient estimate and 95th percent confidence interval of regressing revenue growth on topic characteristic. The coefficient reflects the association of one standard deviation change in climate solutions measure on the dependent variables (in standard deviations). Each plot includes controls for log revenue in period t-1, firm age, debt to asset, CAPEX to asset, and ROA. Each plot shows specifications with and without industry-year fixed effects. See Supplementary Table 6a, b for regression details and results.

## Application 2: Climate solutions and political affiliation of firm locations

We examine how firms operating across states with different political affiliations engage in climate solutions. Prior studies find evidence of partisan polarization in that Republicans are less likely to support climate mitigation policies in the United States[37–40]. Other evidence points to political affiliation being less relevant in climate actions[41], such as in explaining solar panel installations[42], household energy efficiency behavior[43], and cities' adaptation efforts[44]. As such, ex-ante,

it is unclear whether political affiliation is associated with the climate solutions measure. To the extent that political affiliation shapes local policies and beliefs, we could observe a political divide in the climate solutions measure. On the other hand, if companies deem climate solutions economically beneficial, such as when the costs of certain technologies are sufficiently low, firms could develop and deploy climate solutions regardless of differences in political affiliations.

We group firms located in states with predominantly Republican or Democratic voting patterns based on their employee distribution

a

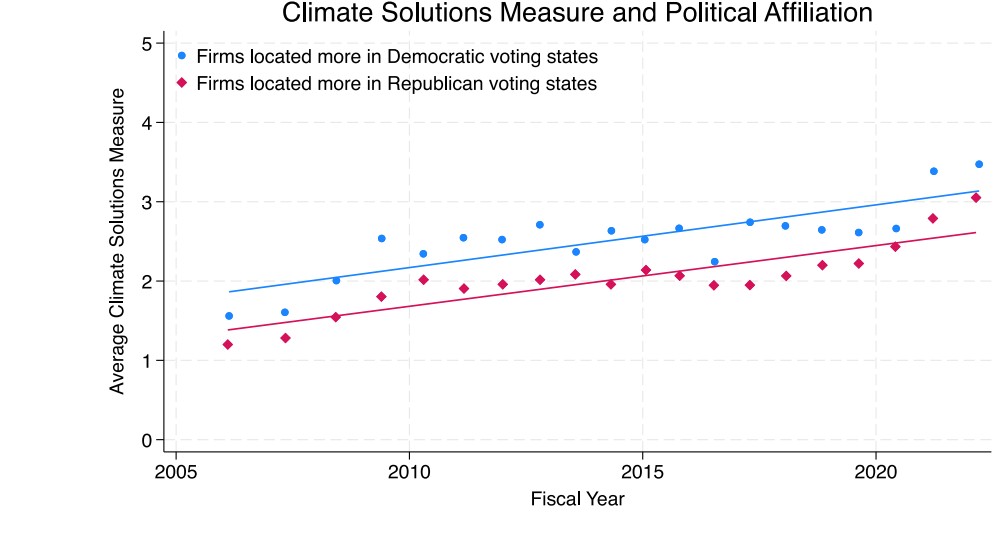

b

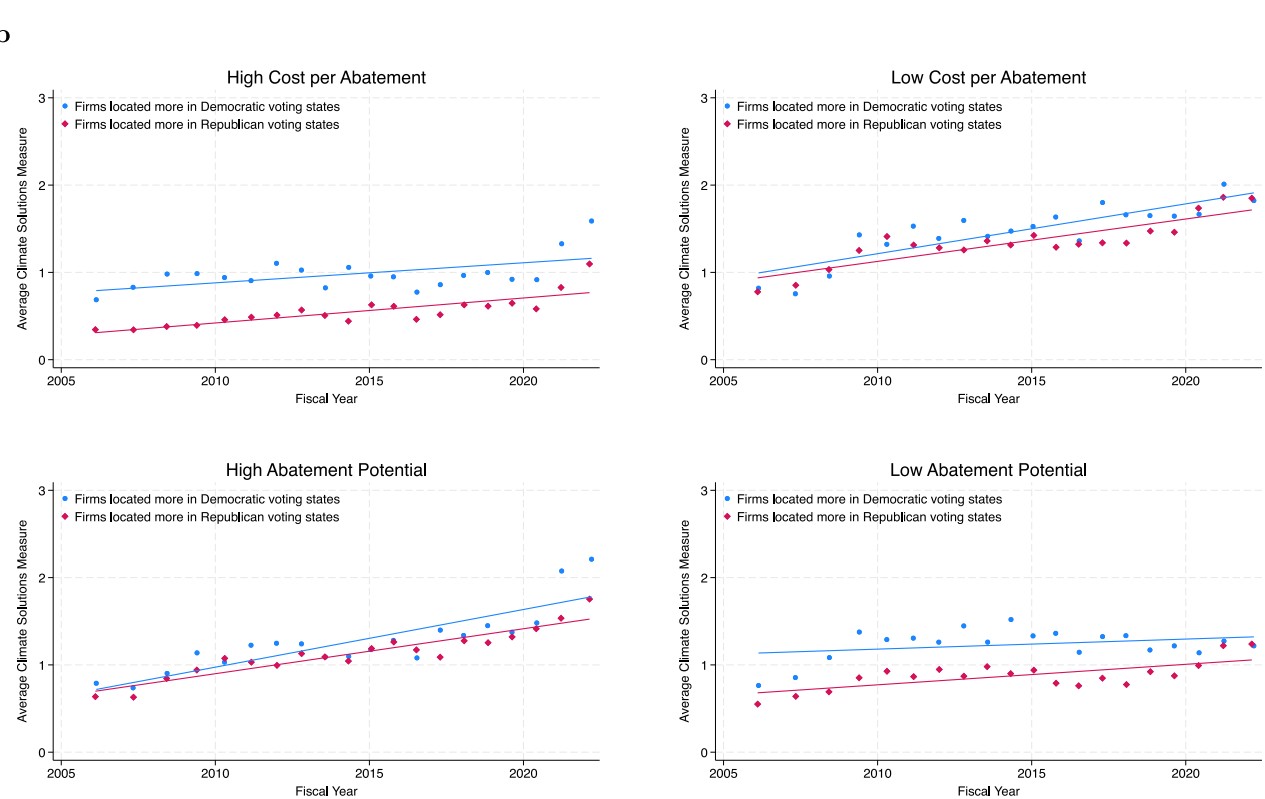

**Fig. 8 | Climate solutions and party affiliation of firm locations. a** Plot of the average climate solutions measure separately by party affiliation over time, including industry fixed effects and controlling for log revenue in period t-1 and firm age. **b** Plot of the average climate solutions measure by topics with high or low cost per abatement and abatement potential separately, by party affiliation over time. Include industry fixed effects and control for log revenue in period *t*-1 and firm age.

in different states, where the political affiliation of each state is based on the 2020 presidential-vote outcomes. State-level presidential-vote share offers a transparent, parsimonious signal of local political orientation, but we acknowledge it is only a proxy for the policy environment relevant to climate action. On average, firms located more in Republican voting states have a lower climate solutions measure than firms located more in Democratic voting states, with average climate solutions measures of 2.1 and 2.5%, respectively. To ensure other firm characteristics are kept constant, we compare within industry and include controls for firm size and age, and find

that this political gap in climate solutions measure persists throughout the sample period (Fig. 8a). Separately examining climate solutions with different topic characteristics reveals variation in the gap (Fig. 8b). In particular, the gap is largest among climate solutions with a high cost per abatement, where on average, firms located more in Republican voting states have a significantly lower measure by 0.14 standard deviations relative to firms located more in Democratic voting states (Supplementary Table 7). In contrast, among technologies with low cost per abatement, the average climate solutions measures of firms located more in Republican and

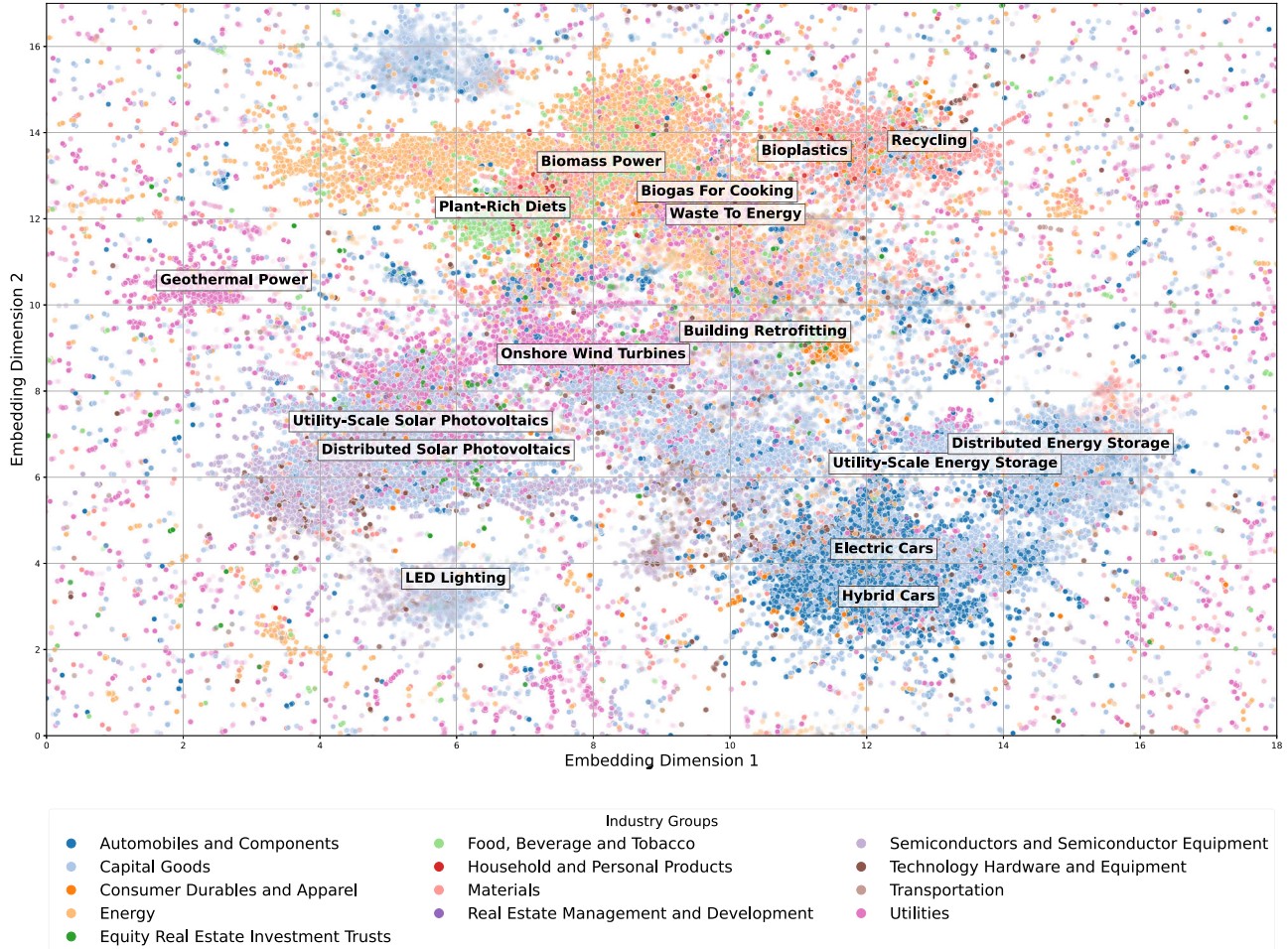

**Fig. 9 | Climate solutions sentence embeddings plot.** Plot after flattening the embeddings of each climate solutions sentence into a two-dimensional vector. Each dot represents a sentence. The colors represent GICS industry groups. We label the 15 most frequent topics that exist in Project Drawdown, and plant-rich diets for illustration purposes, based on where the concentration of such topics is highest in the plot.

Democratic voting states are not statistically different. We observe a similar pattern where the gap is statistically significant for the climate solutions measure with low abatement potential, but not for high abatement potential.

Taken together, the political affiliation analysis highlights descriptive patterns that should be interpreted with caution. Our proxy for political affiliation is based on presidential-vote share, but we acknowledge that climate policy in the United States is often shaped at the state level, and that many firms operate across multiple countries and may be influenced by international policy environments. Additionally, while the observed differences in the climate solutions measure are statistically significant in certain comparisons, they are modest in magnitude. These caveats underscore the need for future research using more granular, multi-level policy data to improve inference.

**Application 3: Climate solutions and industrial convergence**
The rise of climate solutions in company product portfolios raises the likelihood that companies in seemingly unrelated industries could, in the future, compete for the same resources or partner to bring solutions to the market. For example, electric cars and the lithium needed for batteries have made companies in the automobile manufacturing, electrical equipment, oil and gas, and mining industries compete or collaborate in a new value chain for lithium. We examine how the growth of climate solutions transcends traditional industry boundaries and whether this convergence is reflected in stock return

synchronicity, a measure that prior literature uses to capture how closely related firms or industries move together in financial markets[45–47].

To analyze how industries are converging on similar climate solutions, we plot each climate solutions sentence based on their embedding proximity (Fig. 9). Each dot represents a sentence and is color-coded based on the industry group of the firm. Dots on similar climate solutions would cluster in the same area. We observe a blurring of industry boundaries, with previously unrelated industries engaging in similar products because of climate solutions. Specifically, we observe biomass power (e.g., ethanol) in the energy (42%), food, beverage, and tobacco (18%), and materials (18%) industry groups (Fig. 10a). This pattern reflects the use of biofuels, which involve converting crops into fuel, such as ethanol produced by fermenting the sugar content in corn. We also observe a close connection between the automobiles and components, and capital goods industry groups in climate solutions relating to electric vehicle production, where these two industry groups are most represented in the electric cars, hybrid cars, and energy storage topics.

The topic with the least dominance from one industry is building retrofitting, where we observe the interconnectedness of the capital goods (24%), equity real estate investment trusts (23%), and materials (21%) industry groups (Fig. 10a). In contrast, the most concentrated topic is plant-rich diets, where 93% of climate solutions sentences come from the food, beverage, and tobacco industry group. Among renewable energy topic groups, we observe variation where

**a**

| Topics | Rank1 | % | Rank2 | % | Rank3 | % |
|---|---|---|---|---|---|---|
| Plant-Rich Diets | Food Beverage & Tobacco | 93 | Household & Personal Products | 7 | Materials | 0 |
| Electric Cars | Automobiles & Components | 85 | Capital Goods | 7 | Transportation | 2 |
| Hybrid Cars | Automobiles & Components | 84 | Capital Goods | 6 | Semiconductors & Semiconductor Equipment | 3 |
| Geothermal Power | Utilities | 83 | Materials | 5 | Capital Goods | 5 |
| Onshore Wind Turbines | Utilities | 77 | Capital Goods | 16 | Technology Hardware & Equipment | 4 |
| Distributed Solar Photovoltaics | Semiconductors & Semiconductor Equipment | 61 | Capital Goods | 18 | Utilities | 9 |
| LED Lighting | Semiconductors & Semiconductor Equipment | 60 | Capital Goods | 30 | Technology Hardware & Equipment | 5 |
| Distributed Energy Storage | Automobiles & Components | 53 | Capital Goods | 30 | Semiconductors & Semiconductor Equipment | 7 |
| Utility-Scale Solar Photovoltaics | Semiconductors & Semiconductor Equipment | 50 | Utilities | 23 | Capital Goods | 16 |
| Biomass Power | Energy | 42 | Food Beverage & Tobacco | 18 | Materials | 18 |
| Recycling | Materials | 42 | Automobiles & Components | 14 | Capital Goods | 11 |
| Bioplastics | Materials | 39 | Food Beverage & Tobacco | 24 | Household & Personal Products | 18 |
| Utility-Scale Energy Storage | Capital Goods | 37 | Automobiles & Components | 31 | Utilities | 19 |
| Biogas For Cooking | Utilities | 38 | Energy | 15 | Food Beverage & Tobacco | 12 |
| Waste To Energy | Utilities | 28 | Energy | 21 | Capital Goods | 16 |
| Building Retrofitting | Capital Goods | 24 | Equity Real Estate Investment Trusts | 23 | Materials | 21 |

**b**

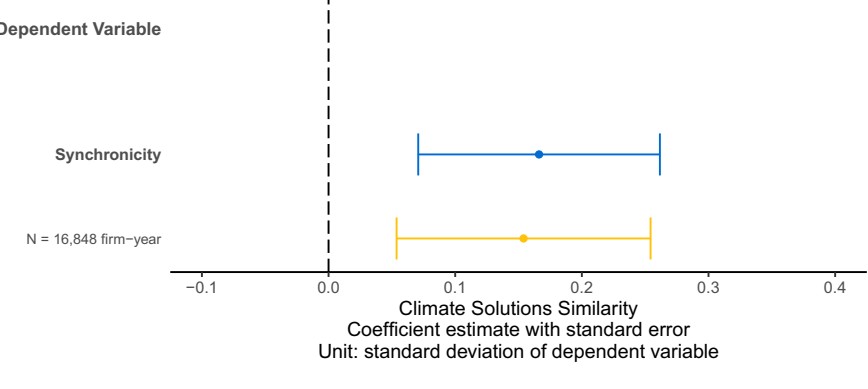

**Fig. 10 | Climate solutions and industry convergence. a** Top three GICS industry groups for each topic labeled in Fig. 9, and their percentage of climate solutions sentences among all climate solutions sentences belonging to this topic. The table is ranked from the highest to the lowest percent of the top-ranked industry group. **b** Plot of the coefficient estimate and 95th percent confidence interval of regressing monthly stock return synchronicity on Climate Solutions Similarity for pairs of GICS industry groups. The coefficient reflects the association of one standard deviation change in climate solutions similarity on the dependent variable (in standard deviations). The plot shows specifications with and without year-month fixed effects. See Supplementary Table 8 for regression details and results.

geothermal power is more concentrated in the utilities industry group, whereas wind and solar are more interconnected with the capital goods industry group, where energy storage is involved.

To examine whether this convergence in climate solution topics across industries is reflected in financial market behavior, we analyze stock return synchronicity between industry group pairs. Stock return synchronicity quantifies the extent to which stock returns across two industries move together, with higher values indicating more similar underlying economic fundamentals[45–47]. Using our climate solutions topics, we compute cosine similarity scores between industry pairs each year based on the distribution of topics firms disclose. We find that industry pairs with higher topic similarity are associated with greater co-movement in their stock returns (Fig. 10b). A one standard deviation increase in climate solution similarity corresponds to a 0.15 standard deviation increase in stock return synchronicity. While the magnitude is modest, this finding suggests that for firms across industries that disclose similar types of climate solutions in financial reports, their economic fundamentals, and thus their stock market performance, are more aligned.

## Discussion

Using AI to analyze companies' business descriptions in regulatory filings across major industries with the potential to provide decarbonization technologies, we document widespread discussion of climate solutions, with 45 percent of firms mentioning products or services in their 10-K business descriptions that relate to climate-solution technologies, indicating potential contributions to future decarbonization efforts. Our results demonstrate that this perspective of climate business opportunity is incremental to that of climate risk. A holistic analysis should integrate both perspectives, equipping users with measures to assess both risk and opportunity. Although much progress has been made in recent years in risk measurement and analysis, less progress has been made in opportunity measurement and analysis. Our motivation has been to contribute to the development of this business opportunity perspective.

Our application of the climate solutions measure to US public firms yields three sets of insights. First, the revenue growth analysis reveals that companies whose filings reference climate-solution products or services are associated with higher revenue growth, consistent with growing demand for climate solutions technologies. However, this finding is not uniform across all industries and technologies but rather depends on specific market conditions and technological attributes. The increase in revenue growth is only statistically significant in industries where innovation is generally protected with patents, and for climate solution topics with higher abatement potential, reflecting higher demand for technologies with the potential to reduce a larger amount of emissions.

Second, we find that firms located more in Republican voting states exhibit lower climate solutions measure than firms located more in Democratic voting states. One potential reason is that political affiliation influences state policies[40], which drives firm incentives to develop or deploy climate solutions. However, this political gap diminishes in areas where climate solutions have lower abatement costs, indicating that as costs for many climate technologies, such as solar panels and wind turbines, continue to decline, firms are more likely to adopt these solutions based on financial viability[22]. These findings complement recent work on the role of multinational firms in addressing climate change, which highlights how firms leverage economic and operational advantages to pursue climate strategies that align with profitability[48].

Third, we observe changes in industry boundaries where seemingly unrelated industries are involved in similar types of climate solutions. As industries converge around shared technologies—such as electric vehicles and lithium batteries, biofuels, and renewable energy storage—firms that once operated independently are now competing for the same resources or forming new partnerships to bring solutions to market. Consistent with industrial convergence having economic implications, industries that focus on the same climate topics exhibit higher stock return synchronicity. These findings suggest that as climate solutions evolve, industries could continue to overlap in unexpected ways, transforming traditional value chains and creating new competitive and collaborative dynamics in the transition to a low-carbon economy. Future research could further explore the financial and innovation implications of such industry convergence.

The use of the text in regulated accounting reports enables the measurement of climate solutions and opens exciting avenues for various stakeholders. For researchers, the data mitigate the measurement challenge and enable researchers to conduct systematic analysis on the drivers and implications of a firm's engagement in climate solutions[49,50]. Relative to other measures on climate opportunities derived from voluntary disclosures, such as earnings conference calls and CDP reports[8,51,52], our measure differs by focusing on products and services that are disclosed in regulated financial filings, with greater credibility, lower noise, and less subject to concerns about the self-selection of disclosures. Furthermore, our measure is more likely to capture climate solutions already integrated into a firm's existing product portfolio, rather than input-based proxies like R&D expenses or labor skills, which reflect investment efforts rather than commercialized solutions. Consistent with this expectation, our climate solutions measure is positively associated with revenue growth, although the evidence does not necessarily imply causality. In contrast, climate exposure and opportunity measures derived from earnings conference calls are more reflective of market attention and not associated with revenue growth when we include them in the same regression with climate solutions measure[8] (Supplementary Table 6 Panel E).

Likewise, investors can leverage the climate solutions measure to identify promising investments, facilitating funds directed to advance climate solutions. Regulators can benefit from using the climate solutions measure to inform policy decisions. Specifically, it can serve as a monitoring tool of the evolution of product portfolios of individual companies, industries, and whole sectors of the economy toward technologies or solutions. This could inform whether the stated goals of policies are achieved and inform analysis of important contingencies that hinder or could accelerate the evolution of product portfolios. Although rapid technological change is widely acknowledged to be the key to the transition to a low-carbon economy[53], the specific technological requirements for this transition have not been rigorously evaluated under existing regulations, such as the Clean Air Act[54]. Using the climate solutions measure can help in the design of policies that strategically allocate capital toward the crucial technologies necessary to meet climate goals and drive environmental progress.

Our study has limitations. First, our analysis is limited to publicly listed US firms and a subset of industries that are most likely to engage in climate solutions. Although this setting provides the advantage of a standardized disclosure format with legal implications, it does not cover the entire economy. Future research could extend our model to different settings, including non-US firms and non-exchange-listed entities. Second, we rely on Project Drawdown and the GPT model to identify climate solution sentences, which may not reflect the comprehensive set of climate solutions. Although we attempt to mitigate this concern by choosing a training set from a variety of representative industries, there could be new solutions that are not captured in our model. Despite this concern, GPT should be able to pick up phrases such as "We sell technology A to help firms reduce carbon," so that even if technology A was not in our training set, GPT can identify the technology as a climate solution based on the context of the sentence. Third, our measurement of climate solutions relies on information disclosed in 10-K filings, which may still be subject to greenwashing concerns. However, compared to

voluntary disclosures, 10-K filings are likely more reliable due to SEC oversight, and firms have faced enforcement actions for misrepresenting climate-related information in these filings[55]. In the Methods section, we provide details on other robustness and sensitivity tests we conduct to gain confidence in our measure. The descriptive results, such as time trends, industry trends, and financial outcomes, also further provide validation for the climate solutions measure. However, we caution the reader that in the absence of a natural experiment with randomized treatment of climate solution products and services, we are unable to establish causality.

# Methods

## Climate solutions GPT model

To measure a firm's engagement in climate solutions, we fine-tune a GPT model to detect climate solutions sentences in the Item 1 Business Description section of 10-K filings from the SEC. We extract textual data for the universe of US public firms that report SEC 10-K filings in the EDGAR database from fiscal year 2005 to 2022. We focus on industries that are pivotal to climate solutions, where our LLM is likely more accurate in identifying climate solutions. Based on reviewing Project Drawdown, we keep 13 (out of 25) GICS industry groups that are central to climate solutions: energy, materials, capital goods, transportation, automobiles & components, consumer durables & apparel, food beverage & tobacco, household & personal products, technology hardware & equipment, semiconductors & semiconductor equipment, utilities, equity real estate investment trusts (REITs), real estate management & development.

We define climate solutions as products and services that develop or deploy new technologies in a transition to a low-carbon economy. We identify climate solution technologies based on guidance from Project Drawdown, which contains a list of technologies that can reduce greenhouse gases in the atmosphere, and are compiled by a network of scientists and researchers[56].

To fine-tune our climate solutions GPT model, we label 3508 sentences into climate solution sentences or not as our training set. These sentences are chosen from 10-K Item 1 sentences that are representative of each of the 13 industry groups, as well as sentences that the model deems more difficult to classify through an active learning approach. We also leverage the fine-tuned GPT model to classify each climate solutions sentence into one of 88 climate solutions topics. We provide a detailed description of the procedure involved in creating the Climate Solutions LLM and the topic model in Supplementary Note 1. We provide details of the labeling process to create our training set in Supplementary Note 2.

Our primary climate solutions measure represents the proportion of sentences identified as climate solutions to the total number of sentences in the 10-K Item 1 Business Description, multiplied by 100 to express the measure as a percentage. Variable definitions and summary statistics of the climate solutions measure and other variables used in this study are presented in Supplementary Tables 1, 2, respectively. This measure assumes that the relative proportion of climate solutions sentences reflects a firm's product or service focus on climate solutions. Our results remain similar when we consider three sets of alternative ways to measure a firm's climate solutions. The first set is CS measure top 50 and CS measure top 100, where we only keep the percent of climate solutions sentences in the top 50 and 100 sentences in 10-K Item 1, respectively. The second set is CS measure weighted 50 and CS measure weighted 100, where we place a larger weight on earlier sentences in 50 and 100-sentence increments, respectively. The third set is CS measure rolling 50 and CS measure rolling 100, which are the percent of climate solutions sentences in the 50- and 100-sentence-segments with the highest ratio of climate solutions sentences, respectively. We present the correlation matrix of these six alternative measures and our primary climate solutions measure in Supplementary

Table 3. Our primary climate solutions measure is highly correlated with each of the six alternative measures at over 90%, providing comfort that our measures are not sensitive to the choice of construction method.

## Validation analysis

We conduct three sets of validation analyses to demonstrate that the climate solutions measure reflects companies developing or deploying products or services that help reduce emissions.

In the first two validation analyses in Fig. 2, we conduct the following ordinary least squares (OLS) regression to compare climate solutions measure to MSCI green revenue, green patents data, and innovation measures for firm $i$:

$$\text{Validation measures}_{i,t} = \beta_0 + \beta_1 \text{Climate Solution Measure}_{i,t} + \sum \text{Controls} + \sum \text{Fixed Effects} + \epsilon \quad (1)$$

In the first validation analysis, the dependent variables are the percent of revenue and patent value that is related to low-carbon products, as classified by MSCI. These two MSCI variables are only available for the latest year using fiscal 2022, resulting in a lower sample size. To directly compare how changes in climate solutions measure relate to changes in MSCI green revenue and green patent percentage, we do not include control variables, but the results remain robust to including control variables for firm size and age. For all regressions, we include one specification without fixed effects and one specification with industry-year fixed effects. Industry-year fixed effects are GICS industry indicators interacted with each year to control for time-varying variation of the outcome variable in each industry. This research design compares the outcome for firms with higher climate solutions measure relative to those with lower climate solutions measure, within each industry-year. Standard errors are clustered at the firm level to address potential correlations across different years within a firm. The results are presented in Supplementary Table 4 Panel A.

In the second validation analysis, we employ four complementary measures of firm-level innovation. R&D expenditures is the research and development expenses (xrd) as disclosed in financial statements, retrieved from Compustat. Following prior literature, we treat missing R&D expenditures as zero and include a dummy variable for firms with missing R&D data[57]. To mitigate concerns about the prevalence of missing data, we also report results limited to industries where the average rate of missing R&D data is below the median, and we show results using specifications where missing values are not replaced with zero. Knowledge capital quantifies the stock of accumulated R&D investment by capitalizing R&D expenditures using industry-specific depreciation rates, reflecting firms' long-term innovation assets[58]. RDC, Research & Development Capitalization, similarly captures the capitalized value of R&D but is estimated based on its expected contribution to future revenues, providing an alternative perspective on the persistence of innovation efforts[59]. Trade secret is the number of trade secrets explicitly referenced in its 10-K filings, serving as a proxy for proprietary knowledge protection strategies[60]. We scale all innovation measures with firm size using revenue retrieved from Compustat, and winsorize the variables at the 1st and 99th percentiles. Together, these measures provide a comprehensive assessment of firms' innovation strategies, spanning both tangible R&D investments and intellectual property protection mechanisms. We control for firm size and age. We use the natural logarithm of revenue in year t-1 to control for firm size. We also control for firm age, measuring the number of years since the firm's inclusion in Compustat, to compare firms at a similar point in their lifecycle. Standard errors are clustered at the firm level. The results are presented in Supplementary Table 4 Panels B, C.

In the third validation test, we use difference-in-differences models to examine how climate solutions measure changes after two major climate policy interventions for firm $i$ in year $t$:

$$Relevant\ Climate\ Solutions\ Measure_{i,t} = \beta_0 + \beta_1 Post_t \\ \times Relevant\ Industries_i \\ + \sum Fixed\ effects + \epsilon \quad (2)$$

For the IRA event, the dependent variable is the IRA climate solutions measure, which is the percent of sentences from 10-K Item 1 containing climate solutions topics that is covered by the IRA. We manually identify topics covered by the IRA, referencing the Guidebook on the Inflation Reduction Act released by the White House[61]. The variable of interest is the interaction term of Post-IRA × High IRA Industries. Post-IRA is an indicator for fiscal years 2021 and 2022, where financial reports are released in 2022 and 2023, after the announcement of IRA in 2022. To identify industries relevant to IRA, we calculate the average IRA climate solutions measure by industry using observations before fiscal year 2021, and classify industries above the median as High IRA Industries. The coefficient on the interaction term represents the change in the IRA climate solutions measure for high IRA Industries relative to low IRA Industries, and in fiscal years 2021 and 2022, relative to earlier years. We include firm fixed effects and year fixed effects, which control for firm-level time-invariant characteristics and annual time trends, respectively. Standard errors are clustered at the firm level.

We conduct a similar analysis with the RPS events, with modifications as RPS is implemented in a staggered time frame and varies by states. The dependent variable is the renewable energy topic, which is the percent of sentences from 10-K Item 1 containing climate solutions topics under the renewable energy topic group. The variable of interest is the interaction term of post-RPS (weighted) × high renewable industries. Post-RPS (weighted) is a weighted average of indicators for states after they pass renewable portfolio standards, where the weight is based on a firm's distribution of employees in each state, obtained from InfoGroup. The coefficient on the interaction term represents the change in the renewable energy topic for high renewable industries relative to low renewable industries, as post-RPS (Weighted) increases by one unit. We include firm fixed effects and year fixed effects, and standard errors are clustered at the firm level. The results are presented in Supplementary Table 4 Panel D.

## Climate solutions and greenhouse gas emissions

In Fig. 3, we conduct the following OLS regression to compare the climate solutions measure to greenhouse gas emissions for firm $i$ in year $t$:

$$Greenhouse\ Gas\ Emissions_{i,t} = \beta_0 + \beta_1 Climate\ Solutions\ Measure_{i,t} \\ + \sum Controls + \sum Fixed\ effects + \epsilon \quad (3)$$

The dependent variables are greenhouse gas emissions measures. We retrieve greenhouse gas emissions data from TruCost. TruCost provides firm-year-level data on absolute and intensity scopes 1, 2, 3 upstream, and 3 downstream emissions based on both firm disclosure and estimates based on a firm's industry activities. As the greenhouse gas measures are skewed to the right, we take the logarithm to better resemble a normal distribution. As with prior specifications, we control for firm size and age, include specifications with and without industry-year fixed effects, and cluster standard errors at the firm level. The results are presented in Supplementary Table 5 Panels A, B. To address potential concerns about the quality

of corporate greenhouse gas emissions data, we repeat the analysis using a subsample of firms that likely report more reliable data. Specifically, we identify firms with a sustainability committee on their board, as this reflects stronger governance over environmental reporting. The results remain similar, as shown in Supplementary Table 5 Panels C, D. In Supplementary Table 5 Panel E, as additional measures related to climate risks, we use emissions scores from Refinitiv and MSCI to capture a firm's climate risk management. For ease of comparison, we standardize the emissions scores to have a mean of 0 and a standard deviation of 1.

## Carbon abatement potential and costs

Project Drawdown provides estimates for each climate solution's abatement potential and costs to achieve the carbon abatement[56]. These estimates are based on analytical models backed by extensive literature and data to estimate the relevant impact based on two scenarios for the period of 2020 to 2050. The first scenario is more conservative and estimates a two-degree temperature increase by 2100. The second scenario is more ambitious and estimates a 1.5-degree temperature increase by 2100. We use the estimates from the first scenario to stay on the conservative side.

For each solution, abatement potential is the $CO_2$ equivalent reduction brought by the technology between 2020 and 2050. In terms of costs, Project Drawdown provides the first cost and the lifetime cost. We focus on the first cost, as data on lifetime cost is less available. The first cost reflects the upfront capital expenditure required to implement the climate solution, and is based on the relative cost to implement these climate solutions between 2020 and 2050 compared to a baseline scenario. The baseline scenario is defined as a scenario where such a climate solution does not exist. For example, the baseline for the climate solution, onshore wind turbines, is based on electricity generated using fossil fuel power plants. As such, negative costs can happen if the climate solution results in a lower cost relative to the baseline scenario (such as the use of LED lighting). To ensure that costs are comparable across different climate solutions, we focus on the first cost divided by the abatement potential. This allows us to compare different climate solution's cost per unit of carbon reduction. Figure 6a plots each topic, where the x-axis shows the average cost per abatement (USD billion per Gt CO2e abatement) and the y-axis shows the abatement potential (Gt CO2e) of the climate solutions technology. Figure 6b plots a similar chart at the industry level. To create this, we first calculate the average climate solutions measure for each topic within an industry. Then, we compute the industry's weighted average cost per abatement and abatement potential, using each topic's relative share within the industry as weights.

## Application 1: Revenue growth

To examine the revenue growth associated with firms with higher climate solutions measures in Fig. 7, we conduct OLS regression models to estimate the following specification for firm $i$ in year $t$:

$$Revenue\ Growth_{i,t} = \beta_0 + \beta_1 Climate\ Solutions\ Measure_{i,t} \\ + \sum Controls + \sum Fixed\ effects + \epsilon \quad (4)$$

The dependent variable is the year-over-year revenue growth, calculated using annual revenue data (revt) from Compustat. We include specifications with and without industry-year fixed effects, and cluster standard errors at the firm level. To mitigate the concern that confounding variables that are correlated with climate solutions measure explain the relationship with revenue growth, we control for the following firm characteristics: firm size, age, leverage, capital

intensity, and profitability. We winsorize financial variables at the 1st and 99th percentile.

We conduct cross-sectional analysis separating firms in industries with high or low patent protection. For each industry, we create a weighted average patent count using firm-level patent counts in MSCI data for fiscal 2022, and weight each observation based on the firm's revenue. In other words, this variable is equal to the sum of a firm's patent counts multiplied by the revenues of a firm over the revenues of the whole industry, across all firms in that industry. We then repeat the baseline revenue growth regression separately for subsamples of firms in industries with low and high patents based on the median value. The baseline and cross-sectional results are presented in Supplementary Table 6 Panel A.

We then repeat the revenue growth analysis, separating topics based on the two climate solutions characteristics: cost per abatement and abatement potential. We first group climate solutions topics into high versus low groups for each of the two characteristics based on the median. We then create scaled measures for each of the four groups of topics, calculated as the percent of sentences from 10-K Item 1 containing climate solutions topics that belong to that group. To enhance comparability across these topic groups, we standardize the measures to have a mean of 0 and a standard deviation of 1. For example, High Cost per Abatement is defined as the percent of sentences from 10-K Item 1 containing climate solutions topics that belong to the high cost per abatement group, standardized. The results are presented in Supplementary Table 6 Panel B.

We conduct four robustness tests to mitigate potential concerns related to the revenue growth results. First, to mitigate concerns about reverse causality, we replace revenue growth with the 1-year forward revenue growth, and 3-year moving average revenue growth from $t = 0$ to $t = 2$, and our results remain robust. Second, we include firm fixed effects to control for time-invariant firm characteristics, and the results remain robust when we use the three-year moving average revenue growth. We focus on the three-year moving average because revenue growth tends to be slow-moving, and using firm fixed effects with one-year revenue growth may absorb too much of the variation, limiting our ability to detect meaningful effects. Third, to ensure comparability between firms with and without climate solutions, we apply entropy balancing on the first three moments of the control variables[62]. As entropy balancing requires discrete treatment variables, we classify firms as having climate solutions if the climate solutions measure is above 1 or 5%, which also helps mitigate concern about the right skew of the main climate solutions measure. Our results remain robust when we repeat the main specifications with these alternative measures for climate solutions, and when using entropy-balanced samples. These robustness tests are presented in Supplementary Table 6 Panels C, D.

Finally, we acknowledge that the $R$-squared is relatively low in this analysis, which likely reflects the wide range of factors that influence revenue growth. We benchmark our $R$-squared against prior studies and find that our explanatory power falls within a reasonable range of the literature ($R$-squared of around 5–15%)[63,64]. To further assess potential concerns related to confounding variables, we follow Oster (2019) to examine coefficient stability in the presence of unobservable factors[65]. Specifically, the method compares the relative movements of coefficients and $R$-squared values from regressions with and without control variables to estimate $\delta$, which captures the proportion of unobservable factors relative to observable factors that will produce a treatment effect of zero. Using an $R$-squared$_{max}$ set to 1.3 times the $R$-squared in Table 6, Panel A, Column 2, as recommended by Oster (2019), our estimated $\delta$ is 5. A $\delta$ of 5 suggests that unobservable factors would need to be five times as influential as observable ones to explain away the estimated effect, providing some comfort over confounding concerns.

## Application 2: Political affiliation of firm locations

To capture the political environment of a firm's operations, we develop a weighted measure based on state-level voting patterns weighted by the geographic distribution of its employees. For each state, we measure Republican vote share as the percentage of votes for the Republican candidate in the 2020 presidential election. We also repeat the analysis using the 2016 presidential election outcomes and find similar results. Employee distribution by state is obtained from InfoGroup, and we compute the weighted average of state-level Republican vote shares, with weights given by the proportion of the firm's employees in each state. We define a firm as being located more in Republican voting states (firms in Republican states) if the weighted measure is above 50%.

To examine how the climate solutions measure differs by the political affiliation of a firm's operation, we conduct OLS regression models to estimate the following specification for firm $i$ in year $t$:

$$Climate\ Solutions\ Measure_{i,t} = \beta_0 + \beta_1 Firms\ in\ Republican\ States_{i,t}$$
$$+ \sum Controls + \sum Fixed\ effects + \epsilon \tag{5}$$

The dependent variable is the climate solutions measure, and we also show results separately for climate solutions topics in the high or low groups in two characteristics, cost per abatement and abatement potential, as defined above. As with prior specifications, we control for firm size and age, include specifications with and without industry-year fixed effects, and cluster standard errors at the firm level. The results are presented in Supplementary Table 7.

## Application 3: Climate solutions topics and industries

For each climate solutions sentence, we plot the two-dimensional embeddings in Fig. 9. To do so, we first generate high-dimensional sentence embeddings using a transformer-based language model from the SentenceTransformers library, trained on large English text corpora to capture semantic meaning. These embeddings are then projected into two dimensions with uniform manifold approximation and projection (UMAP).

In the figure, the reduced two-dimensional embeddings are clustered using hierarchical density-based spatial clustering of applications with noise (HDBSCAN) to identify distinct topic clusters. We identify the 15 climate solutions topics from Project Drawdown with the highest frequency, and then use HDBSCAN to annotate each solution in the cluster where it is more present. Note that for better visualization, we exclude the labels for nuclear power and hydropower in the figure, as these topics are located far from the main concentration of sentences. We also include the topic, plant-rich diets, to illustrate an example of a climate solution more isolated in one industry.

To examine stock return synchronicity for industry group pairs that are becoming more similar in climate solution topics, we estimate the following OLS specification for each pair of industry groups $j$ and $k$ in year-month $t$:

$$Synchronicity_{j,k,t} = \beta_0 + \beta_1 Climate\ Solutions\ Similarity_{j,k,t}$$
$$+ \sum Fixed\ effects + \epsilon \tag{6}$$

The dependent variable, Synchronicity, is the stock return synchronicity between two industries in a given month, constructed following prior literature[46,66]. Specifically, we compute value-weighted monthly stock returns at the four-digit GICS industry group level, then estimate a rolling 36-month time-series regression of the focal industry's returns on the connected industry's returns. The adjusted R-squared from this regression is transformed using the log odds ratio to create an unbounded continuous variable out of a variable originally

bounded by 0 and 1:

$$Synchronicity_{j,k,t} = \ln\left(\frac{R^2}{1-R^2}\right)$$

This measure is computed for each unique industry pair and month, with higher values reflecting stronger co-movement in returns, suggesting greater alignment in economic fundamentals.

The independent variable, climate solutions similarity, is measured as the cosine similarity between the climate solution topic vectors of each industry pair in each year. Specifically, for each industry-year, we first compute the average share of 10-K Item 1 sentences that contain each of the 88 topics across the firms. We then calculate the cosine similarity between these vectors containing the 88 topics for each pair of industries in the same year. A higher value indicates that the two industries disclose more similar climate solution topics in their 10-K filings. We include one specification without fixed effects, and one with year-month fixed effects to account for common shocks or macroeconomic events that may influence all industries' stock return co-movements in a given month. We cluster standard errors at the industry group pair level. The results are presented in Supplementary Table 8.

### Reporting summary
Further information on research design is available in the Nature Portfolio Reporting Summary linked to this article.

### Data availability
The climate solutions data generated in this study are available from GitHub (https://github.com/Climate-Solutions-Project/climate-solutions-10k)[67]. The remaining data are from third-party sources, which we describe in detail in the Methods section.

### Code availability
The codes associated with this study are available from Zenodo (https://doi.org/10.5281/zenodo.16934262)[68].

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

## Acknowledgements

We greatly appreciate the comments of Omar Asensio, Gunther Glenk, Edward Riedl, Mike Toffel, Peter Tufano, and participants at the D^3 Climate Lab Seminar. We thank D^3 lab data scientists Anna Bialas, Sanjana Sharma, George Price, and Samantha Richards for contributions to the Large Language Model fine-tuning and validation. Research assistance from Karan Daryanani, Kelly Fitzpatrick, Niccolo Jacimovi, and Ling Lin are greatly appreciated. We gratefully acknowledge financial support from the Division of Faculty Research and Development of the Harvard Business School.

## Author contributions

S.L., G.S., and S.X. designed and performed the main analysis. S.L., G.S., and S.X. wrote and revised the manuscript. M.A. co-designed and supervised the data science methodology, including the generative AI approach and implementation, and edited the LLM methodology section of the manuscript.

## Competing interests

The authors declare the following competing interests: G.S. has served as an investor, director, and advisor to companies deploying climate solutions technologies. The remaining authors declare no competing interests.
