## [Transparent Peer Review file · Nature Communications]

Tracking Business Opportunities for Climate Solutions using AI in Regulated Accounting Reports

Corresponding Author: Professor Shirley Lu

Version 0:

Reviewer comments:

Reviewer #1

(Remarks to the Author)

Many thanks for submitting this interesting manuscript to Nature Communications. The topic is engaging, as tracking business opportunities for climate solutions using AI in regulated accounting reports is timely and relevant. However, the results are predictable, and the overall contribution – beyond using this new method – remains unclear. I also struggle to see a justification for the manuscript's suitability for an interdisciplinary journal like Nature Communications with the aim to publish work from "all areas of the biological, health, physical, chemical, Earth, social, mathematical, applied, and engineering sciences", rather than a more specialized journal in finance, management, or sustainability.

Clarity of writing / way of arguing: The manuscript is well written and accessible, but statistical results must be more clearly explained and interpreted, with illustrative examples (e.g., Line 115) to reach an interdisciplinary audience.

Contribution: Unfortunately, the contribution to the literature is not clear to me. While developing a climate solutions measure using AI is interesting, AI's utility for large text analysis is not novel. As such, the suggestions (in the discussion section) are trivial how policy makers or business managers may make use of the developed climate solutions measure. Additionally, you lack a comprehensive review of existing research on climate opportunities, which is essential for contextualizing your findings and clarifying the research gap.

Logical reasoning: The manuscript struggles to provide convincing arguments for some of its core claims. The focus on opportunities, for example, lacks persuasive evidence (e.g., Line 213), and the practical relevance of the climate solutions measure is insufficiently justified. Attributing revenue growth solely to climate solutions is speculative, as the analysis does not control for other influencing variables.

Methods/Data: Incorporating additional data sources and comparative analyses would be beneficial to strengthen the manuscript. For instance, referencing how other studies have measured climate opportunities, such as using CDP questionnaire data, and comparing these with the 10-K filings could add depth. This cross-validation could help establish the robustness of your climate solutions measure. I also have concerns regarding the reliability of the financial data, particularly the inclusion of R&D data across all companies. Many sectors do not report R&D, raising questions about skewed data and potential bias. How is this addressed in your analysis? Additionally, Table S1 shows a highly skewed distribution of the Climate Solutions Measure, which is not sufficiently discussed. More transparency about managing this skew in your analysis would improve clarity.

For the OLS regression Climate Solutions is measured by scope 1 emissions intensity. I have doubts about this measurement. First, scope 1 emissions can also be reduced by a lot of different things that are not related to climate solutions / opportunities, e.g. outsourcing of emission intensive business activities; only efficiency increased; inflation, pure revenue increase. Second, through climate solutions also scope 2 emissions can be affected; but this is not captured. Further, this limited focus on scope 1 is especially concerning for industries like automotive, where Scope 3 emissions – representing the majority of climate impact – are crucial for assessing a company's future viability in a decarbonized economy. Third, it is not clear to me whether you utilized CO2 data or GHG data (in the charts it seems the latter, but in the text a clear reference is missing). Fourth, I would like to understand how you addressed data limitation issues; many papers have found that corporate emission data is not very reliable and differs a lot from data provider to data provider.

Finally, I need more information how the Carbon Abatement Costs were estimated. As now, I do not understand this. Furthermore, Figure 6: Industry Convergence needs more elaboration; I don't get it. And one last remark: are the low Adj. R-squares not worrying?

Results: You document that firms incorporating climate solutions into their portfolios exhibit higher revenue growth and investment, and your topic analysis emphasizes solutions with high abatement potential or low costs. However, many findings remain surface-level and predictable. Given market and regulatory trends, the rise of renewable energy and biofuels over time is not surprising. Additionally, there is a notable gap between statistical results (e.g., Line 115 and following) and their interpretation (e.g., Line 152 and following). Providing more precise explanations and a more thorough discussion of Figure 1's significance, such as clarifying what 1-5% engagement levels mean, would enhance your argument. Moreover, your claim that climate solutions represent clear business opportunities is not convincingly demonstrated. Without robust causal analysis and adequate control variables, the evidence remains speculative. Attributing financial performance improvements solely to climate solutions overlooks other influencing factors. Some conclusions, such as whether firms respond to or actively shape demand changes (e.g., Line 160 and following), are speculative and lack empirical support. The discussion of abatement potential and cost also needs to be more developed to draw meaningful conclusions for the field.

I apologize that I cannot be more positive about the current version of your manuscript. I hope my comments will help improve the paper.

Reviewer #2

(Remarks to the Author)

Reviewer #3

(Remarks to the Author)

Please view the attached file: "Referee_Report_NatureComms.pdf"

Version 1:

Reviewer comments:

Reviewer #1

(Remarks to the Author)

Thank you for your revised manuscript and the detailed point-by-point response. I acknowledge the substantial revisions made and appreciate your efforts to improve the structure, strengthen the empirical foundation, and clarify several claims. The newly introduced analyses and expanded validation efforts mark a significant improvement. However, some concerns remain unresolved or only partially addressed. Below, I outline the areas where I believe the manuscript needs further refinement.

Conceptual framing and theoretical foundation: The manuscript still does not sufficiently explain why measuring climate opportunities is so important/relevant. I agree that it is; I suggest to be more explicit. You emphasize the relevance of the "opportunity" perspective by demonstrating a lack of correlation between climate risk and opportunity measures. While the absence of correlation is an interesting result, it does not by itself demonstrate that risk and opportunity represent conceptually distinct dimensions. I recommend either supporting this point with theoretical and empirical arguments or presenting it more cautiously as a suggestion for future research. In addition, the theoretical basis for some key claims – such as the link between the climate solutions measure and revenue growth – should be more clearly grounded in relevant literature.

Interpretation of statistical results and methodological concerns: Your efforts to avoid causal language and to add robustness checks (e.g., entropy balancing, forward-looking revenue growth, firm fixed effects) are excellent. Nevertheless, there are still some areas that could benefit from further clarification or development: (1) The manuscript still does not sufficiently address the possibility of confounding variables. The set of control variables is narrow which limits confidence in the claim that climate solutions and revenue growth are truly correlated. (2) Adjusted R^2 values remain low. While similar values have been reported in the literature, and it is understandable that explanatory power may be limited when introducing a novel metric, the manuscript should nonetheless acknowledge this and discuss potential reasons for the low R^2 values. (3) The decision to impute missing R&D data with zeros raises serious concerns, as it likely distorts the distribution and introduces systematic bias. While alternative proxies have been added, this does not resolve the methodological concern unless more advanced imputation or exclusion strategies are employed. (4) A noticeable gap remains between statistical results and their interpretation. Some conclusions go beyond what is supported by the underlying data. For instance, statements such as "products or services that enable the economy to decarbonize by 2022" (e.g., lines 310f and 346ff) overstate the implications of the text analysis. Such language suggests realized macroeconomic impact, whereas the method merely identifies the presence of climate-related terminology in financial filings. I strongly recommend revising such formulations to better reflect

the scope and limitations of the approach.

Application examples: The newly added use cases (Application Examples 1 and 2) introduce valuable perspectives and strengthen the applied relevance of the introduced metric. However, due to methodological shortcomings, the reliability of the results remains limited.

In Application Example 2, political affiliation is proxied using presidential election results from 2020 (and 2016 as a robustness check), which raises questions given that the data covers the period 2005-2022 and climate policy in the U.S. is often shaped at the state – rather than federal – level. Moreover, many of the firms included in the analysis are likely to operate across multiple countries and, therefore, not solely influenced by U.S. state-level political dynamics. These limitations affect the interpretability of the findings and should be clearly acknowledged. Additionally, the observed differences are relatively small – especially considering that the outcome variable reflects only the proportion of climate-related language in financial filings. Considering the limited magnitude of the observed differences, a more cautious interpretation would be appropriate. Application Example 3, by contrast, still contributes little beyond confirming well-known regulatory and market trends. While the topic has the potential to yield meaningful insights if analyzed in more depth, in its current form, this example does not add substantive value to the manuscript.

Overall, the manuscript shows great progress. The core idea – developing a quantitative measure for climate-related business opportunities based on AI analysis of financial disclosures – is timely and relevant. However, I still see some shortcomings. With further work on refining the empirical models and strengthening the theoretical underpinnings of key relationships, I believe this paper has strong potential.

I hope these comments support the further development of your work.

Reviewer #2

(Remarks to the Author)

Reviewer #3

(Remarks to the Author)

I copy the conclusion here. The rest is in the attached PDF file.

Conclusion:

I remain positive about this paper. In my opinion, the work on the LLM method has not been conducted thoroughly enough. However, the results and implications might not change drastically with a reworked approach. Nevertheless, this work has to be done with a high degree of carefulness and rigor. The analysis seems to diverge towards a really useful state.

Version 2:

Reviewer comments:

Reviewer #1

(Remarks to the Author)

Thank you for submitting the revised version of your manuscript and for providing a detailed point-by-point response. We appreciate your thoughtful and thorough engagement with our previous review comments. In our view, the revisions have significantly strengthened the manuscript, and we are pleased to see that our suggestions were carefully considered and effectively implemented.

At this stage, we have only a few minor suggestions to further enhance clarity and framing:

1. The argument that 'climate solutions represent a growing market under the rise of climate transition risks' (page 10, lines 228f) is particularly compelling. To strengthen the overall framing and relevance of your paper, we recommend highlighting this point earlier in the manuscript, ideally in the introduction.

2. For the sake of clarity, we suggest being more explicit about how the climate opportunity measure you introduce is defined. A brief clarification would help readers understand whether the measure is based solely on the methodological approach or whether it results from the combination of the method and the specific data source used (i.e., 10-K filings).

Overall, we find the manuscript to be in excellent shape and greatly appreciate the improvements made during the revision process. We hope these final remarks are helpful and wish you success with the publication.

Reviewer #2

(Remarks to the Author)

Reviewer #3

(Remarks to the Author)

Dear Authors,

I will not send a PDF this time because I think the manuscript is at a very good level, and you successfully addressed my remaining concerns.

I have to outline that, from a computer science point of view, you improved the paper drastically. Employing five-fold cross-validation consistently, using consistent metrics, evaluating model/temperature on a held-out dataset, and even confirming that potential influences in the regression are negligible is very convincing.

The analysis itself profits from being a bit more conservative in its interpretation. I think I was aligned with the other reviewers in demanding more caution.

In conclusion, I do not have more concerns to add.

All the best,
Reviewer 3

Response to Reviewers 1 and 2 (Manuscript NCOMMS-24-53291-T)

First, we would like to thank you for your insightful and constructive comments. They have been very helpful in revising and improving this version of the paper. We appreciate the opportunity to resubmit and have made a sincere effort to address all of your comments.

In this response, we start by highlighting the major changes to the paper since the first submission. We then provide a detailed response addressing each of your comments, where your comments are reproduced in boxed *italics*, followed by our response in a regular typeface.

Main changes to the manuscript:

1. Strengthen contribution to the literature by:
 - a. Significantly expanding the discussion of the literature on climate opportunities and placing the contribution of our paper in the context of this interdisciplinary literature.
 - b. Clarifying that the source of information leveraged in our analysis (i.e., regulated accounting reports instead of unregulated sustainability disclosure mechanisms) is novel and important.
2. Restructured the Results section to strengthen the contribution and robustness of key results.
 - a. Restructured the paper to more clearly present its contribution by first introducing the measure, then validating it, and then using the measure in three applications to derive key insights.
 - b. Introduced new analyses to validate the measure using a host of analyses showing that it predictably varies with estimated measures of green patents, revenues, innovation measures, and in response to major policy interventions.
 - c. Expanded the greenhouse gas analysis to include all three scopes of emissions, both intensity and absolute, to strengthen the section comparing climate solutions to climate risks.
 - d. Introduced new analyses that examine the association between climate solutions and revenue growth under different conditions (e.g., patent protection and carbon abatement costs) that provide deeper insights.
 - e. Introduced new robustness tests that increase confidence in the interpretation of the results, such as forward-looking revenue growth analysis and entropy-balanced matched sample analysis.
 - f. Introduced new analyses that examine the evolution of the climate solutions measure for firms operating in states with different political affiliations and under different conditions (e.g., carbon abatement costs) that provide deeper insights.
 - g. Sharpened the language to enhance the clarity of result interpretations, and discussion on both the limitations of our analyses and opportunities for future research.
3. LLM methodology enhancement
 - a. Enriched discussion about LLM methodology, such as providing Cohen's Kappa.

- b. Conducted additional robustness checks, such as benchmarking to additional models (DistilRoBERTa, RoBERTa, and DeBERTa) and comparing sensitivity to parameter changes.

Your specific comments:

1. Many thanks for submitting this interesting manuscript to Nature Communications. The topic is engaging, as tracking business opportunities for climate solutions using AI in regulated accounting reports is timely and relevant. However, the results are predictable, and the overall contribution – beyond using this new method – remains unclear. I also struggle to see a justification for the manuscript’s suitability for an interdisciplinary journal like Nature Communications with the aim to publish work from “all areas of the biological, health, physical, chemical, Earth, social, mathematical, applied, and engineering sciences”, rather than a more specialized journal in finance, management, or sustainability.

Response:

Your comment helps us think more thoroughly on our paper’s contribution both in relation to the broader interdisciplinary research on climate opportunities, and on key results that provide new insights. In the revised manuscript, we tried our best to enhance the contribution on both fronts.

First, we revised the front end of the paper to better position it in the broader literature on climate opportunities. Doing so allows us to articulate how our paper adds to this literature. For example, IPCC Chapter 16 on Innovation, Technology Development and Transfer discuss both a lack of good measurement on climate innovation and the value of having systematic analysis of climate solutions—gaps that our paper directly addresses. Additionally, we added reference to papers on climate solutions in interdisciplinary journals, with the observation that most of them focus on specific technologies (e.g., batteries, solar panels, EVs) and are more relevant for the development of climate solutions among startups. This observation presents an opportunity for us to shed light on the overall climate solutions landscape among large incumbent firms that are pivotal for the scaling and deployment of climate solutions. We provide this discussion in the first three paragraphs of the introduction.

Second, we restructured our results section and strengthened our analyses to enhance key takeaways from our results. We restructured the results section to start with introducing our Climate Solutions Measure, provide validation, describe topic characteristics, and then present three sets of applications using our data to study research questions related to climate solutions. By separately showing the three sets of applications, we hope to strengthen each of the analyses and present key takeaways more clearly. We also included additional robustness analyses, where we discuss in more detail in response to your other comments below. Notably, we added one new application with a new set of results: we examine how the political affiliation of a firm’s operation location is associated with their climate solutions. Many of the related papers on this topic on political affiliation are published in interdisciplinary journals, such as Nature Communications and Nature Climate Change (e.g., Mayer and Smith, 2023; Berkebile-Weinberg et al., 2024).

Third, we appreciate the recognition that tracking business opportunities for climate solutions using AI in regulated accounting reports is timely and relevant, and we revised writing to further strengthen this point. In the Discussion section, we positioned our measure in relation to other climate opportunities measures, highlighting the value of using regulated financial reports as a credible source of information. Furthermore, in the introduction, we added a paragraph discussing the significance of being able to rely on regulated financial reports to measure climate solutions:

Applying AI to analyze companies' communication of climate solutions in the Business Description section of the 10-K allows us to derive measures of climate opportunities for each firm-year. While significant efforts have been made to increase corporate disclosure on the implications of climate change for businesses, through the issuance of sustainability reports and other voluntary disclosures (Bochkay et al., Forthcoming; Grewal et al., 2019), this study documents that existing mandatory disclosures contain useful information about climate opportunities. This result is important because efforts for new mandatory climate disclosures are challenged politically and legally (Lashitew and Mu, 2024) and voluntary disclosures suffer from reliability issues (Pinnuck et al., 2021). We hope that collectively, these enhancements help strengthen the contribution and relevance to readers in an interdisciplinary journal like Nature Communications.

2. Clarity of writing / way of arguing: The manuscript is well written and accessible, but statistical results must be more clearly explained and interpreted, with illustrative examples (e.g., Line 115) to reach an interdisciplinary audience.

Response:

We have revised the results section holistically to ensure that the statistical results are more clearly explained and interpreted, with illustrative examples such that it is easier for a broad audience to understand the results. We also examined a number of papers in Nature Communications to ensure we follow a similar way to describe results.

To give an example, when describing the time trend in Climate Solutions Measure (Fig 1a), we provide sample statistics to illustrate the growth between 2005 to 2022:

Consistent with companies increasingly developing climate solutions products and services for their customers, the Climate Solutions Measure is trending upward over time, growing from 1% in 2005 to 4% in 2022 (Fig. 1a). The percentage of firms with a Climate Solutions Measure exceeding 1% doubled from 20% in 2005 to 45% in 2022, whereas those exceeding 5% quadrupled from 5% in 2005 to 20% in 2022.

As another example, when describing industry dispersion for different climate solutions topics (Fig 8b), we provide illustrative examples specifying the industries and the relevant datapoints from the figure.

The topic with the least dominance from one industry is building retrofitting, where we observe the interconnectedness of the capital goods (24%), equity real estate investment trusts (23%), and materials (21%) industry groups (Fig. 8b). In contrast, the most

concentrated topic is plant-rich diets, where 93% of climate solutions sentences come from the food, beverage, and tobacco industry group. Among renewable energy topic groups, we observe variation where geothermal power is more concentrated in the utilities industry group, whereas wind and solar are more interconnected with the capital goods industry group, where energy storage is involved.

3. Contribution: Unfortunately, the contribution to the literature is not clear to me. While developing a climate solutions measure using AI is interesting, AI's utility for large text analysis is not novel. As such, the suggestions (in the discussion section) are trivial how policy makers or business managers may make use of the developed climate solutions measure. Additionally, you lack a comprehensive review of existing research on climate opportunities, which is essential for contextualizing your findings and clarifying the research gap.

Response:

You raise three important points in this comment. The first point is that a comprehensive review of existing research on climate opportunities helps contextualize our findings and clarify research gaps. We implemented this suggestion and provided details in response to your first comment above.

The second point is that AI utility for large text analysis is not novel, and together with comments from Reviewer 3, we added a paragraph in the Discussion section that highlights the strength of our measure relative to other measures on climate opportunities. For convenience, we pasted the relevant section from page 15 in the Discussion section below:

Relative to other measures on climate opportunities derived from voluntary disclosures, such as earnings conference calls and CDP reports, (Sautner et al., 2023; Li et al., 2024; Toetzke et al., 2024), our measure differs by focusing on products and services that are disclosed in regulated financial filings, with greater credibility, lower noise, and less subject to concerns about the self-selection of disclosures. Additionally, our measure is more likely to capture climate solutions already integrated into a firm's existing product portfolio, rather than input-based proxies like R&D expenses or labor skills, which reflect investment efforts rather than commercialized solutions. Consistent with this expectation, our Climate Solutions Measure is positively associated with revenue growth. In contrast, climate exposure and opportunity measures derived from earnings conference calls are more reflective of market attention and not associated with revenue growth when we include them in the same regression with Climate Solutions Measure (Sautner et al., 2023) (Supplementary Table 6 Panel E)

The third point is that our Discussion section, which previously suggest ways stakeholders can use the data, can be enhanced. In the revised manuscript, we rewrote the discussion section to link more closely to the literature and the insights from our results. Specifically, we first highlight insights from our three applications of the Climate Solutions Measure. We then discuss how our measure relates to existing literature on climate opportunities, as pasted in the paragraph above.

Thanks to your suggestion, we believe the Discussion section delivers more sound and robust takeaways for readers.

4. Logical reasoning: The manuscript struggles to provide convincing arguments for some of its core claims. The focus on opportunities, for example, lacks persuasive evidence (e.g., Line 213), and the practical relevance of the climate solutions measure is insufficiently justified. Attributing revenue growth solely to climate solutions is speculative, as the analysis does not control for other influencing variables.

Response:

We agree that we need to be more careful in providing evidence to support core claims. We have revised the paper to ensure that our claims are supported by evidence. For example, in the Results section, we removed all claims that are not backed by our results, such as the previous line 213. Additionally, when providing supporting claims not shown in our results, we made sure we cited relevant papers, such as in the following example on page 10:

At the same time, it is not clear if firms with higher climate solutions will exhibit higher revenue growth because there is high uncertainty regarding the demand for climate solutions given uncertainties in climate regulations, market preference, and technology developments (Blyth et al., 2007).

We also revised language to more carefully describe result interpretations that are supported by our empirical design. For example, in the revenue growth analysis, we are careful not to use causal language (e.g., lead to, cause), and only describe results as firms with higher Climate Solutions Measures are *associated* with higher revenue growth. At the same time, we added robustness tests to mitigate concerns about confounding factors driving our results on revenue growth. We discuss these tests in more detail below in response to comment #9.

5. Methods/Data: Incorporating additional data sources and comparative analyses would be beneficial to strengthen the manuscript. For instance, referencing how other studies have measured climate opportunities, such as using CDP questionnaire data, and comparing these with the 10-K filings could add depth. This cross-validation could help establish the robustness of your climate solutions measure. I also have concerns regarding the reliability of the financial data, particularly the inclusion of R&D data across all companies. Many sectors do not report R&D, raising questions about skewed data and potential bias. How is this addressed in your analysis? Additionally, Table S1 shows a highly skewed distribution of the Climate Solutions Measure, which is not sufficiently discussed. More transparency about managing this skew in your analysis would improve clarity.

Response:

Thanks to your suggestion, we added the following to enhance our methods and data.

First, we added a section to validate our climate solutions measure, including using additional data sources. For convenience, we pasted the relevant sections from pages 5-6 in the Results section below. We also provide more details about the methodology, including variable construction, regression specifications, in the Methods section.

The validation analysis further confirms that the Climate Solutions Measure derived from 10-Ks reflects companies' business portfolio involvement in climate solutions. The Climate Solutions Measure is positively associated with the percentage of green patents and green revenues estimated by MSCI, one of the largest data providers, for the most recent fiscal year with available data (Fig. 2a). A one standard deviation increase in Climate Solutions Measure is associated with a 0.47 and 0.54 standard deviation increase in green revenue and green patent percent, respectively.

Following your suggestion, we also examined the CDP questionnaire data to identify questions relevant to climate opportunities. Specifically, one question asks “Provide details of your products and/or services that you classify as low-carbon products.” (Question C4.5a in CDP 2023). We use the data on the percent of revenue generated from low-carbon products or services (*CDP Low Carbon Revenue %*) and repeat our validation analysis. One challenge is that firms voluntarily disclose to CDP, and hence our sample for this analysis only contains 933 observations spanning 2015-2022 for 259 firms. We present summary statistics and regression results in Appendix 1 to this response document, where *Climate Solutions Measure* is significantly positively associated with *CDP Low Carbon Revenue %* only when industry-year fixed effects are included. One potential reason is that firms self-select to disclose to CDP, and these firms have, on average, 33% low carbon revenue, which limits the variation in the data. As such, we currently do not include this analysis in the paper.

Second, to mitigate concerns about missing values in R&D expense data, we added three additional proxies to measure a firm's innovation. We also include this analysis as part of validation tests to support that our Climate Solutions Measure likely captures firms innovating on climate solutions. We pasted the relevant sections from page 6 in the Results section below. We also provide more details about the methodology, including variable construction, regression specifications, in the Methods section.

As a second validation test, we observe that a higher Climate Solutions Measure is associated with higher innovation across four different measures. With industry-year fixed effects, a one standard deviation higher Climate Solutions Measure is associated with a 3% higher research and development expense scaled by revenue, 9% higher knowledge capital scaled by revenue, 6% higher research and development capitalized (RDC) scaled by revenue, and 10% higher trade secret scaled by revenue (Fig. 2c). In the absence of data on climate investments, this positive association provides support that the Climate Solutions Measure reflects firms engaging in more innovation.

Third, to mitigate concerns about the skewness of Climate Solutions Measure due to a large number of zeros, we provide additional evidence using indicators for observations with Climate Solutions Measure above 1% and 5%. For example, as part of the robustness test for the revenue

growth results, we present findings using this alternative variable for climate solutions. We also describe that this test mitigates concern about the right skew of the main Climate Solutions Measure. We provide this discussion on page 25 in the Methods section. A related note is that we added discussion to contextualize the meanings of firms with Climate Solutions Measure above 1% and 5%, which we provide more detail below in response to your comment #8.

6. For the OLS regression Climate Solutions is measured by scope 1 emissions intensity. I have doubts about this measurement. First, scope 1 emissions can also be reduced by a lot of different things that are not related to climate solutions / opportunities, e.g. outsourcing of emission intensive business activities; only efficiency increased; inflation, pure revenue increase. Second, through climate solutions also scope 2 emissions can be affected; but this is not captured. Further, this limited focus on scope 1 is especially concerning for industries like automotive, where Scope 3 emissions – representing the majority of climate impact – are crucial for assessing a company’s future viability in a decarbonized economy. Third, it is not clear to me whether you utilized CO2 data or GHG data (in the charts it seems the latter, but in the text a clear reference is missing). Fourth, I would like to understand how you addressed data limitation issues; many papers have found that corporate emission data is not very reliable and differs a lot from data provider to data provider.

Response:

We agree with your comment that focusing on scope 1 greenhouse gas emissions intensity is limiting and that other scopes of greenhouse gas emissions can change in relation to climate solutions. In the revised manuscript, we include all scopes of greenhouse gas emissions, both intensity and total emissions, and present the analysis as being more agnostic as to the prediction. We also revised the writing to consistently show that we use greenhouse gas emissions in both the figures and texts. Across all specifications, we find a weak relationship between climate solutions and greenhouse gas emissions, and use this finding to highlight that our measure captures a distinct group of firms that climate risks measures cannot identify. We pasted the relevant sections from pages 7-8 in the Results section below.

Ex-ante, it is not clear whether and how greenhouse gas emissions are correlated with the Climate Solutions Measure. On the one hand, firms in some high-emissions industries, such as utilities and oil and gas, have more innovation opportunities as documented in prior literature through green patents (Cohen et al., 2024). On the other hand, climate solutions opportunities likely also arise in new industries that do not face high transition risks. For example, industries such as electrical equipment or automobile components have a high Climate Solutions Measure but low scope 1 greenhouse gas emissions intensity (Fig. 3a). In contrast, industries such as passenger airlines, marine transportation, or construction materials have high scope 1 greenhouse gas emissions intensity but a comparatively lower Climate Solutions Measure.

Overall, we observe a moderate relationship between climate transition risks and opportunities (Fig. 3b). Unconditional on firm industry membership, a one standard deviation increase in Climate Solutions Measure is associated with a 0.15 standard deviation higher scope 1 greenhouse gas emissions intensity. Conditional on industry membership, we do not find a statistically significant association between risk and opportunity. In other words, when comparing firms within an industry, analyzing the risk measure provides no information about opportunities.

This relationship extends to other scopes of greenhouse gas emissions, and to absolute or intensity measures. Without industry-year fixed effects, the relation between risk and opportunity is statistically significant and varies in the direction across different scopes of greenhouse gas emissions, but the magnitudes are small, less than 0.2 standard deviation of the Climate Solutions Measure. With industry-year fixed effects, most relationships become statistically insignificant, further reinforcing that any observed correlations between climate risks and opportunities are weak and do not hold within industries.

To address potential concerns about the quality of corporate greenhouse gas emissions data, we repeat the analysis using a subsample of firms that likely report more reliable data. Specifically, we identify firms with a sustainability committee on their board, as this reflects stronger governance over environmental reporting. The results remain similar, as shown in Supplementary Table 5 Panels C and D. We provide this discussion on page 22. Additionally, in Appendix 2 to this response document, we replicate Fig. 3b using this subsample, and the results are similar to Fig. 3b presented in the manuscript.

7. Finally, I need more information how the Carbon Abatement Costs were estimated. As now, I do not understand this. Furthermore, Figure 6: Industry Convergence needs more elaboration; I don't get it. And one last remark: are the low Adj. R-squares not worrying?

Response:

We agree that the carbon abatement costs can be further elaborated. We enriched the description on pages 22-23 in the Methods section, and pasted below for convenience:

Project Drawdown provides estimates for each climate solution's abatement potential and costs to achieve the carbon abatement (Project Drawdown, 2020). These estimates are based on analytical models backed by extensive literature and data to estimate the relevant impact based on two scenarios for the period of 2020 to 2050. The first scenario is more conservative and estimates a two-degree temperature increase by 2100. The second scenario is more ambitious and estimates a 1.5-degree temperature increase by 2100. We use the estimates from the first scenario to stay on the conservative side.

For each solution, abatement potential is the CO₂ equivalent reduction brought by the technology between 2020 and 2050. In terms of costs, Project Drawdown provides the first cost and lifetime cost. We focus on the first cost, as data on lifetime cost is less available.

The first cost reflects the upfront capital expenditure required to implement the climate solution, and is based on the relative cost to implement these climate solutions between 2020 and 2050 compared to a baseline scenario. The baseline scenario is defined as a scenario where such a climate solution does not exist. For example, the baseline for the climate solution, online wind turbines, is based on electricity generated using fossil fuel power plants. As such, negative costs can happen if the climate solution results in lower cost relative to the baseline scenario (such as the use of LED lighting). To ensure that costs are comparable across different climate solutions, we focus on the first cost divided by the abatement potential. This allows us to compare different climate solution's cost per unit of carbon reduction.

Fig. 5a plots each topic where the x-axis shows the average cost per abatement (USD billion per Gt CO₂e abatement) and the y-axis shows the abatement potential (Gt CO₂e) of the climate solutions technology. Fig. 5b plots a similar chart at the industry level. To create this, we first calculate the average Climate Solutions Measure for each topic within an industry. Then, we compute the industry's weighted average cost per abatement and abatement potential, using each topic's relative share within the industry as weights.

We also agree that the Figure on industry convergence (previously Fig. 6, now Fig. 8) can be elaborated. To facilitate interpretation of the figure, we added a new table in Fig. 8a, where we list the top 3 industries for each of the topics displayed in Fig. 8a. We also revised the writing to enhance the readability, and the new Fig. 8b allows us to provide more illustrative examples. We pasted the relevant sections from pages 13-14 in the Results section below.

To analyze how industries are converging on similar climate solutions, we plot each climate solutions sentence based on their embedding proximity. Each dot represents a sentence and the sentences are color-coded based on the industry group of the firm. Two dots are closer in proximity if the embeddings are more similar. In other words, dots on similar climate solutions would cluster in the same area.

We observe a blurring of industry boundaries, with previously unrelated industries engaging in similar products because of climate solutions (Fig. 8a). Specifically, we observe biomass power (e.g., ethanol) in the energy (42%), food, beverage and tobacco (18%), and materials (18%) industry groups. This pattern reflects the use of biofuels, which involve converting crops into fuel---such as ethanol produced by fermenting the sugar content in corn. We also observe a close connection between the automobiles and components, and capital goods industry groups in climate solutions relating to electric vehicle production, where these two industry groups are most represented in the electric cars, hybrid cars, and energy storage topics.

The topic with the least dominance from one industry is building retrofitting, where we observe the interconnectedness of the capital goods (24%), equity real estate investment trusts (23%), and materials (21%) industry groups (Fig. 8b). In contrast, the most concentrated topic is plant-rich diets, where 93% of climate solutions sentences come from

the food, beverage, and tobacco industry group. Among renewable energy topic groups, we observe variation where geothermal power is more concentrated in the utilities industry group, whereas wind and solar are more interconnected with the capital goods industry group, where energy storage is involved.

Finally, regarding the low adjusted R-squared, we provide two key points of support. First, for all regressions, we present results with two specifications: one without fixed effects and one with industry-year fixed effects. Across these regressions, we observe that the inclusion of industry-year fixed effects consistently improves the R², explaining more of the variation. Second, for instances where the adjusted R² remains below 20%, we benchmark our results against prior studies that use similar outcome variables. This applies particularly to our main results where innovation measures (Supplementary Table 4, Panel B; R² of 9.5%-13.6% without fixed effects and 14.7%-24.3% with fixed effects) and revenue growth (Supplementary Table 6; R² of 5.9% without fixed effects and 13.5% with fixed effects) are the dependent variables. For example, when revenue growth is the dependent variable, Thornhill 2006 reports a R² of 3% without fixed effects (Table 4); Albring et al., 2013 report a R² of 10.7% with industry fixed effects (Table 2). For R&D intensity, Lin and Wang 2016 report a R² of 14.7% with industry fixed effects (Table 3).

8. Results: You document that firms incorporating climate solutions into their portfolios exhibit higher revenue growth and investment, and your topic analysis emphasizes solutions with high abatement potential or low costs. However, many findings remain surface-level and predictable. Given market and regulatory trends, the rise of renewable energy and biofuels over time is not surprising. Additionally, there is a notable gap between statistical results (e.g., Line 115 and following) and their interpretation (e.g., Line 152 and following). Providing more precise explanations and a more thorough discussion of Figure 1's significance, such as clarifying what 1-5% engagement levels mean, would enhance your argument.

Response:

Thank you for these suggestions; we have followed your suggestions and restructured the results section to enhance the interpretation of the results. We restructured such that we first introduce and validate the measure, then show climate solutions topics (including key characteristics such as abatement potential and cost). Then we present three applications using this data, where we added new analysis to increase the depth of the findings. Below are three key findings or additions.

First, on the revenue growth result, we added a new cross sectional test on patents motivated by IPCC. Below, we pasted the relevant discussion on page 11 from the Results section.

Next, we examine under what conditions the revenue growth is more pronounced. Motivated by IPCC's observation that the role of intellectual property rights in climate innovation is not well understood, with some viewing it as a barrier and others as an enabler, we examine how the relationship with revenue growth varies in industries with more patent protection (Blanco et al., 2022). We separate observations based on the median industry-

level patent amount, and find that the estimated coefficient is only significantly positive in high-patent industries (Fig. 6a).

Second, we added a new set of analysis on political affiliation, showing that among low cost technologies, there is no difference between Republican operating and Democratic operating firms. Below, we pasted the relevant discussion on pages 12-13 from the Results section.

We group firms into majority Democratic operating firms and Republican operating firms based on their employee distribution in different states, where the political affiliation of each state is based on the 2020 presidential vote outcomes. On average, Republican operating firms have a lower Climate Solutions Measure than Democratic operating firms, with average Climate Solutions Measures of 2.1% and 2.5%, respectively. To ensure other firm characteristics are kept constant, we compare within industry and include controls for firm size and age, and find that this political gap in climate solutions persists throughout the sample period (Fig. 7a). Separately examining climate solutions with different topic characteristics reveals variation in the gap (Fig. 7b). Notably, the gap is largest among climate solutions with high cost per abatement, where on average, Republican operating firms have a significantly lower measure by 0.14 standard deviations relative to Democratic operating firms (Supplementary Table 7). In contrast, among technologies with low cost per abatement, the average Climate Solutions Measures of Republican and Democratic operating firms are not statistically different. The lower engagement of firms in Republican operating states for higher abatement cost solutions that require more policy support to be commercialized likely reflects the comparatively lower policy support for climate technologies in these states. We observe a similar pattern where the gap is statistically significant for climate solutions with low abatement potential, but not for high abatement potential.

Third, following your suggestion, we added new validation tests that help enrich the discussion of Figure 1. We clarify the meaning of 1% and 5% engagement levels by benchmarking them to MSCI green revenue and patents. We also added difference-in-differences regression analysis to validate the key observations from Figure 1 relating to changes in Climate Solutions Measures after major political interventions (including IRA and RPS). Below, we pasted the relevant paragraphs on pages 6-7 from the Results section.

The Climate Solutions Measure, green revenues, and green patents measures increase monotonically across groups of Climate Solutions Measures (Fig. 2b). For example, firms with Climate Solutions Measure greater than 0 but less than 1 (Group 1), and greater than 1 but less than 5 (Group 2), have on average green revenue percent of 2 and 6, respectively. This increases to approximately 13 percent for firms with Climate Solutions Measure greater than 5 but less than 10 (Group 3). Firms with Climate Solutions Measure greater than 10 (Group 4) have green revenue percent of more than 35. Green patents increase from approximately 5 percent for Group 1 to approximately 46 percent for Group 4.

In a third validation test, we find that the Climate Solutions Measure varies predictably in response to major events. Most notably, we see the largest jump in the most recent years, which represents the information in fiscal years 2021 and 2022 disclosed in 2022 and 2023, respectively. The Inflation Reduction Act (IRA), passed in 2022, provided hundreds of billions of dollars in incentives for the development and acceleration of climate solutions. In industries with more IRA opportunities, the Climate Solutions Measure on topics covered by the IRA increases significantly in fiscal years 2021 and 2022, relative to prior years (Fig. 2d). We also observe an increase between 2005 and 2010, during a period where over 20 states in the US passed the Renewable Portfolio Standards (RPS) that mandate a minimum ratio of renewable energy supply in a state. The Climate Solutions Measure on topics related to renewable energy increases significantly after the staggered adoption of RPS for firms in industries relevant to renewable energy (Fig. 2d).

9. Moreover, your claim that climate solutions represent clear business opportunities is not convincingly demonstrated. Without robust causal analysis and adequate control variables, the evidence remains speculative. Attributing financial performance improvements solely to climate solutions overlooks other influencing factors. Some conclusions, such as whether firms respond to or actively shape demand changes (e.g., Line 160 and following), are speculative and lack empirical support. The discussion of abatement potential and cost also needs to be more developed to draw meaningful conclusions for the field.

Response:

We agree that our analysis examining the revenue growth of firms with higher Climate Solutions measurements can be enhanced with more robust analyses. We added three sets of robustness tests for this analysis to mitigate concerns with reverse causality or confounding variables. For convenience, we pasted the relevant sections from page 25 in the Methods section below.

We conduct three robustness tests to mitigate potential concerns related to the revenue growth results. First, to mitigate concerns about reverse causality, we replace revenue growth with the one-year forward revenue growth, and three-year moving average revenue growth from $t=0$ to $t=2$, and our results remain robust. Second, we include firm fixed effects to control for time-invariant firm characteristics, and the results remain robust when we use the three-year moving average revenue growth. We focus on the three-year moving average because revenue growth tends to be slow-moving, and using firm fixed effects with one-year revenue growth may absorb too much of the variation, limiting our ability to detect meaningful effects. Third, to ensure comparability between firms with and without climate solutions, we apply entropy balancing on the first three moments of firm size and firm age (Hainmueller, 2012). As entropy balancing requires discrete treatment variables, we classify firms as with climate solutions if *Climate Solutions Measure* is above 1% or 5%, which also helps mitigate concern about the right skew of the main *Climate Solutions*

Measure. Our results remain robust when we repeat the main specifications with these alternative measures for climate solutions, and when using entropy balanced samples.

Additionally, we carefully reviewed the manuscript to ensure that all statements are directly tied to our empirical analysis and do not extend beyond the scope of our results. For example, speculative statements, such as the one referenced on Line 160, have been removed to maintain a clear focus on objective, data-driven findings. We now ensure that all interpretations remain grounded in the presented evidence.

Finally, we agree that the discussion of abatement potential and cost needs to be further developed to draw more meaningful conclusions for the field. We emphasize these two characteristics in the first two applications of the Climate Solutions Measure, demonstrating that they provide incremental insights beyond the overall measure. In the revenue growth analysis, we find that the positive effects are concentrated in technologies with higher abatement potential, potentially reflecting greater demand for solutions with stronger decarbonization capabilities. Similarly, in the political affiliation analysis, we find no significant political divide for technologies with low costs, supporting that firms leverage economic and operational advantages to pursue climate strategies that align with profitability (Allen et al., 2025). We discuss these observations on page 15 in the Discussion section.

Conclusion:

Thank you again for your time and effort in providing these insightful comments and suggestions. We have updated the manuscript based on the totality of comments from you, the Editor, and other reviewers, and we hope that laying out our updates and thought process above has helped speak to your concerns.

Reference:

Albring, S. M., Huang, S. X., Pereira, R., & Xu, X. (2013). The effects of accounting restatements on firm growth. *Journal of Accounting and Public Policy*, 32(5), 357-376.

Allen, F., Barbalau, A., Chavez, E. & Zeni, F. Leveraging the capabilities of multinational firms to address climate change: a finance perspective. *Journal of International Business Studies* 1–20 (2025).

Berkebile-Weinberg, M., Goldwert, D., Doell, K. C., Van Bavel, J. J. & Vlasceanu, M. The differential impact of climate interventions along the political divide in 60 countries. *Nature communications* 15, 3885 (2024).

Blanco, G. et al. Innovation, technology development and transfer. In IPCC, 2022: Climate Change 2022: Mitigation of Climate Change. Contribution of Working Group III to the Sixth Assessment Report of the Intergovernmental Panel on Climate Change, 2674–2814 (Cambridge University Press, 2022).

Blyth, W. et al. Investment risks under uncertain climate change policy. *Energy Policy* 35, 5766–5773 (2007).

Bochkay, K., Hales, J. & Serafeim, G. Disclosure standards and communication norms: Evidence of voluntary sustainability standards as a coordinating device for capital markets. *Review of Accounting Studies* (Forthcoming).

Cohen, L., Gurun, U. G. & Nguyen, Q. H. The esg-innovation disconnect: Evidence from green patenting. Available at SSRN 3718682 (2024).

Grewal, J., Riedl, E. J. & Serafeim, G. Market reaction to mandatory nonfinancial disclosure. *Management Science* 65, 3061–3084 (2019).

Hainmueller, J. Entropy balancing for causal effects: A multivariate reweighting method to produce balanced samples in observational studies. *Political analysis* 20, 25–46 (2012).

Lashitew, A. & Mu, Y. Corporate opposition to climate change disclosure regulation in the united states. *Climate Policy* 1–16 (2024).

Li, X., Luo, L. & Tang, Q. Climate risk and opportunity exposure and firm value: An international investigation. *Business Strategy and the Environment* (2024).

Lin, J. C., & Wang, Y. (2016). The R&D premium and takeover risk. *The Accounting Review*, 91(3), 955-971.

Mayer, A. P. & Smith, E. K. Multidimensional partisanship shapes climate policy support and behaviours. *Nature Climate Change* 1–8 (2023).

Pinnuck, M., Ranasinghe, A., Soderstrom, N. & Zhou, J. Restatement of csr reports: Frequency, magnitude, and determinants. *Contemporary Accounting Research* 38, 2376–2416 (2021).

Project Drawdown. *The drawdown review 2020 - climate solutions for a new decade* (2020).

Sautner, Z., Van Lent, L., Vilkov, G. & Zhang, R. Firm-level climate change exposure. *The Journal of Finance* 78, 1449–1498 (2023).

Thornhill, S. (2006). Knowledge, innovation and firm performance in high-and low-technology regimes. *Journal of business venturing*, 21(5), 687-703.

Toetzke, M., Probst, B., Feuerriegel, S., Anadon, L. D. & Hoffmann, V. H. Machine learning can help track climate technology innovation. Available at SSRN 4810933 (2024).

Appendix 1: Climate Solutions Measure and CDP Low Carbon Revenue

Panel A: Summary Statistics of CDP Low Carbon Revenue % by Fiscal Year		
	N	Mean
2015	54	32.523
2016	65	30.690
2017	96	29.592
2018	113	34.874
2019	138	36.789
2020	154	37.769
2021	160	32.003
2022	153	30.016
Total	933	33.375

Panel B: Regression		
	CDP Low Carbon Revenue %	
	(1)	(2)
Climate Solutions Measure	0.741 (1.40)	1.345*** (3.03)
N	933	933
Adj. R-squared	0.011	0.214
Gind-year FE	No	Yes
Clusters	Firm	Firm

Appendix 2: Climate Solutions and Climate Risks

Response to Reviewer 3 (Manuscript NCOMMS-24-53291-T)

First, we would like to thank you for your insightful and constructive comments. They have been very helpful in revising and improving this version of the paper. We appreciate the opportunity to resubmit and have made a sincere effort to address all of your comments.

In this response, we start by highlighting the major changes to the paper since the first submission. We then provide a detailed response addressing each of your comments, where your comments are reproduced in boxed *italics*, followed by our response in a regular typeface.

Main changes to the manuscript:

1. Strengthen contribution to the literature by:
 - a. Significantly expanding the discussion of the literature on climate opportunities and placing the contribution of our paper in the context of this interdisciplinary literature.
 - b. Clarifying that the source of information leveraged in our analysis (i.e., regulated accounting reports instead of unregulated sustainability disclosure mechanisms) is novel and important.
2. Restructured the Results section to strengthen the contribution and robustness of key results.
 - a. Restructured the paper to more clearly present its contribution by first introducing the measure, then validating it, and then using the measure in three applications to derive key insights.
 - b. Introduced new analyses to validate the measure using a host of analyses showing that it predictably varies with estimated measures of green patents, revenues, innovation measures, and in response to major policy interventions.
 - c. Expanded the greenhouse gas analysis to include all three scopes of emissions, both intensity and absolute, to strengthen the section comparing climate solutions to climate risks.
 - d. Introduced new analyses that examine the association between climate solutions and revenue growth under different conditions (e.g., patent protection and carbon abatement costs) that provide deeper insights.
 - e. Introduced new robustness tests that increase confidence in the interpretation of the results, such as forward-looking revenue growth analysis and entropy-balanced matched sample analysis.
 - f. Introduced new analyses that examine the evolution of the climate solutions measure for firms operating in states with different political affiliations and under different conditions (e.g., carbon abatement costs) that provide deeper insights.
 - g. Sharpened the language to enhance the clarity of result interpretations, and discussion on both the limitations of our analyses and opportunities for future research.
3. LLM methodology enhancement
 - a. Enriched discussion about LLM methodology, such as providing Cohen's Kappa.

- b. Conducted additional robustness checks, such as benchmarking to additional models (DistilRoBERTa, RoBERTa, and DeBERTa) and comparing sensitivity to parameter changes.

Your specific comments:

1. Overview of the paper

The paper explores how advancements in Large Language Models (LLMs) can be leveraged to identify climate-related business opportunities in regulated financial disclosures. Focusing on over 39,710 10-K filings from 4,483 U.S. public firms between 2005 and 2022, the study identifies a growing trend in companies incorporating climate solutions—such as electric vehicles and renewable energy—into their product portfolios. Thereby, the paper introduces the Climate Solutions Measure. It shows that companies integrating climate solutions into their product offerings experience increased revenue growth and investment. Moreover, a topic analysis highlights a concentration on climate solutions with higher abatement potential or lower costs.

Analysis

The paper exhibits both strengths and weaknesses. There are three core strengths. First, the paper explores an important perspective of business opportunities of climate change through technological climate solutions. This shift of perspective is of high importance since it outlines a way to advance with problems of climate change, rather than focusing on a passive, mitigative, and often backward looking perspective of risk. Second, the paper demonstrates a clever identification strategy of climate opportunities by analyzing the "Item 1 Business Description" section of 10-Ks. This section is under scrutiny by the U.S. SEC, auditors, and lawyers and is signed off by the CEO and CFO. Therefore, it is not just marketing communication but carries a higher-order validity. Third, the scope of investigations with the measure is suitable. Differentiating opportunities from risk, investigating the financial implications, as well as potential real-world effects of climate solutions, seem to be the core set of questions of interest in the field.

Response:

We appreciate the recognition of the importance of focusing on climate opportunities, the credibility of using 10-K filings for measurement, and the relevance of the questions we explore regarding climate solutions.

While I truly believe in the value of the paper, I have to outline a multitude of weaknesses. I will go into detail in the firm belief that this could help to improve the paper. I will structure the weaknesses in two major areas: the NLP methodology and the analysis conducted.

On the side of the NLP methodology, I trust that the method can successfully build a proxy for what it is intended to measure. For the core measure, the paper designs a binary classifier and creates a dataset with active learning using ClimateBERT (Webersinke et al., 2022). Then, the

gpt-3.5-turbo-1106 checkpoint is fine-tuned with the data. However, I find some uncertainties and shortcomings in the design and evaluation of the methodology.

First, the labeling process is unclear to me. The paper states that "on average, two researchers annotate the same tasks to obtain some measure of dispersion" (lines 732-733). I can neither find the measure of dispersion, supposedly Cohen's Kappa, nor is it clear, what "on average" means in the context of labeling.

Response:

We agree that the NLP methodology can be further supported with additional documentation and robustness checks. We appreciate the reviewer's constructive suggestions. We have implemented every one of the suggestions, and provide details on each below.

First, we expanded the discussion on the labeling process and added Cohen's Kappa. For convenience, we pasted the relevant discussion from page 72 in Supplementary Note 2 below:

Two researchers annotate the same tasks to obtain some measure of dispersion. In case of a close verdict or a tie between the annotators, the authors of this paper discuss the sentence in depth before reaching an agreement. Out of 3,508 sentences, annotators agreed on 2,905, while the remaining sentences had disagreements. To assess the degree of annotator agreement, we calculate Cohen's Kappa, which is 0.6653 with a 95% confidence interval of 0.64 to 0.6907. This indicates a substantial level of agreement in the labeling process.

2. Second, the paper argues with Stammbach et al., 2023 for the choice of the dataset size. I do not fundamentally doubt the dataset size, but the largest model trained in the Stammbach et al., 2023 paper is RoBERTalarge with 355 million parameters. GPT-3.5-turbo-1106 has an estimated 110-175 billion parameters. If you want to provide evidence about the sufficiency of the dataset size, you should follow the setup in Stammbach et al., 2023 on the performance of a model on the development set as a function of training on different fractions of the training dataset (see Figure 6).

Response:

Thank you for this helpful suggestion, we follow a similar setup as in Fig 6 of Stammbach et al., 2023 to evaluate the performance of the model as a function of training on different fractions of the training dataset. For convenience, we pasted the relevant discussion from page 61 in Supplementary Note 1 below:

Additionally, we evaluate the sufficiency of our training set size by examining how model performance changes as we increase the size of the training dataset. Specifically, we keep a held-out dataset using 20% of the training set, and examine the model performance on this held-out set when we train a GPT-3.5-turbo-1106 model using 0%, 20%, 40%, 60%, and 80% of the training set. Figure A2 shows the largest increase in model performance when the model is fine-tuned with 20% of the training set, compared to the non-fine-tuned

model when 0% training data is provided. This increase reflects the value of fine-tuning the GPT model for the specific task of identifying climate solutions sentences. As the proportion of training set increases from 20% to 80%, we do not observe large improvements in model performance, which provides comfort that our training set is sufficient and that we do not anticipate large improvements in model performance if we were to annotate additional sentences.

Figure A2

3. Third, in my eyes, it is a best practice to disclose and explore all datasets and fine-tuning settings. This means the train-validation test split for the training and evaluation needs to be described. Are the metrics calculated on the validation or test set? Furthermore, for the sake of reproducibility, the main hyperparameters for fine-tuning should be disclosed (at least batch size, learning rate, epochs). Finally, a holistic finetuning setup might profit by simply exploring more models than ClimateBERT. Candidates might be Distil-RoBERTa (Sanh et al., 2020) RoBERTa (Liu et al., 2019), their pre-trained sentence-level versions for the environmental domain (Schimanski et al., 2024), or DeBERTa (He et al., 2021). This is particularly useful for understanding the impact of different sizes (ClimateBERT has 82 million parameters, GPT-3.5-turbo-1106 has 110-175 billion parameters).

Response:

Thank you for these suggestions to enhance the transparency of our methodology. You raise three key points, we discuss each below.

First, we have clarified our train-validation-test split to ensure clarity in how the model is evaluated. We pasted the relevant discussion on page 63.

We employ 5-fold cross-validation to assess our model, optimizing the use of our labeled dataset. This method ensures comprehensive evaluation by partitioning the dataset into five subsets, where each subset serves as a test set while the remaining are used for training, iteratively. For each fold, we designate 20% of the labeled dataset as a holdout set for testing, while the remaining 80% is used to fine-tune a GPT-3.5-turbo-1106 model. The trained model is then evaluated on the held-out 20%, and this process is repeated across all five folds.

Second, we have disclosed the fine-tuning hyperparameters for our GPT-based model, which follow recommended default settings, with epochs set to 3, batch size to 7, and a learning rate multiplier of 2. This sentence is included on page 63. On page 64, we also describe how we choose the optimal parameters for ClimateBERT model: by conducting a grid search over key hyperparameters, including learning rates (5e-05, 2e-05, 1e-05, 5e-06), epsilons (1e-08, 1e-07), and dropout probabilities (0.1, 0.2, 0.3). The optimal model for which our accuracy and F1 scores are based on has a learning rate of 5e-05, epsilon of 1e-08, and dropout of 0.1.

Third, we explored the three alternative models suggested: DistilRoBERTa, RoBERTa, and DeBERTa. Across all models, the accuracy and F1 score are lower than that of ClimateBERT, which also has a performance lower than that of the fine-tuned GPT model we use in the paper. We added this discussion on page 64. We include the accuracy and F1 of the three alternative models below:

DistilRoBERTa: Accuracy: 0.80, F1 Score: 0.76

RoBERTa: Accuracy: 0.38, F1 Score: 0.55

DeBERTa: Accuracy: 0.77, F1 Score: 0.72

4. Fourth, while I like the discussion about the determinism of GPT-3.5-turbo-1106, I do not understand why the model is set to temperature 0.1 and not to temperature 0. This should mitigate the non-deterministic behavior even more than 0.1 (it will still be non-deterministic). While this is an important detail, the evaluation strategy is even more important. The paper states that the "discrepancy observed between any two columns was six rows (0.17% of all rows in the dataset)". Does this mean, the fine-tuned model was evaluated on the fine-tuning data? If this is the case, the results would be obscured by data leakage.

Response:

You raise two important points in this comment. The first relates to the choice of temperature set as 0.1 as opposed to 0. We initially chose 0.1 to reflect a sufficiently low temperature. To address concerns that by not setting a temperature of 0, the model's output is subject to potentially significant variability, we duplicated our experiment of predicting if a sentence is related to a

climate solution with the same model but updated the temperature to 0. When comparing the two model's predictions, the models disagreed on 10 out of 3,508 sentences, accounting for only 0.29% of the data. Furthermore, McNemar's test, a common statistical test for comparing two binary classifiers, was performed to assess the degree to which the two models misclassify the same samples. The test found that the two models do not statistically significantly ($p = 0.34$) differ between their predictions.

The second point is about the potential concern about data leakage when we compare discrepancy on the data used for fine-tuning. We agree that a more representative analysis should use sentences outside of the data used for fine-tuning. Accordingly, in the revised manuscript, we randomly selected 1,000 sentences from outside the training set and apply the fine-tuned GPT model five times. The maximum discrepancy observed between any two columns was 1 row. We provide this discussion on page 65. Thank you for this suggestion that led us to make this change.

5. Fifth, in my understanding, the paper uses the fine-tuned GPT-3.5-turbo-1106 on the topic modeling task. This might even decrease the performance on this task given it was trained on a different classification task. A more comprehensive exploration might help the understanding.

Collectively, I would recommend reworking the NLP methodology more thoroughly.

Response:

We agree with the potential concern that fine-tuning GPT-3.5-turbo-1106 may not necessarily improve performance on the topic modeling task compared to a non-finetuned GPT model. To address this, we conducted additional tests comparing the fine-tuned GPT model with the non-finetuned version using our labeled set of 700 sentences. The results indicate that the fine-tuned model performs better in topic classification. Specifically, the fine-tuned GPT achieved an accuracy of 88%, with only 8 sentences differing across three repeated runs. In contrast, the non-finetuned GPT model achieved a lower accuracy of 74% and produced 75 disagreements across three repeated runs. One possible reason for this difference is that the fine-tuned model has a stronger domain understanding of climate solutions, leading to more consistent classifications (Ouyang et al., 2022). Given these findings, we use the fine-tuned GPT for the topic modeling task and provide a detailed discussion of this decision on page 66.

6. On the side of the analysis, I have the impression that a more solid argumentation and grounding in literature for measure and results is needed. Regarding the measure, a clearer placement in past NLP approaches to quantify and explore business opportunities in climate change is lacking. For instance, Sautner et al., 2023 derives a firm-level climate change opportunity exposure measure. This article is cited but only as a risk measure. This measure is also applied by other work (Ma et al., 2023). Toetzke et al., 2024 are also quantifying climate technology innovations with NLP – thereby delivering closely related research. Besides, there are approaches beyond

NLP to assess business opportunities of climate change (Li et al., 2024). Reflecting on these past projects could help to refine the scope and novel contribution of the present work.

Response:

We agree that a clearer placement in relation to other measures on climate opportunities helps strengthen the contribution of our measure. We added a paragraph in the Discussion section that highlights the strength of our measure relative to other measures on climate opportunities. We also added a new analysis where we include both the climate exposure and climate opportunity measure from Sautner et al., 2023 as additional independent variables in the revenue growth analysis, and find that our Climate Solutions Measure remains significantly positively associated revenue growth, whereas we do not observe significant associations between the measures from Sautner et al., 2023 and revenue growth (Supplementary Table 6 Panel E).

For convenience, we pasted the relevant section from page 15 in the Discussion section below:

Relative to other measures on climate opportunities derived from voluntary disclosures, such as earnings conference calls and CDP reports, (Sautner et al., 2023; Li et al., 2024; Toetzke et al., 2024), our measure differs by focusing on products and services that are disclosed in regulated financial filings, with greater credibility, lower noise, and less subject to concerns about the self-selection of disclosures. Additionally, our measure is more likely to capture climate solutions already integrated into a firm's existing product portfolio, rather than input-based proxies like R&D expenses or labor skills, which reflect investment efforts rather than commercialized solutions. Consistent with this expectation, our Climate Solutions Measure is positively associated with revenue growth. In contrast, climate exposure and opportunity measures derived from earnings conference calls are more reflective of market attention and not associated with revenue growth when we include them in the same regression with Climate Solutions Measure (Sautner et al., 2023) (Supplementary Table 6 Panel E).

7. Another factor that might need more justification is possible concerns about cheap talk or greenwashing. Mentioning these green technologies might entail reputational, legal, or financial advantages for companies. Yet, we know that communication can be deceiving – subject to cheap talk or greenwashing (Bingler et al., 2024). Can you make a convincing argument that the measure is not susceptible to these patterns?

Footnote 1: See for instance in the rapidly growing green fund landscape until 2022: <https://www.morningstar.com/sustainable-investing/2022-us-sustainable-funds-landscape-5-charts>.

Response:

In the revised paper, we added a new section providing three sets of validation analyses to mitigate concerns about greenwashing or cheap talk in our Climate Solutions Measure. For convenience,

we pasted the relevant sections from pages 5-7 in the Results section below. We also provide more details about the methodology, including variable construction, regression specifications, in the Methods section.

The validation analysis further confirms that the Climate Solutions Measure derived from 10-Ks reflects companies' business portfolio involvement in climate solutions. The Climate Solutions Measure is positively associated with the percentage of green patents and green revenues estimated by MSCI, one of the largest data providers, for the most recent fiscal year with available data (Fig. 2a). A one standard deviation increase in Climate Solutions Measure is associated with a 0.47 and 0.54 standard deviation increase in green revenue and green patent percent, respectively. The Climate Solutions Measure, green revenues, and green patents measures increase monotonically across groups of Climate Solutions Measures (Fig. 2b). For example, firms with Climate Solutions Measure greater than 0 but less than 1 (Group 1), and greater than 1 but less than 5 (Group 2), have on average green revenue percent of 2 and 6, respectively. This increases to approximately 13 percent for firms with Climate Solutions Measure greater than 5 but less than 10 (Group 3). Firms with Climate Solutions Measure greater than 10 (Group 4) have green revenue percent of more than 35. Green patents increase from approximately 5 percent for Group 1 to approximately 46 percent for Group 4.

As a second validation test, we observe that a higher Climate Solutions Measure is associated with higher innovation across four different measures. With industry-year fixed effects, a one standard deviation higher Climate Solutions Measure is associated with a 3% higher research and development expense scaled by revenue, 9% higher knowledge capital scaled by revenue, 6% higher research and development capitalized (RDC) scaled by revenue, and 10% higher trade secret scaled by revenue (Fig. 2c). In the absence of data on climate investments, this positive association provides support that the Climate Solutions Measure reflects firms engaging in more innovation.

In a third validation test, we find that the Climate Solutions Measure varies predictably in response to major events. Most notably, we see the largest jump in the most recent years, which represents the information in fiscal years 2021 and 2022 disclosed in 2022 and 2023, respectively. The Inflation Reduction Act (IRA), passed in 2022, provided hundreds of billions of dollars in incentives for the development and acceleration of climate solutions. In industries with more IRA opportunities, the Climate Solutions Measure on topics covered by the IRA increases significantly in fiscal years 2021 and 2022, relative to prior years (Fig. 2d). We also observe an increase between 2005 and 2010, during a period where over 20 states in the US passed the Renewable Portfolio Standards (RPS) that mandate a minimum ratio of renewable energy supply in a state. The Climate Solutions Measure on topics related to renewable energy increases significantly after the staggered adoption of RPS for firms in industries relevant to renewable energy (Fig. 2d).

8. I find a similar pattern is visible in the results. A more solid argumentation and grounding in literature might be helpful. For instance, when assessing climate risk, carbon emissions are used. However, it is possible that your subset of highly innovative companies shows different emission patterns as shown by prior research (Cohen et al., 2020). If successfully innovating, a company's direct emissions might even rise due to more production. How does this play out as more climate risk? As you mentioned, the climate risk literature is more advanced, and other, clearer metrics might be more suitable. Another example of unclearities is your assumption that companies with higher climate solutions measure should demonstrate greater revenue growth. Why is this? While I would love to believe it is true, there is no backing for this claim. Notably, financing green projects is often rather associated with fewer returns in financial markets (Baker et al., 2022). Also, there is a potential for reverse causality at play. It might be that companies with high revenue growth can afford to invest in green technologies, rather than the revenue growth stemming from these technologies. More precise argumentation and grounding in literature could also help your regression analyses. For instance, it is unclear to me why the first regression runs without firm size when it might play a role in sustainability efforts and communication (Drempetic et al., 2019; Bissoondoyal-Bheenick et al., 2023). It is important to state that my concern is not that all of the above assumptions are wrong. Rather, I view that the current state of the paper lacks justifying and contextualizing its setup and results.

Response:

Your comment raised important considerations to help enhance our Results section. Below, we highlight three main changes we made in response to your comment.

First, we structured the results section to better motivate our analyses by referencing related literature. Specifically, after introducing the Climate Solutions Measure, providing validation, and describing topic characteristics, we focus on three empirical applications of the data. For each application, we start the section by motivating the importance of the question based on existing literature, and then present results. To give an example, in the next point, we describe how we revised the section related to revenue growth.

Second, we revised the revenue growth analyses to add more solid argumentation and conduct robustness tests to mitigate empirical concerns. We agree with your comment that ex-ante, it is not clear that companies with higher climate solutions measure should demonstrate greater revenue growth. As such, we examine the relationship between climate solutions and revenue growth as an empirical question, as motivated below on page 10:

If the climate business opportunities communicated in the Business Description section of the 10-K indicate a company's capacity to commercialize climate solutions, and that climate solutions represent a growing market under the rise of climate transition risks, then firms with higher Climate Solutions Measure might demonstrate greater revenue growth. At the same time, it is not clear if firms with higher climate solutions will exhibit higher revenue growth because there is high uncertainty regarding the demand for climate solutions given uncertainties in climate regulations, market preference, and technology

developments (Blyth et al., 2007). As such, we use the Climate Solutions Measure to examine if firms with climate solutions experience higher revenue growth. We also examine how this relationship varies with prevalence of patent protection across industries and across climate solutions with different characteristics.

Additionally, we added three sets of robustness tests for this analysis to mitigate concerns with reverse causality or confounding variables. For convenience, we pasted the relevant sections from page 25 in the Methods section below.

We conduct three robustness tests to mitigate potential concerns related to the revenue growth results. First, to mitigate concerns about reverse causality, we replace revenue growth with the one-year forward revenue growth, and three-year moving average revenue growth from $t=0$ to $t=2$, and our results remain robust. Second, we include firm fixed effects to control for time-invariant firm characteristics, and the results remain robust when we use the three-year moving average revenue growth. We focus on the three-year moving average because revenue growth tends to be slow-moving, and using firm fixed effects with one-year revenue growth may absorb too much of the variation, limiting our ability to detect meaningful effects. Third, to ensure comparability between firms with and without climate solutions, we apply entropy balancing on the first three moments of firm size and firm age (Hainmueller, 2012). As entropy balancing requires discrete treatment variables, we classify firms as with climate solutions if *Climate Solutions Measure* is above 1% or 5%, which also helps mitigate concern about the right skew of the main *Climate Solutions Measure*. Our results remain robust when we repeat the main specifications with these alternative measures for climate solutions, and when using entropy balanced samples.

Third, regarding the relation between climate solutions and greenhouse gas emissions, we also present the analysis being more agnostic as to the prediction. We incorporated your argument that prior literature find that some high-emitting industries have more climate innovation (Cohen et al., 2020), and that a firm's direct emissions might even increase if climate solutions lead to rise in production. As such, we expanded our greenhouse gas measure from scope 1 to include all three scopes of emissions, for both intensity and absolute measures. Across all specifications, we find a weak relationship between climate solutions and greenhouse gas emissions, and use this finding to highlight that our measure captures a distinct group of firms that climate risks measures cannot identify. The relevant discussions are included on page 8 in the Results section.

9. I genuinely like the idea of the paper. The approach seems promising and effective. However, when assessing the details of the paper, it comes down to two questions. First: Do I believe in your measure? Partially to yes. The creation and evaluation of the model and measure are not yet well aligned with best practices. Second: Do I believe in your results? Partially, leaning towards no. I think, more thorough argumentation is needed, better grounding in literature, and clearer control for confounders.

Response:

Thank you again for your time and effort in providing these insightful comments and suggestions. We hope the elaborated description and additional robustness tests on the LLM provide more confidence in our measurement. Together with the other reviewers' comments, we made significant edits to enhance the results section to provide more thorough argumentation, more reference to the literature, and additional robustness tests to address concerns about confounders. We hope that laying out our updates and thought process above has helped speak to your concerns.

Reference:

Blyth, W. et al. Investment risks under uncertain climate change policy. *Energy Policy* 35, 5766–5773 (2007).

Hainmueller, J. Entropy balancing for causal effects: A multivariate reweighting method to produce balanced samples in observational studies. *Political analysis* 20, 25–46 (2012).

Li, X., Luo, L. & Tang, Q. Climate risk and opportunity exposure and firm value: An international investigation. *Business Strategy and the Environment* (2024).

Ouyang, L., Wu, J., Jiang, X., Almeida, D., Wainwright, C., Mishkin, P., ... & Lowe, R. (2022). Training language models to follow instructions with human feedback. *Advances in neural information processing systems*, 35, 27730-27744.

Sautner, Z., Van Lent, L., Vilkov, G. & Zhang, R. Firm-level climate change exposure. *The Journal of Finance* 78, 1449–1498 (2023).

Toetzke, M., Probst, B., Feuerriegel, S., Anadon, L. D. & Hoffmann, V. H. Machine learning can help track climate technology innovation. Available at SSRN 4810933 (2024).

Response to Reviewers 1 and 2 (Manuscript NCOMMS-24-53291A)

First, we would like to thank you for your insightful and constructive comments. They have been very helpful in revising and improving this version of the paper. We appreciate the opportunity to resubmit and have made a sincere effort to address all of your comments.

In this response, we start by highlighting the major changes to the paper since the previous submission. We then provide a detailed response addressing each of your comments, where your comments are reproduced in boxed *italics*, followed by our response in a regular typeface.

Main changes to the manuscript:

1. Strengthened conceptual framing to further enhance the value of the Climate Solutions Measure:
 - a. Expanded and clarified the conceptual frameworks by linking to more relevant literature, including the distinction between climate risks and opportunities, and the connection between climate solutions and revenue growth.
 - b. Introduced a new analysis in Application Example 3 that demonstrates higher stock return synchronicity for industries with more similar climate solutions topics, deriving further insights into economic fundamentals from Application Example 3.
 - c. Revised writing throughout the paper to more clearly acknowledge the descriptive nature of the analysis, the limitations of certain proxies (e.g., in Application Example 2), and the interpretation of weak correlations (e.g., between climate risks and opportunities), and to suggest these for future research.
2. Improved empirical interpretation and robustness:
 - a. Enhanced the revenue growth analysis by expanding control variables, benchmarking explanatory power against prior studies, and applying Oster (2019) to assess robustness to unobserved confounders.
 - b. Improved treatment of missing R&D data through a combination of indicator variables, robustness checks excluding the standard imputation process used by the literature, and industry-level restrictions.
 - c. Clarified interpretive language to ensure claims are consistent with the descriptive nature of the text-based measure and do not overstate implications.
3. Strengthened large language model (LLM) methodology discussion:
 - a. Clarified evaluation methodology and reporting, including specifying the use of five-fold cross-validation with reporting performance metrics for the average across the five folds.
 - b. Enhanced benchmark model analysis by adding EnvironmentalBERT, improving RoBERTa, and reporting performance of all models across industries in the manuscript.
 - c. Conducted sensitivity analyses to assess the effect of nondeterminism due to temperature settings and added random noise to Climate Solutions Measure to confirm robustness of downstream results.

Your specific comments:

1. Thank you for your revised manuscript and the detailed point-by-point response. I acknowledge the substantial revisions made and appreciate your efforts to improve the structure, strengthen the empirical foundation, and clarify several claims. The newly introduced analyses and expanded validation efforts mark a significant improvement. However, some concerns remain unresolved or only partially addressed. Below, I outline the areas where I believe the manuscript needs further refinement.

Response:

Thank you for your thoughtful feedback and for acknowledging the improvements in our revised manuscript. We also value your continued engagement and have carefully addressed the remaining concerns with further clarification and revisions, as detailed in our responses below.

2. Conceptual framing and theoretical foundation: The manuscript still does not sufficiently explain why measuring climate opportunities is so important/relevant. I agree that it is; I suggest to be more explicit. You emphasize the relevance of the “opportunity” perspective by demonstrating a lack of correlation between climate risk and opportunity measures. While the absence of correlation is an interesting result, it does not by itself demonstrate that risk and opportunity represent conceptually distinct dimensions. I recommend either supporting this point with theoretical and empirical arguments or presenting it more cautiously as a suggestion for future research. In addition, the theoretical basis for some key claims – such as the link between the climate solutions measure and revenue growth – should be more clearly grounded in relevant literature.

Response:

We thank the reviewers for highlighting the importance of strengthening the conceptual framing and theoretical foundation of the manuscript. We agree that the distinction between climate risks and opportunities—and the motivation for measuring climate opportunities—should be made more explicit. In response, we have made three main revisions:

First, we revised the beginning of the Introduction to more clearly articulate the conceptual distinction between climate risks and opportunities. We now include a dedicated paragraph outlining the differences in financial characteristics, policy implications, and industry focus. This paragraph draws on relevant economic literature to support the distinction, including the role of emissions externalities versus knowledge spillovers and their implications for policy design. For convenience, we pasted this paragraph (paragraph 2 of introduction) below:

Climate risks and climate opportunities represent distinct dimensions of the low-carbon transition, with different financial characteristics, policy implications, and industry focus. Firms exposed to climate risk, such as those in high-emitting industries, have incentives to reduce emissions and often incur additional costs to do so (Gillingham and Stock, 2018). Carbon pricing policies provide incentives for these firms to internalize the social cost of

emissions. In contrast, firms pursuing climate opportunities generate value by meeting growing demand for decarbonization technologies and services, creating revenue growth potential (Kogan et al., 2017). These firms, such as battery producers, need not have high emissions themselves. Unlike emissions externalities, which are addressed through carbon pricing regulations, other policies, such as innovation subsidies, help mitigate the underinvestment in climate innovations subject to knowledge spillovers that benefit other firms (Acemoglu et al., 2012, 2016). Recognizing this, the Sixth Assessment Report of the IPCC emphasizes the importance of developing and deploying climate solutions and calls for a systemic view of climate innovation to guide effective policy design (Blanco et al., 2022).

Second, following your suggestion, we now present the empirical finding of a weak correlation between risk and opportunity with more caution. Rather than interpreting it as evidence that the two are conceptually distinct, we frame it as suggestive and highlight it as a potential direction for future research. Also, we added a new supporting table showing the association between Climate Solutions Measure and commercial sustainability corporate ratings, which reflect a firm's carbon risk management. The magnitude of the relationship is also moderate. These updates are reflected on page 8, and we pasted this paragraph below for convenience.

As an additional supporting measure, we examine a firm's climate risk management using emissions scores from Refinitiv and MSCI in Supplementary Table 5 Panel E. Consistent with the greenhouse gas results, we find that the relationship between emissions scores and Climate Solutions Measure is modest (less than 0.15 standard deviations). Taken together, these empirical observations suggest that climate risks and opportunities are weakly related. Future research can further examine this potential distinction and explore the implications of different firms being exposed to climate risk versus innovating to capture climate opportunities.

Third, we have strengthened the theoretical link between our Climate Solutions Measure and revenue growth. In the revised manuscript (see pages 10-11), we clarify how firms capturing climate opportunities may benefit from growing demand for decarbonization technologies and reference relevant literature on innovation and firm growth to support this relationship (Eberhart et al., 2004; Kogan et al., 2017).

3. Interpretation of statistical results and methodological concerns: Your efforts to avoid causal language and to add robustness checks (e.g., entropy balancing, forward-looking revenue growth, firm fixed effects) are excellent. Nevertheless, there are still some areas that could benefit from further clarification or development: (1) The manuscript still does not sufficiently address the possibility of confounding variables. The set of control variables is narrow which limits confidence in the claim that climate solutions and revenue growth are truly correlated. (2) Adjusted R^2 values remain low. While similar values have been reported in the literature, and it is understandable that explanatory power may be limited when introducing a novel metric, the manuscript should nonetheless acknowledge this and discuss potential reasons for the low R^2 values. (3) The decision

to impute missing R&D data with zeros raises serious concerns, as it likely distorts the distribution and introduces systematic bias. While alternative proxies have been added, this does not resolve the methodological concern unless more advanced imputation or exclusion strategies are employed. (4) A noticeable gap remains between statistical results and their interpretation. Some conclusions go beyond what is supported by the underlying data. For instance, statements such as “products or services that enable the economy to decarbonize by 2022” (e.g., lines 310f and 346ff) overstate the implications of the text analysis. Such language suggests realized macroeconomic impact, whereas the method merely identifies the presence of climate-related terminology in financial filings. I strongly recommend revising such formulations to better reflect the scope and limitations of the approach.

Response:

We thank the reviewers for the thoughtful and constructive feedback. The points raised have helped us clarify and improve several aspects of our empirical analysis and interpretation. We address each concern in turn below.

(1) & (2) Confounding Variables and Low R-squared Values:

We acknowledge the concern regarding potential omitted variable bias and the relatively low explanatory power in our revenue growth analysis. To address these points, we made three key improvements.

First, we expanded the set of control variables in the revenue growth regressions (including robustness tests). In addition to firm size and age, we now control for profitability (ROA), leverage (debt-to-equity ratio), and capital intensity (capital expenditures over assets)—firm characteristics commonly associated with both investment capacity and revenue growth.

Second, we explicitly acknowledge and discuss the relatively low R-squared values in the revised manuscript. We note that our R-squared values are in line with prior literature and likely reflect the wide range of factors that influence revenue growth, many of which may be unobservable or difficult to measure.

Third, to evaluate the robustness of our results to potential unobserved confounding, we apply the method proposed by Oster (2019). This approach provides a formal test for how much influence unobservable factors would need to exert to fully explain away the observed relationship. Our results suggest that unobservables would need to be very substantially more influential than observables to eliminate the effect, mitigating concerns about omitted variable bias.

For convenience, we paste the newly added paragraph relating to the latter two points from pages 27-28 (methodology) of the manuscript below for reference:

Finally, we acknowledge that the R-squared is relatively low in this analysis, which likely reflects the wide range of factors that influence revenue growth. We benchmark our R-squared against prior studies and find that our explanatory power falls within a reasonable range of the literature (R-squared of around 5%-15%) (Thornhill, 2006; Albring et al., 2013). To further assess potential concerns related to confounding variables, we follow

Oster (2019) to examine coefficient stability in the presence of unobservable factors. Specifically, the method compares the relative movements of coefficients and R-squared values from regressions with and without control variables to estimate δ , which captures the proportion of unobservable factors relative to observable factors that will produce a treatment effect of zero. Using an $R\text{-squared}_{\max}$ set to 1.3 times the R-squared in Table 6, Panel A, Column 2, as recommended by Oster (2019), our estimated δ is 5. A δ of 5 suggests that unobservable factors would need to be five times as influential as observable ones to explain away the estimated effect, providing some comfort over confounding concerns.

(3) Treatment of Missing R&D Data:

We agree with the concerns regarding the missing R&D values. In our main specification, we now follow prior literature by replacing missing R&D values with zero, and also include a dummy variable for firms with missing R&D (Koh and Reeb, 2015). To further mitigate this concern, we incorporated two additional robustness tests. First, we limit the analysis to industries where the average rate of missing R&D data is below the median, focusing on sectors where reporting is more reliable. Second, we estimate a version of the model where missing values are left unfilled rather than imputed as zero. The results remain consistent across both approaches, helping alleviate concerns that our findings are driven by how missing R&D data is handled. Additionally, we have three additional innovation measures, which mitigate concern about reliance on R&D data. This discussion is provided on page 21-22 (methodology).

(4) Interpretation of Statistical Results:

We carefully reviewed the manuscript to ensure that all interpretations are well-supported by the underlying data. In particular, we revised or removed language that could be interpreted as overstating the implications of our text-based measure. For example, we replaced the quoted phrase with the more precise wording “companies whose filings reference climate-solution products or services” (page 16). Additionally, we expanded the limitations paragraph (pages 18-19) to include the following clarification: “However, we caution the reader that, in the absence of a natural experiment with randomized treatment of climate solution products and services, we are unable to establish causality.”

4. Application examples: The newly added use cases (Application Examples 1 and 2) introduce valuable perspectives and strengthen the applied relevance of the introduced metric. However, due to methodological shortcomings, the reliability of the results remains limited.

In Application Example 2, political affiliation is proxied using presidential election results from 2020 (and 2016 as a robustness check), which raises questions given that the data covers the period 2005-2022 and climate policy in the U.S. is often shaped at the state – rather than federal – level. Moreover, many of the firms included in the analysis are likely to operate across multiple countries and, therefore, not solely influenced by U.S. state-level political dynamics. These limitations affect the interpretability of the findings and should be clearly acknowledged.

Additionally, the observed differences are relatively small – especially considering that the outcome variable reflects only the proportion of climate-related language in financial filings. Considering the limited magnitude of the observed differences, a more cautious interpretation would be appropriate. Application Example 3, by contrast, still contributes little beyond confirming well-known regulatory and market trends. While the topic has the potential to yield meaningful insights if analyzed in more depth, in its current form, this example does not add substantive value to the manuscript.

Response:

We thank the reviewers for acknowledging that Application Examples 1 and 2 strengthen the applied relevance of the paper. In response to the concerns raised, we have revised the writing to more clearly acknowledge the methodological limitations of Application Example 2, and we have incorporated new analysis to strengthen the contribution of Application Example 3. We provide further details on these revisions below.

Application Example 2

We agree with the interpretive limitations of this application. In response, we have added the following paragraph on page 13 to explicitly acknowledge the key caveats regarding our political affiliation proxy, the cross-jurisdictional nature of firm operations, and the modest magnitude of observed effects:

Taken together, the political affiliation analysis highlights descriptive patterns that should be interpreted with caution. Our proxy for political affiliation is based on presidential vote share, but we acknowledge that climate policy in the United States is often shaped at the state level, and that many firms operate across multiple countries and may be influenced by international policy environments. Additionally, while the observed differences in the Climate Solutions Measure are statistically significant in certain comparisons, they are modest in magnitude. These caveats underscore the need for future research using more granular, multi-level policy data to improve inference.

Application Example 3

We appreciate the reviewers' feedback on Application Example 3. In response, we have added a new analysis examining how the convergence of climate solutions across industries is reflected in financial market behavior, specifically through stock return synchronicity. This extension helps to move beyond descriptive patterns and provides a complementary perspective on the economic relevance of industry convergence around climate solutions. We believe this analysis strengthens the contribution of Application Example 3 by linking firms' climate solutions disclosures to market-based outcomes. The newly added paragraph on page 15 is included below:

To examine whether this convergence in climate solution topics across industries is reflected in financial market behavior, we analyze stock return synchronicity between industry group pairs. Stock return synchronicity quantifies the extent to which stock returns across two industries move together, with higher values indicating more similar underlying

economic fundamentals (Roll, 1988; Morck et al., 2000; Chan and Hameed, 2006). Using our climate solutions topics, we compute cosine similarity scores between industry pairs each year based on the distribution of topics firms disclose. We find that industry pairs with higher topic similarity are associated with greater co-movement in their stock returns (Fig. 8c). A one standard deviation increase in climate solution similarity corresponds to a 0.15 standard deviation increase in stock return synchronicity. While the magnitude is modest, this finding suggests that as firms across industries disclose similar types of climate solutions in financial reports, their economic fundamentals, and thus their stock market performance, become more aligned.

5. Overall, the manuscript shows great progress. The core idea – developing a quantitative measure for climate-related business opportunities based on AI analysis of financial disclosures – is timely and relevant. However, I still see some shortcomings. With further work on refining the empirical models and strengthening the theoretical underpinnings of key relationships, I believe this paper has strong potential.

I hope these comments support the further development of your work.

Response:

We sincerely thank the reviewers for the encouraging feedback and for recognizing the relevance and potential of the core idea. We also greatly appreciate your time and effort in providing these insightful comments and suggestions. We have updated the manuscript based on the totality of comments from you, the Editor, and other reviewers, and we believe the manuscript is much improved thanks to the constructive feedback.

Reference:

Acemoglu, D., Aghion, P., Bursztyn, L., & Hemous, D. (2012). The environment and directed technical change. *American economic review*, 102(1), 131-166.

Acemoglu, D., Akcigit, U., Hanley, D., & Kerr, W. (2016). Transition to clean technology. *Journal of political economy*, 124(1), 52-104.

Albring, S. M., Huang, S. X., Pereira, R., & Xu, X. (2013). The effects of accounting restatements on firm growth. *Journal of Accounting and Public Policy*, 32(5), 357-376.

Blanco, G., de Coninck, H. C., Agbemabiese, L., Diagne, E. H. M., Anadon, L. D., Lim, Y. S., ... & Winkler, H. (2022). Innovation, technology development and transfer. In *IPCC, 2022: Climate change 2022: Mitigation of climate change. Contribution of working group III to the sixth assessment report of the intergovernmental panel on climate change* (pp. 2674-2814). Cambridge University Press.

- Chan, K., & Hameed, A. (2006). Stock price synchronicity and analyst coverage in emerging markets. *Journal of Financial Economics*, 80(1), 115-147.
- Eberhart, A. C., Maxwell, W. F., & Siddique, A. R. (2004). An examination of long-term abnormal stock returns and operating performance following R&D increases. *The journal of finance*, 59(2), 623-650.
- Gillingham, K., & Stock, J. H. (2018). The cost of reducing greenhouse gas emissions. *Journal of Economic Perspectives*, 32(4), 53-72.
- Kogan, L., Papanikolaou, D., Seru, A., & Stoffman, N. (2017). Technological innovation, resource allocation, and growth. *The quarterly journal of economics*, 132(2), 665-712.
- Koh, P. S., & Reeb, D. M. (2015). Missing r&d. *Journal of Accounting and Economics*, 60(1), 73-94.
- Morck, R., Yeung, B., & Yu, W. (2000). The information content of stock markets: why do emerging markets have synchronous stock price movements?. *Journal of financial economics*, 58(1-2), 215-260.
- Oster, E. (2019). Unobservable selection and coefficient stability: Theory and evidence. *Journal of Business & Economic Statistics*, 37(2), 187-204.
- Roll, R. (1988). R^2 . *The Journal of Finance*, 43(3), 541–566. <https://doi.org/10.2307/2328183>
- Thornhill, S. (2006). Knowledge, innovation and firm performance in high-and low-technology regimes. *Journal of business venturing*, 21(5), 687-703.

Response to Reviewer 3 (Manuscript NCOMMS-24-53291A)

First, we would like to thank you for your insightful and constructive comments. They have been very helpful in revising and improving this version of the paper. We appreciate the opportunity to resubmit and have made a sincere effort to address all of your comments.

In this response, we start by highlighting the major changes to the paper since the previous submission. We then provide a detailed response addressing each of your comments, where your comments are reproduced in boxed *italics*, followed by our response in a regular typeface.

Main changes to the manuscript:

1. Strengthened conceptual framing to further enhance the value of the Climate Solutions Measure:
 - a. Expanded and clarified the conceptual frameworks by linking to more relevant literature, including the distinction between climate risks and opportunities, and the connection between climate solutions and revenue growth.
 - b. Introduced a new analysis in Application Example 3 that demonstrates higher stock return synchronicity for industries with more similar climate solutions topics, deriving further insights into economic fundamentals from Application Example 3.
 - c. Revised writing throughout the paper to more clearly acknowledge the descriptive nature of the analysis, the limitations of certain proxies (e.g., in Application Example 2), and the interpretation of weak correlations (e.g., between climate risks and opportunities), and to suggest these for future research.
2. Improved empirical interpretation and robustness:
 - a. Enhanced the revenue growth analysis by expanding control variables, benchmarking explanatory power against prior studies, and applying Oster (2019) to assess robustness to unobserved confounders.
 - b. Improved treatment of missing R&D data through a combination of indicator variables, robustness checks excluding the standard imputation process used by the literature, and industry-level restrictions.
 - c. Clarified interpretive language to ensure claims are consistent with the descriptive nature of the text-based measure and do not overstate implications.
3. Strengthened large language model (LLM) methodology discussion:
 - a. Clarified evaluation methodology and reporting, including specifying the use of five-fold cross-validation with reporting performance metrics for the average across the five folds.
 - b. Enhanced benchmark model analysis by adding EnvironmentalBERT, improving RoBERTa, and reporting performance of all models across industries in the manuscript.
 - c. Conducted sensitivity analyses to assess the effect of nondeterminism due to temperature settings and added random noise to Climate Solutions Measure to confirm robustness of downstream results.

Your specific comments:

1. Overview of the Adjustments

The authors make a significant effort to improve the quality of the paper by including more literature, reinforcing key results with new experiments, and trying to advance the LLM methodology.

Response:

Thank you for your thoughtful feedback and for acknowledging the improvements in our revised manuscript. We also value your continued engagement and have carefully addressed the remaining concerns with further clarification and revisions, as detailed in our responses below.

2. Analysis

However, there seems to be a persisting misconduct in training and using the models. Since this is the major building block of the paper for constructing their main contribution, the Climate Solution Measure, it is important that the construction is clear and flawless. Yet again, I have to outline several shortcomings. First, the authors seem to use two different methods in evaluating the main model, GPT-3.5-turbo-1106. On the one hand, they report a performance score on a single held-out test set. On the other hand, they report using a five-fold cross-validation. Which score is finally reported? And which score is used to compare to other models? If you are using the best fold, then the comparison is not valid.

Response:

Thank you for this helpful comment and the opportunity to clarify our evaluation procedure. We confirm that all reported performance scores for GPT-3.5-turbo-1106—including both the accuracy (84%) and F1 score (79%)—are based on the average across five iterations from our five-fold cross-validation procedure. We do not report results from the best-performing fold, and we do not use a separate single held-out test set outside of the cross-validation. We have further updated the manuscript to clarify these points. We pasted the relevant paragraphs from page 74 below.

We employ five-fold cross-validation to assess our model, optimizing the use of our labeled dataset. This method ensures comprehensive evaluation by partitioning the dataset into five parts (folds) and running five iterations. In each iteration, four folds (80%) are used to fine-tune the GPT-3.5-turbo-1106 model, and the remaining fold (20%) is used as a validation set. We report model performance as the average across the five iterations. When presenting results by industry, we present average performance metrics using only the iteration in which each sentence appears in the validation set.

The model demonstrates an average accuracy of 84.09% across five folds, with a standard deviation of 1.93% between folds, indicating consistency in performance across different subsets. Moreover, we report an average F1 score of 79.50% across five folds with a

standard deviation of 2.32% between the folds. The F1 score, being the harmonic mean of precision (the percentage of predicted positives that are truly positive) and recall (the percentage of true positives that are predicted as positives), provides a balanced measure of the model's accuracy, particularly valuable in the context of binary classification. It is especially pertinent for evaluating performance in imbalanced datasets, where traditional accuracy metrics may not fully capture the effectiveness of the model in distinguishing between the binary classification.

This comment prompted us to carefully review the clarity of our methodology section, including soliciting feedback from computer scientists familiar with NLP model evaluation. Through this process, we identified two areas in the writing that may have caused confusion: First, we now ensure that the analyses of BERT-based models (i.e., ClimateBERT, EnvironmentalBERT, RoBERTa, DistilRoBERTa, and DeBERTa) are all also evaluated using five-fold cross-validation, and we now state this explicitly in the manuscript. On page 76, we clarify that the reported accuracy and F1 scores for all benchmark models reflect the average performance across five folds, consistent with the GPT-based model.

Second, for the industry-level performance breakdown, we report F1 and accuracy scores by industry using a pooled set of model predictions. Specifically, we concatenate the validation set predictions from all five iterations, such that each of the 3,508 labeled sentences appears once—when it was in the validation set for its respective fold. We then show the average industry-level F1 and accuracy scores based on this complete set of predictions. We explicitly stated this on page 74 by adding this sentence: “When presenting results by industry, we present average performance metrics using only the iteration in which each sentence appears in the validation set.”

We believe these clarifications improve the transparency and clarity of our methodology, and thank you again for raising this important point.

Second, when comparing to the BERT-based models, there seems to be an error in the training process. The extremely low results for RoBERTa (accuracy: 0.38) are extremely uncommon. It is most likely that something in the training code is wrong. This is a problem because the authors likely used the same code for the other BERT-based models. Thus, this makes me question the other results as well. Besides, despite stating that the authors compare to all reasonable models, they leave out the EnvironmentalBERT models (Schimanski et al., 2024) from the last iteration of reviews. In line with ClimateBERT, these should perform the best on this environmental/climate task and therefore are the best comparison points. Is there any reason to leave them out? While this comparison itself may be minor, the observations made here make me question the rigor applied to get the model usage completely right.

Response:

Thank you for raising these important issues. We have carefully re-examined our approach, updated our analysis, and provided more detailed documentation in the paper.

First, regarding the RoBERTa model, we acknowledge that the previously reported accuracy of 0.38 resulted from a training difficulty. Specifically, when we run RoBERTa initially, the best model predicts all sentences as 0, resulting in a F1 score of 0. As such, we stopped at an epoch before it classifies all as zero, resulting in a lower accuracy rate of 0.38. Upon investigation, we found that the RoBERTa model, when trained using the learning rate of 5e-5, exhibited a tendency to converge prematurely and predict all sentences as class 0. To address this issue, we performed a basic grid search on learning rate and modified the learning rate by lowering it by a factor of 10 to 5e-6, allowing the model to train more gradually and avoid collapsing into a single-class prediction. With this adjustment, RoBERTa now yields an accuracy of 0.818 and an F1 score of 0.775, which are more in line with expectations and other benchmarks.

Second, we apologize for omitting the EnvironmentalBERT model in our earlier response. At the time, we thought of ClimateBERT as the topic-specific model, and interpreted the suggestion as benchmarking against a few standard BERT-based models, hence we included the non-topic specific models, RoBERTa, DistilRoBERTa, and DeBERTa. We agree that EnvironmentalBERT is highly relevant and have now included it in our updated comparisons. To improve transparency and allow for clearer evaluation, we now report the performance (F1 and accuracy) of all comparison models by industry group in Supplementary Table A3. Across these models, the F1 and accuracy rates are below the fine-tuned GPT we use. In particular, the fine-tuned GPT-3.5 outperforms the other models more in correctly identifying climate solutions sentences in industries with fewer climate solutions, such as in Equity Real Estate Investment Trusts and Household and Personal Products. We provide this discussion on page 76.

We hope these updates help address the concerns regarding our model evaluation, and we appreciate the opportunity to strengthen the manuscript through these improvements.

Third, and along those same lines, changing the temperature to 0 for GPT-3.5-turbo-1106 represents a clear change in the model setup. The authors argue that "models disagreed on 10 out of 3,508 sentences, accounting for only 0.29% of the data". This means that, again, these results are obscured by data leakage. You cannot compare models trained on a dataset with the very dataset. The models literally know the answer through training. I want to outline that this is extremely important. The change in temperature to 0 means that the Climate Solution Measure will likely change, and with that, potentially the downstream results of this study. I urge the authors to be very careful when revising the construction of their model and measures. It is likely needed to reestimate the entire Climate Solution Metric.

Response:

Thank you for raising this important point. We agree that any change in model setup—particularly the temperature parameter—should be carefully considered due to its potential implications for model outputs and downstream results. We took this concern seriously and conducted several additional analyses to assess the potential impact of nondeterminism between temperature settings of 0 and 0.1. Overall, our findings suggest that nondeterminism is minimal and that results are highly comparable across both settings. Below, we describe our assessment and explain our

rationale for re-running the Climate Solutions Measure for all sentences from the most recent fiscal year end filings in our data.

First, we re-ran the fine-tuned GPT-3.5-turbo-1106 model at temperature 0 on all fiscal year 2022 sentences that are not in the 3,508-sentence training set. Out of approximately 500,000 sentences, only 299 (0.06%) produced different outputs compared to the version run with temperature 0.1. When aggregated to the firm-year level, Climate Solutions Measure based on a temperature of 0 has a **99.97%** correlation with the existing version with a temperature of 0.1. In other words, there are minimal differences between the two measures. This low discrepancy provides reassurance that our results are likely robust to this parameter choice, which we also examine (see below).

Next, we conducted a more thorough assessment for nondeterminism at both temperature settings. We randomly pick 10,000 sentences from fiscal year 2022, and repeat the classification process 5 times using temperature 0.1 and 5 times using temperature 0. At temperature of 0.1, the number of disagreement between any two rows is 14 out of 10,000 (0.14%). Even at temperature 0, we observed small differences across runs (2 out of 10,000 sentences). These findings confirm that both settings of temperature are not fully deterministic, and the level of variation is relatively low. This observation is consistent with recent literature, noting that a temperature of 0 does not guarantee fully deterministic outputs (Atil et al., 2025). We also searched the literature and most support that a temperature of 0.1 provides low variability and that small temperature differences yield similar outputs (Chen et al., 2023; Zong and Krishnamachari, 2023; Patel et al., 2024; Renze, 2024).

Given these findings—and the considerable time and cost associated with re-running the full dataset across all fiscal years—we have kept the Climate Solutions Measure at temperature 0.1.¹ Nonetheless, we have updated the manuscript to clearly explain this consideration relating to nondeterminism (see pages 76-77). Additionally, we attempt to mitigate the concern about nondeterminism in downstream results by conducting sensitivity analyses to test the robustness of our main findings. Specifically, we attempt to introduce random variation that resembles the 0.14% disagreement from the non-determinism analysis above and re-run the regression analyses in the paper. We do so by adding a normally distributed noise with a mean of zero and a standard deviation of 0.14%, 0.28% (2x stretch test), and 1.4% (10x stretch test) to Climate Solutions Measure across the full sample, and truncate values below zero to maintain interpretable results. These three variants have correlations of 99.98%, 99.93%, and 98.58% with the original Climate Solutions Measure, respectively. The regression results remain robust, providing some comfort regarding the potential impact of the non-determinism concern. We described this sensitivity analysis on page 77. We also include the regression results of this sensitivity analysis for the revenue growth analysis in Appendix Table 1 to this response document.

¹ Re-running the classification at temperature 0 for fiscal year 2022 took 4 days and approximately \$250 for the binary climate solutions task. Extending this to include the topic classification step would further increase time and cost. Applying both tasks to the full dataset from 2005 to 2022 is estimated to take several months and cost close to \$10,000.

We are grateful to the reviewer for highlighting this issue about nondeterminism, which prompted a more detailed evaluation of model consistency and improved transparency in our methodology. We hope these additional analyses help mitigate the concerns.

This is unfortunate because my major concerns regarding the analysis were well-addressed by the authors. There are only two points remaining. First, I still think there should be a test for alternative measures of climate risk beyond emissions. Second, I am not sure whether there could be even better strategies to outrule greenwashing. Even after the IRA, it would make sense for companies to increase their communication around climate solutions in search of attracting sustainable investors.

Response:

Thank you for recognizing the revisions made and for your two points.

First, in response to the suggestion to incorporate alternative measures of climate risk beyond emissions, we added a new test using scores from sustainability data providers (Refinitiv and MSCI), which reflect broader assessments of firms' carbon risk management. As suggested by another reviewer, we now interpret the weak correlation between climate risks and opportunities more cautiously and highlight this distinction as an important direction for future research. These updates are reflected on page 8, and we pasted this paragraph below for convenience.

As an additional supporting measure, we examine a firm's climate risk management using emissions scores from Refinitiv and MSCI in Supplementary Table 5 Panel E. Consistent with the greenhouse gas results, we find that the relationship between emissions scores and Climate Solutions Measure is modest (less than 0.15 standard deviations). Taken together, these empirical observations suggest that climate risks and opportunities are weakly related. Future research can further examine this potential distinction and explore the implications of different firms being exposed to climate risk versus innovating to capture climate opportunities.

Second, regarding the concern about greenwashing, on page 10, we have added a discussion in the limitations section acknowledging that even 10-K filings may contain strategic language around climate issues. However, we note that 10-K filings are more reliable than voluntary disclosures due to SEC oversight and legal enforcement, and we cite recent SEC enforcement actions to support this point. We hope that this added discussion—along with our extensive validation analyses showing that our Climate Solutions Measure is positively associated with green revenue, green patents, innovation metrics, and climate-solutions related policies—helps to mitigate this concern. In addition, following another reviewer's suggestion, we carefully reviewed the manuscript to ensure our language does not overstate the implications of the measure. In the results sections, we now refer to our measure more precisely as "climate solutions disclosed in 10-K" to better reflect its scope.

3. Conclusion

I remain positive about this paper. In my opinion, the work on the LLM method has not been conducted thoroughly enough. However, the results and implications might not change drastically with a reworked approach. Nevertheless, this work has to be done with a high degree of carefulness and rigor. The analysis seems to diverge towards a really useful state.

Response:

We sincerely thank the reviewer for the encouraging feedback and for recognizing the relevance and potential of the core idea. We agree that the work, particularly related to the LLM methodology, must be conducted with a high degree of carefulness and rigor. We appreciate the helpful comments that prompted us to clarify and evaluate more thoroughly on our methodology. Consistent with the reviewer's expectations, the results and implications remain unchanged, while at the same time, the additional tests and discussions help improve the reader's understanding of the utility of AI for the study of climate solutions in mandatory and regulated 10-K filings. We believe the manuscript is much improved thanks to the constructive feedback.

Reference:

Atil, B., Aykent, S., Chittams, A., Fu, L., Passonneau, R. J., Radcliffe, E., ... & Baldwin, B. (2025). Non-determinism of "deterministic" LLM settings. *arXiv*.

<https://arxiv.org/abs/2408.04667>

Chen, L., Zaharia, M., & Zou, J. (2024). How is ChatGPT's behavior changing over time?. *Harvard Data Science Review*, 6(2).

Oster, E. (2019). Unobservable selection and coefficient stability: Theory and evidence. *Journal of Business & Economic Statistics*, 37(2), 187-204.

Patel, D., Timsina, P., Raut, G., Freeman, R., levin, M. A., Nadkarni, G. N., ... & Klang, E. (2024). Exploring temperature effects on large language models across various clinical tasks. *medRxiv*, 2024-07.

Renze, M. (2024, November). The effect of sampling temperature on problem solving in large language models. In *Findings of the Association for Computational Linguistics: EMNLP 2024* (pp. 7346-7356).

Schimanski, T., Reding, A., Reding, N., Bingler, J., Kraus, M., & Leippold, M. (2024). Bridging the gap in ESG measurement: Using NLP to quantify environmental, social, and governance communication. *Finance Research Letters*, 61, 104979.

Zong, M., & Krishnamachari, B. (2023). Solving math word problems concerning systems of equations with GPT models. *Machine Learning with Applications*, 14, 100506.

Appendix 1: Climate Solutions Measure and Revenue Growth Sensitivity Test

	(1) Revenue Growth	(2) Revenue Growth	(3) Revenue Growth	(4) Revenue Growth
Climate Solutions Measure	0.003 ^{***} (3.82)			
Climate Solutions Measure (0.14% noise)		0.003 ^{***} (3.81)		
Climate Solutions Measure (0.28% noise)			0.003 ^{***} (3.78)	
Climate Solutions Measure (1.40% noise)				0.003 ^{***} (3.75)
Lagged Revenue (log)	-0.036 ^{***} (-15.23)	-0.036 ^{***} (-15.22)	-0.036 ^{***} (-15.22)	-0.036 ^{***} (-15.21)
Age (log)	-0.072 ^{***} (-15.00)	-0.072 ^{***} (-15.00)	-0.072 ^{***} (-15.01)	-0.072 ^{***} (-15.02)
Debt to Asset	-0.047 ^{***} (-3.96)	-0.047 ^{***} (-3.96)	-0.047 ^{***} (-3.96)	-0.047 ^{***} (-3.95)
CAPEX to Asset	0.935 ^{***} (9.41)	0.935 ^{***} (9.41)	0.935 ^{***} (9.41)	0.935 ^{***} (9.41)
ROA	-0.001 (-0.09)	-0.001 (-0.09)	-0.001 (-0.09)	-0.001 (-0.09)
N	32861	32861	32861	32861
Adj. R-squared	0.149	0.149	0.149	0.149
Gind-year FE	Yes	Yes	Yes	Yes
Clusters	Firm	Firm	Firm	Firm

t statistics in parentheses

* $p < .10$, ** $p < .05$, *** $p < .01$

Response to Reviewer Comments (Manuscript NCOMMS-24-53291B)

We would like to thank the reviewers for the insightful and constructive comments and guidance. They have been very helpful in revising and improving the paper. Below, we provide a detailed response addressing each of the remaining reviewer comments, where the comments are reproduced in boxed *italics*, followed by our response in a regular typeface.

Reviewer 1 comments:

1. The argument that ‘climate solutions represent a growing market under the rise of climate transition risks’ (page 10, lines 228f) is particularly compelling. To strengthen the overall framing and relevance of your paper, we recommend highlighting this point earlier in the manuscript, ideally in the introduction.

Response:

This is a great suggestion, we revised the first paragraph of the introduction to incorporate this important point:

Previous version:

However, the transition to a low-carbon economy is not merely a source of risks; it also presents business opportunities for firms innovating for the climate transition (Cenci et al., 2023). Examples include electric vehicle production, renewable energy generation, green hydrogen for heavy industry, and plant-based foods.

Revised version:

However, the transition to a low-carbon economy is not merely a source of risks; it also presents business opportunities for firms innovating for the climate transition (Cenci et al., 2023). These opportunities represent a growing market under rising climate transition risks, with examples including electric vehicle production, renewable energy generation, green hydrogen for heavy industry, and plant-based foods.

2. For the sake of clarity, we suggest being more explicit about how the climate opportunity measure you introduce is defined. A brief clarification would help readers understand whether the measure is based solely on the methodological approach or whether it results from the combination of the method and the specific data source used (i.e., 10-K filings).

Response:

We agree and more explicitly clarify that the measure is a combination of the methodology approach (apply LLM) and on the specific data source used (10-K Item 1). We highlight this when we first introduce our measure in the introduction:

Previous version:

In this paper, we measure firms engaging in climate-related business opportunities using data that is consistently disclosed over a long time period with regulatory scrutiny: business descriptions from regulated financial filings. In particular, we harness the power of large language models (LLMs) to present evidence of the prevalence and evolution of climate solutions for 4,483 US public firms across 13 GICS industry groups (47 GICS industries) from fiscal years 2005 to 2022 (reports released in calendar years 2006 to 2023).

Revised version:

In this paper, we develop a measure of climate solutions that draws on both the capabilities of large language models (LLMs) and uses data that is consistently disclosed over a long time period with regulatory scrutiny: business descriptions from regulated financial filings. We define climate solutions as products and services that develop or deploy technologies in a transition to a low-carbon economy. We present evidence of the prevalence and evolution of climate solutions for 4,483 US public firms across 13 GICS industry groups (47 GICS industries) from fiscal years 2005 to 2022 (reports released in calendar years 2006 to 2023).

**Referee Report on
"Tracking Business Opportunities for
Climate Solutions using AI in
Regulated Accounting Reports"**

October 19, 2024

1 Overview of the paper

The paper explores how advancements in Large Language Models (LLMs) can be leveraged to identify climate-related business opportunities in regulated financial disclosures. Focusing on over 39,710 10-K filings from 4,483 U.S. public firms between 2005 and 2022, the study identifies a growing trend in companies incorporating climate solutions—such as electric vehicles and renewable energy—into their product portfolios. Thereby, the paper introduces the Climate Solutions Measure. It shows that companies integrating climate solutions into their product offerings experience increased revenue growth and investment. Moreover, a topic analysis highlights a concentration on climate solutions with higher abatement potential or lower costs.

2 Analysis

The paper exhibits both strengths and weaknesses. There are three core strengths. First, the paper explores an important perspective of business opportunities of climate change through technological climate solutions. This shift of perspective is of high importance since it outlines a way to advance with problems of climate change, rather than focusing on a passive, mitigative, and often backward-looking perspective of risk. Second, the paper demonstrates a clever identification strategy of climate opportunities by analyzing the "Item 1 Business Description" section of 10-Ks. This section is under scrutiny by the U.S. SEC, auditors, and lawyers and is signed off by the CEO and CFO. Therefore, it is not just marketing communication but carries a higher-order validity. Third, the scope of investigations with the measure is suitable. Differentiating opportunities from risk, investigating the financial implications, as well as potential real-world effects of climate solutions, seem to be the core set of questions of interest in the field.

While I truly believe in the value of the paper, I have to outline a multitude of weaknesses. I will go into detail in the firm belief that this could help to improve the paper. I will structure the weaknesses in two major areas: the NLP methodology and the analysis conducted.

On the side of the NLP methodology, I trust that the method can successfully build a proxy for what it is intended to measure. For the core measure, the paper designs a binary classifier

and creates a dataset with active learning using ClimateBERT (Webersinke et al., 2022). Then, the gpt-3.5-turbo-1106 checkpoint is fine-tuned with the data. However, I find some uncertainties and shortcomings in the design and evaluation of the methodology. First, the labeling process is unclear to me. The paper states that "on average, two researchers annotate the same tasks to obtain some measure of dispersion" (lines 732-733). I can neither find the measure of dispersion, supposedly Cohen's Kappa, nor is it clear, what "on average" means in the context of labeling. Second, the paper argues with Stambach et al., 2023 for the choice of the dataset size. I do not fundamentally doubt the dataset size, but the largest model trained in the Stambach et al., 2023 paper is *RoBERTa_{large}* with 355 million parameters. GPT-3.5-turbo-1106 has an estimated 110-175 billion parameters. If you want to provide evidence about the sufficiency of the dataset size, you should follow the setup in Stambach et al., 2023 on the performance of a model on the development set as a function of training on different fractions of the training dataset (see Figure 6). Third, in my eyes, it is a best practice to disclose and explore all datasets and fine-tuning settings. This means the train-validation-test split for the training and evaluation needs to be described. Are the metrics calculated on the validation or test set? Furthermore, for the sake of reproducibility, the main hyperparameters for fine-tuning should be disclosed (at least batch size, learning rate, epochs). Finally, a holistic fine-tuning setup might profit by simply exploring more models than ClimateBERT. Candidates might be Distil-RoBERTa (Sanh et al., 2020) RoBERTa (Liu et al., 2019), their pre-trained sentence-level versions for the environmental domain (Schimanski et al., 2024), or DeBERTa (He et al., 2021). This is particularly useful for understanding the impact of different sizes (ClimateBERT has 82 million parameters, GPT-3.5-turbo-1106 has 110-175 billion parameters). Fourth, while I like the discussion about the determinism of GPT-3.5-turbo-1106, I do not understand why the model is set to temperature 0.1 and not to temperature 0. This should mitigate the non-deterministic behavior even more than 0.1 (it will still be non-deterministic). While this is an important detail, the evaluation strategy is even more important. The paper states that the "discrepancy observed between any two columns was six rows (0.17% of all rows in the dataset)". Does this mean, the the fine-tuned model was evaluated on the fine-tuning data? If this is the case, the results would be obscured by data leakage. Fifth, in my understanding, the paper uses the fine-tuned GPT-3.5-turbo-1106 on the topic modeling task. This might even decrease the performance on this task given it was trained

on a different classification task. A more comprehensive exploration might help the understanding. Collectively, I would recommend reworking the NLP methodology more thoroughly.

On the side of the analysis, I have the impression that a more solid argumentation and grounding in literature for measure and results is needed. Regarding the measure, a clearer placement in past NLP approaches to quantify and explore business opportunities in climate change is lacking. For instance, Sautner et al., 2023 derives a firm-level climate change opportunity exposure measure. This article is cited but only as a risk measure. This measure is also applied by other work (Ma et al., 2023). Toetzke et al., 2024 are also quantifying climate technology innovations with NLP – thereby delivering closely related research. Besides, there are approaches beyond NLP to assess business opportunities of climate change (Li et al., 2024). Reflecting on these past projects could help to refine the scope and novel contribution of the present work. Another factor that might need more justification is possible concerns about cheap talk or greenwashing. Mentioning these green technologies might entail reputational, legal, or financial advantages for companies.¹ Yet, we know that communication can be deceiving – subject to cheap talk or greenwashing (Bingler et al., 2024). Can you make a convincing argument that the measure is not susceptible to these patterns?

I find a similar pattern is visible in the results. A more solid argumentation and grounding in literature might be helpful. For instance, when assessing climate risk, carbon emissions are used. However, it is possible that your subset of highly innovative companies shows different emission patterns as shown by prior research (Cohen et al., 2020). If successfully innovating, a company's direct emissions might even rise due to more production. How does this play out as more climate risk? As you mentioned, the climate risk literature is more advanced, and other, clearer metrics might be more suitable. Another example of unclearities is your assumption that companies with higher climate solutions measure should demonstrate greater revenue growth. Why is this? While I would love to believe it is true, there is no backing for this claim. Notably, financing green projects is often rather associated with fewer returns in financial markets (Baker et al., 2022). Also, there is a potential for reverse causality at play. It might be that companies with high revenue growth can afford to invest in green technologies, rather than the revenue growth stemming from these technologies.

¹See for instance in the rapidly growing green fund landscape until 2022: <https://www.morningstar.com/sustainable-investing/2022-us-sustainable-funds-landscape-5-charts>.

More precise argumentation and grounding in literature could also help your regression analyses. For instance, it is unclear to me why the first regression runs without firm size when it might play a role in sustainability efforts and communication (Drempetic et al., 2019; Bissoondoyal-Bheenick et al., 2023). It is important to state that my concern is not that all of the above assumptions are wrong. Rather, I view that the current state of the paper lacks justifying and contextualizing its setup and results.

3 Conclusion

I genuinely like the idea of the paper. The approach seems promising and effective. However, when assessing the details of the paper, it comes down to two questions. First: Do I believe in your measure? Partially to yes. The creation and evaluation of the model and measure are not yet well-aligned with best practices. Second: Do I believe in your results? Partially, leaning towards no. I think, more thorough argumentation is needed, better grounding in literature, and clearer control for confounders.

References

- Baker, M., Bergstresser, D., Serafeim, G., and Wurgler, J. (2022). The pricing and ownership of us green bonds. *Annual Review of Financial Economics*, 14(Volume 14, 2022):415–437.
- Bingler, J. A., Kraus, M., Leippold, M., and Webersinke, N. (2024). How cheap talk in climate disclosures relates to climate initiatives, corporate emissions, and reputation risk. *Journal of Banking Finance*, 164:107191.
- Bissoondoyal-Bheenick, E., Brooks, R., and Do, H. X. (2023). Esg and firm performance: The role of size and media channels. *Economic Modelling*, 121:106203.
- Cohen, L., Gurun, U. G., and Nguyen, Q. H. (2020). The esg-innovation disconnect: Evidence from green patenting. Working Paper 27990, National Bureau of Economic Research.
- Drempetic, S., Klein, C., and Zwergel, B. (2019). The influence of firm size on the esg score: Corporate sustainability ratings under review. *Journal of Business Ethics*, 167:333 – 360.
- He, P., Liu, X., Gao, J., and Chen, W. (2021). Deberta: Decoding-enhanced bert with disentangled attention.
- Li, X., Luo, L., and Tang, Q. (2024). Climate risk and opportunity exposure and firm value: An international investigation. *Business Strategy and the Environment*, 33(6):5540–5562.
- Liu, Y., Ott, M., Goyal, N., Du, J., Joshi, M., Chen, D., Levy, O., Lewis, M., Zettlemoyer, L., and Stoyanov, V. (2019). Roberta: A robustly optimized bert pretraining approach.
- Ma, R., Yuan, R., and Fu, X. (2023). Climate change opportunity and corporate investment: Global evidence. *Journal of Climate Finance*, 3:100013.
- Sanh, V., Debut, L., Chaumond, J., and Wolf, T. (2020). Distilbert, a distilled version of bert: smaller, faster, cheaper and lighter.
- Sautner, Z., Van Lent, L., Vilkov, G., and Zhang, R. (2023). Firm-level climate change exposure. *The Journal of Finance*, 78(3):1449–1498.

-
- Schimanski, T., Reding, A., Reding, N., Bingler, J., Kraus, M., and Leippold, M. (2024). Bridging the gap in esg measurement: Using nlp to quantify environmental, social, and governance communication. *Finance Research Letters*, 61:104979.
- Stammach, D., Webersinke, N., Bingler, J. A., Kraus, M., and Leippold, M. (2023). Environmental claim detection.
- Toetzke, M., Probst, B., Feuerriegel, S., Anadon, L. D., and Hoffmann, V. (2024). Machine learning can help track climate technology innovation.
- Webersinke, N., Kraus, M., Bingler, J. A., and Leippold, M. (2022). Climatebert: A pretrained language model for climate-related text.

**Second Referee Report on
"Tracking Business Opportunities for
Climate Solutions using AI in
Regulated Accounting Reports"**

April 9, 2025

1 Overview of the Adjustments

The authors make a significant effort to improve the quality of the paper by including more literature, reinforcing key results with new experiments, and trying to advance the LLM methodology.

2 Analysis

However, there seems to be a persisting misconduct in training and using the models. Since this is the major building block of the paper for constructing their main contribution, the Climate Solution Measure, it is important that the construction is clear and flawless. Yet again, I have to outline several shortcomings. First, the authors seem to use two different methods in evaluating the main model, GPT-3.5-turbo-1106. On the one hand, they report a performance score on a single held-out test set. On the other hand, they report using a five-fold cross-validation. Which score is finally reported? And which score is used to compare to other models? If you are using the best fold, then the comparison is not valid. Second, when comparing to the BERT-based models, there seems to be an error in the training process. The extremely low results for RoBERTa (accuracy: 0.38) are extremely uncommon. It is most likely that something in the training code is wrong. This is a problem because the authors likely used the same code for the other BERT-based models. Thus, this makes me question the other results as well. Besides, despite stating that the authors compare to all reasonable models, they leave out the EnvironmentalBERT models (Schimanski et al., 2024) from the last iteration of reviews. In line with ClimateBERT, these should perform the best on this environmental/climate task and therefore are the best comparison points. Is there any reason to leave them out? While this comparison itself may be minor, the observations made here make me question the rigor applied to get the model usage completely right. Third, and along those same lines, changing the temperature to 0 for GPT-3.5-turbo-1106 represents a clear change in the model setup. The authors argue that "models disagreed on 10 out of 3,508 sentences, accounting for only 0.29% of the data". This means that, again, these results are obscured by data leakage. You cannot compare models trained on a dataset with the very dataset. The models literally know the answer through training. I want to outline that this is extremely important. The change in temperature to 0 means

that the Climate Solution Measure will likely change, and with that, potentially the downstream results of this study. I urge the authors to be very careful when revising the construction of their model and measures. It is likely needed to reestimate the entire Climate Solution Metric.

This is unfortunate because my major concerns regarding the analysis were well-addressed by the authors. There are only two points remaining. First, I still think there should be a test for alternative measures of climate risk beyond emissions. Second, I am not sure whether there could be even better strategies to outrule greenwashing. Even after the IRA, it would make sense for companies to increase their communication around climate solutions in search of attracting sustainable investors.

3 Conclusion

I remain positive about this paper. In my opinion, the work on the LLM method has not been conducted thoroughly enough. However, the results and implications might not change drastically with a reworked approach. Nevertheless, this work has to be done with a high degree of carefulness and rigor. The analysis seems to diverge towards a really useful state.

References

Schimanski, T., Reding, A., Reding, N., Bingler, J., Kraus, M., and Leippold, M. (2024). Bridging the gap in esg measurement: Using nlp to quantify environmental, social, and governance communication. *Finance Research Letters*, 61:104979.